# Analog In-memory Training on General Non-ideal Resistive Elements: The Impact of Response Functions

**Zhaoxian Wu**[∗]**, Quan Xiao**[∗]**, Tayfun Gokmen**[♯]**, Omobayode Fagbohungbe**[♯]**, Tianyi Chen**[†∗]

[∗]Cornell University, New York, NY
[♯]IBM T. J. Watson Research Center, Yorktown Heights, NY
[†]Rensselaer Polytechnic Institute, Troy, NY
`{zw868, qx232}@cornell.edu,`
`{tgokmen, Omobayode.Fagbohungbe}@us.ibm.com,`
`tianyi.chen@cornell.edu`

## Abstract

As the economic and environmental costs of training and deploying large vision or language models increase dramatically, analog in-memory computing (AIMC) emerges as a promising energy-efficient solution. However, the training perspective, especially its training dynamics, is underexplored. In AIMC hardware, the trainable weights are represented by the conductance of resistive elements and updated using consecutive electrical pulses. While the conductance changes by a constant in response to each pulse, in reality, the change is scaled by asymmetric and non-linear *response functions*, leading to a non-ideal training dynamics. This paper provides a theoretical foundation for gradient-based training on AIMC hardware with non-ideal response functions. We demonstrate that asymmetric response functions negatively impact `Analog SGD` by imposing an implicit penalty on the objective. To address the issue, we propose `residual learning` algorithm, which provably converges exactly to a critical point by solving a bilevel optimization problem. We demonstrate that the proposed method can be extended to address other hardware imperfections, such as limited response granularity. As we know, it is the first paper to investigate the impact of a class of generic non-ideal response functions. The conclusion is supported by simulations validating our theoretical insights.

## 1 Introduction

The remarkable success of large vision and language models is underpinned by advances in modern hardware accelerators, such as GPU, TPU [1], NPU [2], and NorthPole chip [3]. However, the computational demands of training these models are staggering. For instance, training LLaMA [4] cost $2.4 million, while training GPT-3 [5] required $4.6 million, highlighting the urgent need for more efficient computing hardware. Current mainstream hardware relies on the Von Neumann architecture, in which the physical separation of memory and processing units creates a bottleneck due to frequent, costly data movement between them.

In this context, the industry has turned its attention to *analog in-memory computing (AIMC) accelerators* based on resistive crossbar arrays [6–10], which excel at accelerating ubiquitous, computationally intensive matrix-vector multiplications (MVMs) operations. In AIMC hardware, the weights (matrices) are represented by the conductance states of the *resistive elements* in analog crossbar arrays [11, 12], while the input and output of MVM are analog signals like voltage and current. Leveraging

---

[∗]The work was done when the authors were at Rensselaer Polytechnic Institute. The work was supported by IBM through the IBM-Rensselaer Future of Computing Research Collaboration, the National Science Foundation Projects 2401297, 2532349, and 2532653, and by the Cisco Research Award.

39th Conference on Neural Information Processing Systems (NeurIPS 2025).

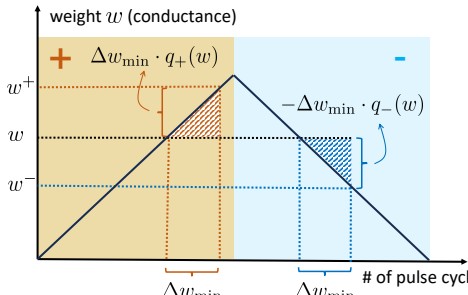 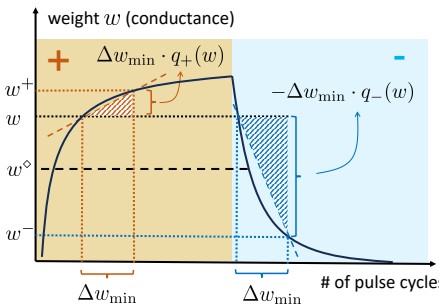

Figure 1: The weight's response curve. Positive and negative pulses are fired continuously on the left and right halves, respectively. One pulse is fired per cycle. Given $w$, the weight becomes $w^+$ or $w^-$ after one positive and negative pulse, respectively. The response factors $q_+(w)$ and $q_-(w)$ are approximately the slope of the curve at $w$, and $\Delta w_{\min}$ is the response granularity. **(Left)** Ideal response functions $q_+(w) \equiv q_-(w) \equiv 1$. Every point is a symmetric point. **(Right)** Asymmetric response functions $q_+(w) \neq q_-(w)$ almost everywhere expect for the symmetric point $w^\diamond$.

Kirchhoff's and Ohm's laws, AIMC hardware achieves $10\times$-$10,000\times$ energy efficiency than GPU [13–15] in model inference.

Despite its high efficiency, *analog training* is considerably more challenging than *inference* since it involves frequent weight updates. Unlike digital hardware, where the weight increment can be applied to the original weight in the memory cell, the weights in AIMC hardware are changed by the so-called *pulse update*.

**Pulse update.** When receiving electrical pulses from its peripheral circuits, the resistive elements change their conductance in response to the pulse polarity [16]. Receiving a pulse at each pulse cycle, the conductance is updated by $\Delta w_{\min} \cdot q_+(w)$ or $\Delta w_{\min} \cdot q_-(w)$, depending on the pulse polarity, where $\Delta w_{\min}$ is *response granularity*, and $q_+(w)$ and $q_-(w)$ are *response functions*. Geometrically, $q_+(w)$ and $q_-(w)$ are the slopes of *response curves*; see Figure 1. All $\Delta w_{\min}$, $q_+(w)$, and $q_-(w)$ are element-specific parameters or functions that are set before training and hence remain fixed during training. Typically, $\Delta w_{\min}$ is known while $q_+(w)$ and $q_-(w)$ are not.

**Gradient-based training implemented by analog update.** Supported by pulse update, the gradient-based training algorithms are used to optimize the weights. Consider a standard training problem with objective $f(\cdot) : \mathbb{R}^D \to \mathbb{R}$ and a model parameterized by $W \in \mathbb{R}^D$

$$W^* := \underset{W \in \mathbb{R}^D}{\arg\min}\ f(W) := \mathbb{E}_\xi[f(W; \xi)] \tag{1}$$

where $\xi$ is a random data sample. Similar to stochastic gradient descent (SGD) in digital training (`Digital SGD`), the gradient-based training algorithm on AIMC hardware, `Analog SGD`, updates the weights by stochastic gradients $\nabla f(W_k; \xi_k)$. `Digital SGD` updates the weight by $W_{k+1} = W_k - \alpha \nabla f(W_k; \xi_k)$ with learning rate $\alpha$. Given a *desired update* $\Delta W = -\alpha \nabla f(W_k; \xi_k)$, AIMC hardware implements `Analog SGD` by sending $|[\Delta W]_d|/\Delta w_{\min}$ pulses to the $d$-th element. Ideally, $q_+(w) = q_-(w) = 1$ for every conductance states. If so, with each pulse updating $[W_k]_d$ by $\Delta w_{\min}$, $[W_k]_d$ is ultimately updated by about $[\Delta W]_d$.

**Challenges of analog training.** Despite its ultra-efficiency, gradient-based training on AIMC hardware is challenging. First, the generic response functions are *asymmetric* (i.e. $q_+(w) \neq q_-(w)$), and *non-linear* [17–19]. Due to the variation of response functions and conductance states, gradients are scaled by different magnitudes across different coordinates, leading to biased gradients. Furthermore, the response granularity $\Delta w_{\min}$ is a constant. When the gradients or the learning rate decay below $\Delta w_{\min}$, pulse update no longer provides sufficient precision to perform gradient descent [20]. Other imperfections include, but are not limited to, noisy input/output (IO) of MVM operations and analog-digital conversion error [18]. This paper aims to investigate the impact of non-ideal response functions and develop a method to mitigate their negative effects. We also discuss extending the proposed method to deal with other hardware imperfections.

## 1.1 Main results

Complementing existing empirical studies in analog in-memory computing, this paper aims to build a rigorous theoretical foundation of analog training. By introducing bias to the gradient, the asymmetric

response function plays a central role in differentiating digital and analog training. In contrast, the other non-idealities hinder the training process by causing precision-related issues. Therefore, we approach the problem progressively, beginning with a simplified case that involves only the asymmetric response functions, and extending the proposed methods to more general scenarios.

As a warm-up, building upon the pulse update mechanism, we propose the following discrete-time mathematical model to characterize the trajectory of `Analog SGD`

$$\texttt{Analog SGD} \qquad W_{k+1} = W_k - \alpha \nabla f(W_k; \xi_k) \odot F(W_k) - \alpha |\nabla f(W_k; \xi_k)| \odot G(W_k) \quad (2)$$

where $\alpha > 0$ is the learning rate and $\xi_k$ is the data sample of iteration $k$; $|\cdot|$ and $\odot$ represent the element-wise absolute value and multiplication, respectively; and $F(\cdot)$ and $G(\cdot)$ are hardware-specific matrix which are defined by $q_+(\cdot)$ and $q_-(\cdot)$. In Section 2, we will explain the underlying rationale of (2). Compared with the standard `Digital SGD`, the gradients in (2) are scaled by $F(\cdot)$ and an extra bias term is introduced. Typically, hardware imperfections lead to non-ideal response functions, i.e., $F(\cdot) \not\equiv 1$ and $G(\cdot) \not\equiv 0$. Thus, we ask a natural question that

**Q1)** *What is the impact of non-ideal response functions and how to alleviate it?*

Recently, [21] partially answers the question by showing that `Analog SGD` suffers from a convergence issue due to the asymmetric update, and a heuristic algorithm, `Tiki-Taka` [22–24], converges exactly by reducing the weight drift. However, their work is limited to a special case of *linear response functions*, which are in the form of $q_+(w) = 1 - w/\tau, q_-(w) = 1 + w/\tau$ with hardware-specific parameter $\tau > 0$. Given more general $q_+(w)$ and $q_-(w)$, the convergence of `Tiki-Taka` does not trivially hold, even though the response functions are still linear.

**Gap between theory for special linear and generic response functions.** Consider a more generic linear response $q_+(w) = (1 + c_{\texttt{Lin}})(1 - w/\tau), q_-(w) = (1 - c_{\texttt{Lin}})(1 + w/\tau)$ with a parameter $c_{\texttt{Lin}}$, which reduces to the setting in [21] when $c_{\texttt{Lin}} = 0$. Figure 2 shows the damage from a non-zero $c_{\texttt{Lin}}$ to `Tiki-Taka`. Consistent with the conclusion in [21], `Tiki-Taka` significantly outperforms `Analog SGD` when $c_{\texttt{Lin}} = 0$. However, when $c_{\texttt{Lin}}$ is perturbed from $0.1$ to $0.3$, `Tiki-Taka` degrades dramatically and even becomes worse than `Analog SGD` does. The modification is slight, but the convergence guarantee in [21] fails, and the convergence of `Tiki-Taka`

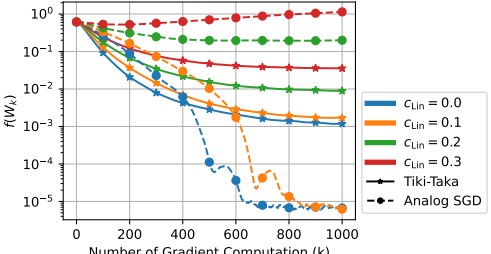

Figure 2: Comparison of `Analog SGD` and `Tiki-Taka` under different parameter $c_{\texttt{Lin}}$. The error plateau in the order $10^{-5}$ comes from the limited response granularity $\Delta w_{\min} = 10^{-4}$.

is harmed significantly. This counter-example indicates a gap between the theory for special linear and generic response functions, and necessitates the study of the analog training with generic response functions and the exploration of exact convergence conditions.

Ignoring other imperfections temporarily, this paper first analyzes the impact of response functions. We show that `Analog SGD` suffers from asymptotic error due to the mismatch between the algorithmic *stationary point* and physical *symmetric point*. Inspired by that, we propose a novel algorithm framework that aligns two points, overcoming the asymmetric issues. Building on that, we endeavor to extend the proposed algorithm to more practical scenarios that involve other imperfections like limited granularity and noisy readings, prompting a second critical question:

**Q2)** *How to extend the framework to address the limited response granularity and noisy IO issues?*

To answer this question, we propose two mechanisms to further overcome these two issues.

**Our contributions.** This paper makes the following contributions:

   **C1)** Building on the pulse update equation, we propose an approximate discrete-time dynamics for analog update. Enabled by this, we study the impact of response functions directly, without being limited to specific element candidates.

   **C2)** Based on that, we show that instead of optimizing $f(\cdot)$, `Analog SGD` optimizes another penalized objective implicitly. An implicit penalty is introduced by the asymmetric response functions, which attract the weights towards symmetric points. Consequently, `Analog SGD` can only converge to the optimal point inexactly.

**C3)** We propose a novel `Residual Learning` theoretical framework to alleviate the asymmetric update and implicit penalty issues. `Residual Learning` explicitly introduces another *residual array*, which has a stationary point 0. This framework leads to `Tiki-Taka` heuristically proposed in [22] while it offers an understanding of how `Tiki-Taka` deals with the challenge from generic response functions. By properly zero-shifting so that the stationary and symmetric points overlap, `Residual Learning` provably converges to a critical point.

**C4)** Building on C3), we propose a variant, `Residual Learning` v2, tailored for more practical training scenarios. We propose introducing a digital buffer to filter out reading errors caused by IO noise. Furthermore, we propose a threshold-based transfer rule to alleviate instability caused by limited granularity.

## 1.2 Prior art

**AIMC training.** Analog training has shown promising early successes with tremendous energy advantage [25, 26]. Among them, on-chip training, which performs forward, backward, and update directly on analog chips [22–24, 27, 28] is considered to be the most efficient paradigm, but it is more sensitive to hardware imperfections. Sacrificing energy efficiency for robustness, hybrid digital-analog off-chip training is proposed [29–32], which offloads some computation burden to digital components. This paper focuses on the more challenging on-chip training setting.

**Energy-based model and equilibrium propagation.** AIMC training leverages back-propagation to compute the gradient signals. Recently, a class of energy-based models has been studied, which performs equilibrium propagation to compute gradient signals [33–37]. Focusing on the training dynamics instead of concrete gradient computing, our work is orthogonal to them and is expected to provide insight for algorithm designs of energy-based model training.

## 2 Analog Training with Generic Response Functions

This section examines the discrete-time dynamics of analog training and introduces the challenges posed by generic response functions. After that, we introduce a family of response functions that reflect crucial physical properties that interest us.

**Compact formulations of analog update.** We first investigate the dynamics of one element $w$ in $W \in \mathbb{R}^D$. This paper adopts $w$ to represent the element of the weight $W_k$ without specifying its index. As we discuss in Section 1, the response granularity $\Delta w_{\min}$ is scaled by the response functions $q_+(w)$ or $q_-(w)$. Since a desired update $\Delta w$ requires a series of pulses with each scaled by approximately $q_+(w)$ or $q_-(w)$, it is sensible that the $\Delta w$ is approximately scaled by $q_+(w)$ or $q_-(w)$ as well. Accordingly, we propose that an approximate dynamics of analog update is given by $w' \approx U_q(w, \Delta w)$, where $U_q(w, \Delta w)$ is defined by

$$U_q(w, \Delta w) := \begin{cases} w + \Delta w \cdot q_+(w), & \Delta w \geq 0, \\ w + \Delta w \cdot q_-(w), & \Delta w < 0. \end{cases} \tag{3}$$

The update (3) holds at each resistive element. At the $k$-th iteration, We stack all the weights $w_k$ and expected increment $\Delta w_k$ together into vectors $W_k, \Delta W_k \in \mathbb{R}^D$. Similarly, the response functions $q_+(\cdot)$ and $q_-(\cdot)$ are stacked into $Q_+(\cdot)$ and $Q_-(\cdot)$, respectively. Let the notation $U_Q(W_k, \Delta W)$ on matrices $W_k$ and $\Delta W$ denote the element-wise operation on $W_k$ and $\Delta W$, i.e. $[U_Q(W_k, \Delta W)]_d := U_{[Q]_d}([W_k]_d, [\Delta w]_d), \forall d \in [D]$ with $[D] := \{1, 2, \cdots, D\}$ denoting the index set. The element-wise update (3) can be expressed as $W_{k+1} = U_Q(W_k, \Delta W_k)$. Leveraging the symmetric decomposition [21, 22], we decompose $Q_-(W)$ and $Q_+(W)$ into symmetric component $F(\cdot)$ and asymmetric component $G(\cdot)$

$$F(W) := (Q_-(W) + Q_+(W))/2, \quad \text{and} \quad G(W) := (Q_-(W) - Q_+(W))/2, \tag{4}$$

which leads to a compact form of the `Analog Update`

$$\text{Analog Update} \qquad W_{k+1} = W_k + \Delta W_k \odot F(W_k) - |\Delta W_k| \odot G(W_k). \tag{5}$$

**Gradient-based training algorithms on AIMC hardware.** In (5), the desired update $\Delta W_k$ varies based on different algorithms. Replacing $\Delta W_k$ with the stochastic gradient $\nabla f(W_k; \xi_k)$, we obtain

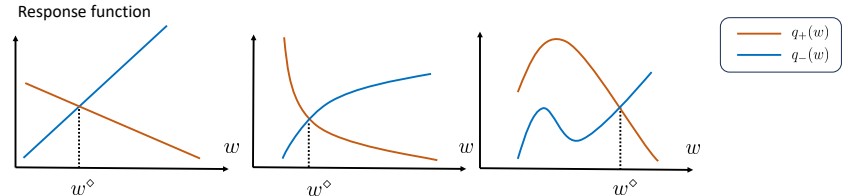

Figure 3: Examples of response functions from Definition 1; $w^\diamond$ is the symmetric point.

the dynamics of `Analog SGD` shown in (2). This update is reduced to the mathematical form for linear response functions in [21] as a special case; see Appendix B for details.

**Response function class.** Before proceeding to the study of response functions, we first define the response function class that interests us. Since the behavior of resistive elements is always governed by physical laws, the function class should reflect certain crucial physical properties.

The most crucial property of the response functions is the *asymmetric update*, i.e., $q_-(w) \neq q_+(w)$ for most of $w$. Specifically, if a point $w^\diamond$ satisfies $q_-(w^\diamond) = q_+(w^\diamond)$, we say $w^\diamond$ is a *symmetric point*. Stacking all $w^\diamond$ into a vector $W^\diamond \in \mathbb{R}^D$. Observe that the function $G(W)$ is large if $q_-(w)$ and $q_+(w)$ are significantly different, while it is almost zero around $W^\diamond$. At the same time, $F(W)$ is the average of the response functions in two directions. As we will see in Sections 3.2 and 4, the ratio $\frac{G(W)}{\sqrt{F(W)}}$ plays a critical role in the convergence behaviors.

In addition to the asymmetric update, the function class should possess other properties. First, the conductance increases upon receipt of a positive pulse, and vice versa, resulting in positive response functions. In addition, we assume that the response functions are differentiable (and hence continuous) for mathematical tractability. Taking all factors into account, we define the following class of response functions.

**Definition 1** (Response function class). $q_+(\cdot)$ *and* $q_-(\cdot)$ *satisfy*

- *(**Positive-definiteness**) There exist positive constants $q_{\min} > 0$ and $q_{\max} > 0$ such that $q_{\min} \leq q_+(w) \leq q_{\max}$ and $q_{\min} \leq q_-(w) \leq q_{\max}, \forall w$; and,*

- *(**Differentiable**) The response functions $q_+(\cdot)$ and $q_-(\cdot)$ are differentiable.*

Definition 1 covers a wide range of response functions, including but not limited to PCM, ReRAM, ECRAM, and others mentioned in Section A. Figure 3 showcases three examples from the response functions class, including linear, non-linear but monotonic, and even non-monotonic functions.

## 3 Implicit Penalty and Inexact Convergence of Analog SGD

This section introduces a critical impact of the response functions, *implicit penalized objective*. Affected by this, `Analog SGD` can only converge inexactly with a non-diminishing asymptotic error.

### 3.1 Implicit penalty

We first give an intuition through a situation where $W_k$ is already a critical point, i.e., $\mathbb{E}_\xi[\nabla f(W_k; \xi)] = 0$. Recall that stochastic gradient descent on digital hardware (`Digital SGD`) is stable in expectation, i.e. $\mathbb{E}_{\xi_k}[W_{k+1}] = W_k - \mathbb{E}_{\xi_k}[\alpha \nabla f(W_k; \xi_k)] = W_k$. However, this does not work for `Analog SGD`

$$\mathbb{E}_{\xi_k}[W_{k+1}] = W_k - \mathbb{E}_{\xi_k}[\alpha \nabla f(W_k; \xi_k) \odot F(W_k) - \alpha |\nabla f(W_k; \xi_k)| \odot G(W_k)] \quad (6)$$
$$= W_k - \alpha \mathbb{E}_{\xi_k}[|\nabla f(W_k; \xi_k)| \odot G(W_k)] \neq W_k.$$

Consider a simplified version that the weight is a scalar ($D = 1$) and the function $G(W)$ is strictly monotonically decreasing[2] to help us gain intuition on the impact of the drift in (6). Recall $G(W^\diamond) = 0$ at the symmetric point $W^\diamond$. $G(W) > 0$ when $W > W^\diamond$ and $G(W) < 0$ otherwise. Consequently, (6) indicates that $\mathbb{E}_{\xi_k}[W_{k+1}] < W_k$ when $W_k > W^\diamond$ and $\mathbb{E}_{\xi_k}[W_{k+1}] > W_k$ otherwise. It implies that $W_k$ suffers from a drift tendency towards $W^\diamond$. In addition, the penalty coefficient proportional

---

[2]It happens when both $q_+(\cdot)$ and $q_-(\cdot)$ are strictly monotonic.

to the noise level since the drift is proportional to $\mathbb{E}_{\xi_k}[|\nabla f(W_k; \xi_k)|]$, which is the first moment of noise $\mathbb{E}_{\xi_k}[|\nabla f(W_k; \xi_k) - \mathbb{E}_\xi[\nabla f(W_k; \xi)]|]$ in essence.

The following theorem formalizes the implicit penalty effect. Before that, we define an accumulated asymmetric function $R_c(\cdot) : \mathbb{R}^D \to \mathbb{R}^D$, whose derivative is $R(W) := \frac{G(W)}{F(W)}$, i.e. $\frac{d[R_c(W)]_d}{d[W]_d} = [R(W)]_d = \frac{[G(W)]_d}{[F(W)]_d}$. If $R(W)$ is strictly monotonic, $R_c(W)$ reaches its minimum at the symmetric point $W^\diamond$ where $R(W^\diamond) = 0$, so that it penalizes the weight away from the symmetric point.

**Theorem 1** (Implicit penalty, short version). *Suppose $W^*$ is the unique minimizer of problem* (1). *Let* $\Sigma := \mathbb{E}_\xi[|\nabla f(W^*; \xi)|] \in \mathbb{R}^D$. Analog SGD *implicitly optimizes the following penalized objective*

$$\min_W \ f_\Sigma(W) := f(W) + \langle \Sigma, R_c(W) \rangle. \tag{7}$$

The full version of Theorem 1 and its proof are deferred to Appendix G. In Theorem 1, $R_c(W)$ plays the role of a penalty to force the weight towards a symmetric point. As shown in Appendix G, $R_c(W)$ has a simple expression on linear response functions when $c_{\text{Lin}} = 0$, leading (7) to $\min_W \ f_\Sigma(W) := f(W) + \frac{\Sigma}{2\tau}\|W\|^2$ which is an $\ell_2$ regularized objective. In addition, the implicit penalty has a coefficient proportional to the noise level $\Sigma$ and inversely proportional to the dynamic range $\tau$. It implies that the implicit penalty becomes active only when gradients are noisy, and the noise amplifies the effect.

> With noisy gradients, an **implicit penalty** attracts Analog SGD towards symmetric points.

## 3.2 Inexact Convergence of Analog SGD under generic devices

Due to the implicit penalty, Analog SGD only converges to a critical point inexactly. Before showing that, We introduce a series of assumptions on the objective, as well as noise.

**Assumption 1** (Objective). *The objective $f(W)$ is $L$-smooth and is lower bounded by $f^*$.*

**Assumption 2** (Unbiasness and bounded variance). *The stochastic gradient is unbiased and has bounded variance $\sigma^2$. i.e., $\mathbb{E}_\xi[\nabla f(W; \xi)] = \nabla f(W)$ and $\mathbb{E}_\xi[\|\nabla f(W; \xi) - \nabla f(W)\|^2] \le \sigma^2$.*

Assumption 1–2 are standard in non-convex optimization [38]. This paper considers the average squared norm of the gradient as the convergence metric, given by $E_K^{\text{ASGD}} := \frac{1}{K}\sum_{k=0}^{K-1}\|\nabla f(W_k)\|^2$. Now, we establish the convergence of Analog SGD.

**Theorem 2** (Inexact convergence of Analog SGD). *Under Assumption 1–2, if the learning rate is set as $\alpha = O(1/\sqrt{K})$, it holds that*

$$E_K^{ASGD} \le O\left(\sqrt{\sigma^2/K} + \sigma^2 S_K^{ASGD}\right) \tag{8}$$

*where $S_K^{ASGD}$ denotes the amplification factor given by $S_K^{ASGD} := \frac{1}{K}\sum_{k=0}^{K-1}\left\|\frac{G(W_k)}{\sqrt{F(W_k)}}\right\|_\infty^2$.*

The proof of Theorem 2 is deferred to Appendix H. Theorem 2 suggests that the convergence metric $E_K^{\text{ASGD}}$ is upper bounded by two terms: the first term vanishes at a rate of $O(\sqrt{\sigma^2/K})$, which matches the Digital SGD's convergence rate [38] up to a constant; the second term contributes to the *asymptotic error* of Analog SGD, which does not vanish with the number of iterations $K$.

**Impact of saturation/asymmetric update.** The exact expression of $S_K^{\text{ASGD}}$ depends on the specific noise distribution and thus is difficult to reach. However, $S_K^{\text{ASGD}}$ reflects the saturation degree near the critical point $W^*$ when $W_k$ converges to a neighborhood of $W^*$. If $W^*$ is far from the symmetric point $W^\diamond$, $S_K^{\text{ASGD}}$ becomes large, leading to a large $E_K^{\text{ASGD}}$ and a large asymptotic error. In contrast, if $W^*$ remains close to the symmetric point $W^\diamond$, the asymptotic error is small.

## 4 Mitigating Implicit Penalty by Residual Learning

The asymptotic error in Analog SGD is a fundamental issue that arises from the mismatch between the symmetric point and the critical point. An idealistic remedy for the inexact convergence is carefully shifting the weights to ensure the stationary point is close to a symmetric point. However, determining

the appropriate shifting is challenging, as the critical point is unknown before training. Therefore, an ideal solution to address this issue is to jointly construct a sequence with a proper stationary point and a proper shift of the symmetric point.

**Residual learning.** Our solution overlaps the algorithmic stationary point and physical symmetric point on the special point 0. Besides the main analog array, $W_k$, we maintain another array, $P_k$, whose stationary point should be 0. A natural choice is the *residual* of the weight, $P^*(W)$, defined by the $P$ that minimizes the objective $f(W + \gamma P)$ with a non-zero $\gamma$. Notice that $P^*(W_k) \to 0$ as $W_k \to W^*$. Additionally, the goal of the main array is to minimize the residual so that the model $W_k$ approaches optimality. This process can be formulated as the following bilevel problem, whose optimal points can be proved to be those of $f(W)$

Residual Learning $\quad \min_{W \in \mathbb{R}^D} \|P^*(W)\|^2, \quad \text{s.t. } P^*(W) \in \arg\min_{P \in \mathbb{R}^D} f(W + \gamma P).$ (9)

Now we propose a gradient-based method to solve (9). The stochastic gradient of $f(W + \gamma P)$ with respect to $P$, given by $\nabla_P f(W + \gamma P; \xi) = \gamma \nabla f(W + \gamma P; \xi)$, is accessible with fair expense, enabling us to introduce a sequence $P_k$ to track the residual of $W_k$ by optimizing $f(W_k + \gamma P)$

$$P_{k+1} = P_k - \alpha \nabla f(\bar{W}_k; \xi_k) \odot F(P_k) - \alpha |\nabla f(\bar{W}_k; \xi_k)| \odot G(P_k).$$ (10)

where $\bar{W}_k := W_k + \gamma P_k$ is the mixed weight. We then derive the hyper-gradient of the upper-level objective. Notice $\nabla \|P^*(W)\|^2 = 2\nabla P^*(W) P^*(W)$. Assuming $W^*$ is the unique minimum of $f(\cdot)$, we know $P^*(W)$ satisfies $\gamma P^*(W) + W = W^*$. Taking gradient with respective to $W$ on both sides, we have $\nabla P^*(W) = -\frac{1}{\gamma} I$ and hence $\nabla \|P^*(W)\|^2 = -\frac{2}{\gamma} P^*(W)$. Approximating $P^*(W)$ by $P_k$ and absorbing $\frac{2}{\gamma}$ into the learning rate $\beta$, we reach the update of the main array

$$W_{k+1} = W_k + \beta P_{k+1} \odot F(W_k) - \beta |P_{k+1}| \odot G(W_k).$$ (11)

Featuring moving the residual $P_k$ to $W_k$, (11) is referred to as *transfer* process. The updates (10) and (11) are performed alternatively until convergence. `Tiki-Taka` mentioned in [21] is the special case with linear response functions and $\gamma = 0$.

On the response functions side, it is naturally required to let zero be a symmetric point, i.e., $G(0) = 0$, which can be implemented by the zero-shifting technique [39] by subtracting a reference array.

**Convergence properties of `Residual Learning`.** We begin by analyzing the convergence of `Residual Learning` without considering the zero-shift first, which enables us to understand how zero-shifted response functions affect convergence.

If the optimal point $W^*$ exists and is unique, the solution of the lower-level objective has a closed form $P^*(W) := \frac{W^* - W}{\gamma}$. At that time, the upper-level objective equals $\|W^* - W\|^2$. However, the solutions of $f(\cdot)$ are generally non-unique, especially for non-convex objectives with multiple local minima. To ensure the existence and uniqueness of $W^*$, we assume the objective is strongly convex.

**Assumption 3** ($\mu$-strong convexity)**.** *The objective $f(W)$ is $\mu$-strongly convex.*

Under the strongly convex assumption, the optimal point $W^*$ is unique and hence the optimal solution of the lower-level problem in (9) is unique. Since the requirement of strong convexity is non-essential in the development of bilevel optimization [40–43], we believe the proof can be extended to more general cases and will extend it for future work.

Involving two sequences $W_k$ and $P_k$, `Residual Learning` converges in different senses, including: (a) the residual array $P_k$ converges to the optimal point $P^*(W_k)$; (b) $W_k$ converges to the critical point of $f(\cdot)$ or the optimal point $W^*$; (c) the sum $\bar{W}_k = W_k + \gamma P_k$ converges to a critical point where $\nabla f(\bar{W}_k) = 0$. Taking all these into account, we define the convergence metric as

$$E_K^{\text{RL}} := \frac{1}{K} \sum_{k=0}^{K-1} \mathbb{E}\left[\|\nabla f(\bar{W}_k)\|^2 + O(\|P_k - P^*(W_k)\|^2) + O(\|W_k - W^*\|^2)\right].$$ (12)

For simplicity, the constants in front of some terms in $E_K^{\text{RL}}$ are hidden. Now, we provide the convergence of `Residual Learning` with generic responses.

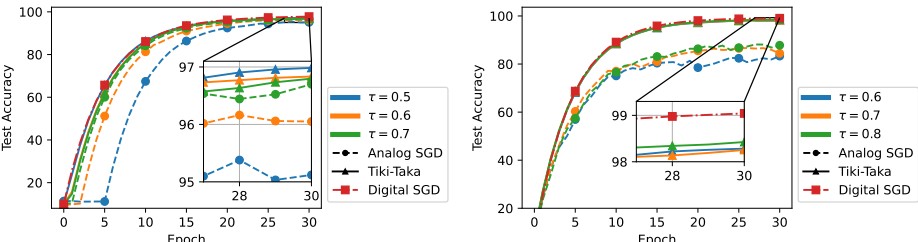

Figure 4: Test accuracy of training on MNIST dataset under different $\tau$; **(Left)** FCN. **(Right)** CNN.

**Theorem 3** (Convergence of `Residual Learning`). *Under Assumptions 1–3, with the learning rate* $\alpha = O\left(\sqrt{1/\sigma^2 K}\right)$, $\beta = O(\alpha\gamma^{3/2})$, *it holds for* `Residual Learning` *that*

$$E_K^{RL} \leq O\left(\sqrt{\sigma^2/K} + \sigma^2 S_K^{RL}\right) \tag{13}$$

*where* $S_K^{RL}$ *denotes the amplification factor of* $P_k$ *given by* $S_K^{RL} := \frac{1}{K}\sum_{k=0}^{K} \left\|\frac{G(P_k)}{\sqrt{F(P_k)}}\right\|_\infty^2$.

The proof of Theorem 3 is deferred to Appendix I. Theorem 3 claims that `Residual Learning` converges at the rate $O\left(\sqrt{\sigma^2/K}\right)$ to a neighbor of critical point with radius $O(\sigma^2 S_K^{RL})$, which share almost the same expression with the convergence of `Analog SGD`. The difference lies in the amplification factor $S_K^{RL}$ and $S_K^{ASGD}$, where the former depends on $P_k$ while the latter depends on $W_k$.

> **Impact of response functions.** Response function affects the `Analog SGD` and `Residual Learning` similarly. However, attributed to the residual array, constructing response functions to enable exact convergence of `Residual Learning` is viable.

As we have discussed, $P_k$ tends to $P^*(W_k)$ which tends to $0$ given $W_k$ tends to $W^*$. Therefore, response functions with $G(P) = 0$ when $P = 0$ are required for the exact convergence.

**Assumption 4.** *(Zero-shifted symmetric point)* $P = 0$ *is a symmetric point, i.e.* $G(0) = 0$.

Under it and the Lipschitz continuity of the response functions, it holds directly that $\left\|\frac{G(P_k)}{\sqrt{F(P_k)}}\right\|_\infty \leq L_S\|P_k\|_\infty$ for a constant $L_S \geq 0$. Consequently, when $P_k \to P^*(W_k) \to 0$ as $W_k \to W^*$, the asymptotic error disappears. Formally, the following corollary holds true.

**Corollary 1** (Exact convergence of `Residual Learning`). *Under Assumption 4 and the conditions in Theorem 3, if* $\gamma \geq \Omega(q_{min}^{-2/5})$, *it holds that* $E_K^{RL} \leq O\left(\sqrt{\sigma^2 L/K}\right)$.

The proof of Corollary 1 is deferred to Appendix I.5. Corollary 1 demonstrates the failure of `Tiki-Taka` in Figure 2. The symmetric point is $w^\diamond = c_{Lin}\tau$ in this example, which violates Assumption 4 when $c_{Lin} \neq 0$ and hence introduces asymptotic error into `Residual Learning`.

## 5 Extension of Residual Learning: limited granularity and noisy IO

This section extends `Residual Learning` to practical scenarios with additional hardware imperfections. To be specific, we consider the *noisy IO* and *limited granularity* as examples. We highlight that we are not trying to diminish the importance of imperfection, but rather focus on two of the primary ones known to be crucial.

IO of resistive crossbar arrays introduces noise during the reading of $P_{k+1}$ in the transfer process (11), given by $W_{k+1} = W_k + \beta(P_{k+1} + \varepsilon_k) \odot F(W_k) - \beta|P_{k+1} + \varepsilon_k| \odot G(W_k)$ with a noise $\varepsilon_k$. It incurs the implicit penalty issues again, leading to a penalized upper-level objective $\|P^*(W)\|^2 + \langle \Sigma_\varepsilon, R_c(W)\rangle$, as claimed by Theorem 1, where $\Sigma_\varepsilon = \mathbb{E}[|\varepsilon_k|]$ is assumed to be a constant. To filter out the noise, we propose to use a digital buffer $H_k$ to take a moving average of noisy $P_{k+1}$ signals by

$$H_{k+1} = (1-\beta)H_k + \beta(P_{k+1} + \varepsilon_{k+1}). \tag{14}$$

| | CIFAR10 | | | | |
|---|---|---|---|---|---|
| | DSGD | ASGD | TT/RL | TTv2 | RLv2 |
| ResNet18 | $95.43\pm0.13$ | $84.47\pm3.40$ | $94.81\pm0.09$ | $95.31\pm0.05$ | $95.12\pm0.14$ |
| ResNet34 | $96.48\pm0.02$ | $95.43\pm0.12$ | $96.29\pm0.12$ | $96.60\pm0.05$ | $96.42\pm0.13$ |
| ResNet50 | $96.57\pm0.10$ | $94.36\pm1.16$ | $96.34\pm0.04$ | $96.63\pm0.09$ | $96.56\pm0.08$ |
| | CIFAR100 | | | | |
| | DSGD | ASGD | TT/RL | TTv2 | RLv2 |
| ResNet18 | $81.12\pm0.25$ | $68.98\pm1.01$ | $76.17\pm0.23$ | $78.56\pm0.29$ | $79.83\pm0.13$ |
| ResNet34 | $83.86\pm0.12$ | $78.98\pm0.55$ | $80.58\pm0.11$ | $81.81\pm0.15$ | $82.85\pm0.19$ |
| ResNet50 | $83.98\pm0.11$ | $79.88\pm1.26$ | $80.80\pm0.22$ | $82.82\pm0.33$ | $83.90\pm0.20$ |

Table 1: Fine-tuning ResNet models with the *power response* on CIFAR10/100. Test accuracy is reported. DSGD, ASGD, TT/RL, TTv2, and RLv2 represent Digital SGD, Analog SGD, Residual Learning/Tiki-Taka, and Residual Learning v2, respectively.

Intuitively, with a fixed $P_{k+1}$, $H_k$ will converge to a neighborhood of $P_{k+1}$ with radius $O(\beta)$. Therefore, a sufficiently small $\beta$ renders $H_k$ a fair approximation of noiseless $P_k$, enabling optimizing the upper-level objective with clearer signals. After that, $H_{k+1}$ is transferred to $W_k$ as follows

$$W_{k+1} = W_k + \beta H_{k+1} \odot F(W_k) - \beta|H_{k+1}| \odot G(W_k). \tag{15}$$

Furthermore, the transfer process suffers from a constant error of $O(\Delta w_{\min})$ due to the discrete pulse firing, each of which changes the weight by $O(\Delta w_{\min})$. To overcome these issues, we propose introducing a threshold mechanism that does not transfer the entire $H_{k+1}$ to $W_k$ at each iteration, as in (15). Instead, we compute an intermediate value by $H_{k+\frac{1}{2}} = (1-\beta)H_k + \beta(P_{k+1} + \varepsilon_{k+1})$ first. At each coordinate $d$, if the value $|[H_{k+\frac{1}{2}}]_d| \geq \Delta w_{\min}$, one pulse will be fired to $[W_k]_d$ and update the digital buffer by $[H_{k+1}]_d = [H_{k+\frac{1}{2}}]_d - \Delta w_{\min}$ or $[H_{k+1}]_d = [H_{k+\frac{1}{2}}]_d + \Delta w_{\min}$, where the sign of increment is determined by the sign of $[H_{k+\frac{1}{2}}]_d$. Otherwise, no transfer is triggered if the intermediate value falls below the threshold, i.e., $[H_{k+1}]_d = [H_{k+\frac{1}{2}}]_d$. The proposed algorithms are referred to as Residual Learning v2.

## 6 Numerical Simulations

In this section, we verify the main theoretical results by simulations on both synthetic datasets and real datasets. We use the open source toolkit IBM Analog Hardware Acceleration Kit (AIHWKIT) [44] to simulate the behaviors of Analog SGD, Residual Learning (which reduces to Tiki-Taka). Each simulation is repeated three times, and the mean and standard deviation are reported. We consider two types of response functions in our simulations: power and exponential response functions with dynamic ranges $[-\tau, \tau]$ and the symmetric point being 0, as required by Corollary 1. More details, simulations, and ablation studies can be found in Appendix K. The code of our simulations is available at github.com/Zhaoxian-Wu/analog-training.

**FCN/CNN @ MNIST.** We train a fully-connected network (FCN) and a convolutional neural network (CNN) on the MNIST dataset and compare the performance of Analog SGD and Tiki-Taka under various dynamic range $\tau$ on power responses; see the results in Figure 4. By tracking residual, Residual Learning outperforms Analog SGD and reaches comparable accuracy with Digital SGD. For both architectures, the accuracy of Residual Learning drops by $< 1\%$. In contrast, Analog SGD takes a few epochs to achieve a noticeable increase in accuracy in FCN training, rendering a slower convergence rate than Residual Learning. In CNN training, Analog SGD's accuracy increases more slowly than Residual Learning, eventually settling at about 80%. It is consistent with the theoretical claims.

**ResNet @ CIFAR10/CIFAR100.** We fine-tune three ResNet models with different scales on CIFAR10/CIFAR100 datasets. The power response functions are used, whose results are shown in Table 1. The results show that the Tiki-Taka outperforms Analog SGD by about $1.0\%$ in most of the cases in ResNet34/50, and the gap even reaches about $7.0\%$ for ResNet18 training on the CIFAR100 dataset. On top of that, we also compare the proposed Residual Learning v2 and Tiki-Taka v2.

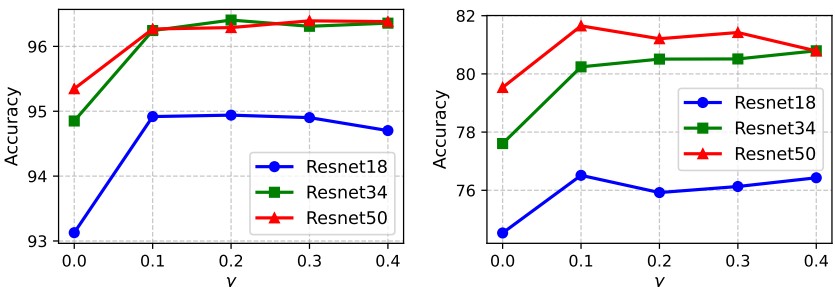

Figure 5: The test accuracy of ResNet family models after 100 epochs trained by `Residual Learning` under different $\gamma$ in (10); **(Left)** CIFAR10. **(Right)** CIFAR100.

Both of them outperform `Residual Learning` since they introduce a digital buffer to filter out the reading noise. However, `Residual Learning v2` outperforms `Tiki-Taka v2` on the CIFAR100 dataset, demonstrating the benefit from the bilevel formulation.

**Ablation study on $\gamma$.** We conduct simulations to study the impact of mixing coefficient $\gamma$ in (10) on the CIFAR10 or CIFAR100 dataset in the ResNet training tasks. The results are presented in Figure 5, which shows that `Residual Learning` achieves a great accuracy gain from increasing $\gamma$ from 0 to 0.1, while the gain saturates from 0.1 to 0.4. Therefore, we conclude that `Residual Learning` benefits from a non-zero $\gamma$, and the performance is robust to the $\gamma$ selection.

## 7 Conclusions and Limitations

This paper studies the impact of a generic class of asymmetric and non-linear response functions on gradient-based training in analog in-memory computing hardware. We first formulate the dynamics of `Analog Update` based on the pulse update rule. Based on it, we show that `Analog SGD` implicitly optimizes a penalized objective and hence can only converge inexactly. To overcome this issue, we propose a `Residual Learning` framework which solves a bilevel optimization problem. Explicitly aligning the algorithmic stationary point and physical symmetric point, `Residual Learning` provably converges to the optimal point exactly. Furthermore, we demonstrate how to extend `Residual Learning` to overcome the noisy reading and limited update granularity issues. The efficiency of the proposed method is verified through simulations. One limitation of this work is that the current analysis considers only the three hardware imperfections. While they are known to be crucial for analog training, it is also important to extend our convergence analysis and methods to more practical scenarios involving more imperfections in future work.

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

# Appendix for "Analog In-memory Training on Non-ideal Resistive Elements: Understanding the Impact of Response Functions"

## Table of Contents

## A  Literature Review

This section briefly reviews literature that is related to this paper, as complementary to Section 1.

**Training on AIMC hardware.** Analog training has shown promising early successes in tasks such as face classification [25] and digit classification [26], achieving $1,000\times$ lower energy consumption than digital implementations. Researchers are also exploring approaches to mitigate the impact of hardware non-idealities. For example, [27, 28] proposes leveraging the momentum technique to stabilize training by reducing noise. To address other potential non-idealities, a hybrid training paradigm is also being explored. [29] leverages the chip-in-the-loop technique to train models layer-by-layer, while [30] proposes to train the backbone in the digital domain and train the last

layer in the analog domain. In general, these works have provided valuable insights into analog training, shedding light on many critical technical challenges. However, their focus has largely been on experimental and simulation aspects, with limited systematic and theoretical analysis of how specific imperfections affect the training process. In our paper, we present an alternative viewpoint and novel tools to explore the effects of non-idealities.

**Resistive element.** A series of works seeks various resistive elements that have near-constant or at least symmetric responses. The leading candidates currently include PCM [45, 46], ReRAM [47–49], CBRAM [50, 51], ECRAM [52, 53], MRAM [54, 55], FTJ [56] or flash memory [57–59].

However, a resistive element with symmetric updates may not be the best option for manufacturing. For example, although ECRAM provides almost symmetric updates, it remains less competitive than ReRAM, which offers faster response speed and lower pulse voltage [49]. The suitability of the resistive elements is evaluated using metrics across multiple dimensions, including the number of conductance states, retention, material endurance, switching energy, response speed, manufacturing cost, and cell size. Among them, this paper is only interested in the impact of response functions in the training.

**Imperfection of AIMC hardware.** Besides the response functions, analog training suffers from all kinds of hardware imperfection, especially when the task's scale increases, like asymmetric update [17, 19], reading/writing noise [18, 60, 61], device/cycle variations [62], non-linear current response due to IR drop [18, 6, 63]. This paper mainly focuses on asymmetric response functions. However, this paper is not trying to diminish the importance of other hardware imperfections but rather focuses on one of the primary ones known to be very important [19, 16].

**Hardware-aware training.** For inference on AIMC hardware purposes, models pretrained on digital hardware will be programmed on analog hardware. Due to hardware imperfections, the pretrained models suffer performance drops. Hardware-aware training (HWA) is a technique designed to bridge the gap between ideal pretrained models and non-ideal programmed models. In contrast to standard training methods, hardware-aware training explicitly incorporates device-specific imperfections, such as weight drift [20], device fail [64], bounded dynamic range [65], quantization error from ADC [66–68], device variation [69], and non-linear current output [70], into the training loop. By modeling these constraints during training, the learned parameters become inherently more robust to real-world deployment conditions. It is worth highlighting that HWA is still performed on digital hardware, and the trained model will be programmed onto AIMC hardware. On the contrary, this paper considers a different, more challenging setting in which training is performed directly on analog hardware.

**Gradient-based training on AIMC hardware.** A series of works focuses on implementing back-propagation (BP) and gradient-based training on AIMC hardware. The seminal work [16, 71] leverages the rank-one structure of the gradient and implements `Analog SGD` by a stochastic pulse update scheme, *rank-update*. Rank-update significantly accelerates the gradient descent step by avoiding the $O(N^2)$-element computation of gradients and instead using two vectors with $O(N)$ elements for the update, where $N$ is the number of matrix rows and columns. To alleviate the *asymmetric update issue*, researchers also design various of `Analog SGD` variants, `Tiki-Taka` algorithm family [22–24]. The key components of `Tiki-Taka` are the introduction of a *residual array* to stabilize training. Apart from the rank-update, a hybrid scheme that performs forward and backward passes in the analog domain but computes gradients in the digital domain has been proposed in [31, 32]. Their solution, referred to as *mixed-precision update*, provides a more accurate gradient signal but requires 5×-10× higher overhead compared to the rank-update scheme [24].

Attributed to these efforts, analog training has empirically shown great promise, achieving accuracy comparable to that of digital training on chip prototypes while reducing energy consumption and training time [72, 73]. Simultaneously, the parallel acceleration solution with AIMC hardware is under exploration [74]. Despite its good performance, it remains mysterious when and why the analog training works.

**Theoretical foundation of gradient-based training.** The closely related result comes from the convergence study of `Tiki-Taka` [21]. Similar to our work, they attempt to model the dynamics and provide the convergence properties of `Analog SGD` and `Tiki-Taka`. However, their work is limited to a special linear response function. Furthermore, their paper considers a simplified version of `Tiki-Taka`, with a hyper-parameter $\gamma = 0$ (see Section 4). As we will show empirically and

theoretically, `Tiki-Taka` benefits from a non-zero $\gamma$. Consequently, We compare the results briefly in Table 2 and comprehensively in Appendix B.

| | $\gamma$ | Generic response | Linear response |
|---|---|---|---|
| `Tiki-Taka` [21] | $= 0$ | ✗ | $O\left(\sqrt{\frac{1}{K}}\frac{1}{1-33P_{\max}^2/\tau^2}\right)$ |
| `Tiki-Taka` [Corollary 1] | $\neq 0$ | $O\left(\sqrt{\frac{1}{K}}\frac{1}{M_{\min}^{\mathrm{RL}}}\right)$ | $O\left(\sqrt{\frac{1}{K}}\frac{1}{1-P_{\max}^2/\tau^2}\right)$ |

Table 2: Comparison between our paper and [21]. Mixing-coefficient $\gamma$ is a hyper-parameter of `Tiki-Taka`. "Generic response" and "Linear response" columns are the convergence rates in the corresponding settings. $K$ represents the number of iterations. $M_{\min}^{\mathrm{RL}}$ and $P_{\max}^2/\tau^2 < 1$ measure the saturation while the former one reduces to the latter on linear response functions.

**Energy-based models and equilibrium propagation.** Apart from achieving explicit gradient signals by the BP, there are also attempts to train models based on *equilibrium propagation* (EP, [33]), which provides a biologically plausible alternative to traditional BP. EP is applicable to a series of energy-based models, where the forward pass is performed by minimizing an energy function [34, 35]. The update signal in EP is computed by measuring the output difference between a free phase and an active phase. EP eliminates the need for BP non-local weight transport mechanism, making it more compatible with neuromorphic and energy-efficient hardware [36, 37]. We highlight here that the approach to attain update signals (BP or EP) is orthogonal to the update mechanism (pulse update). Their difference lies in the objective $f(W_k)$, which is hidden in this paper. Therefore, building upon the pulse update, our work is applicable to both BP and EP.

**Physical neural network.** The model executing on AIMC hardware, which leverages resistive crossbar array to accelerate MVM operation, is a concrete implementation of physical neural networks (PNNs, [75, 76]). PNN is a generic concept of implementing neural networks via a physical system in which a set of tunable parameters, such as holographic grating [77], wave-based systems [78], and photonic networks [79]. Our work particularly focuses on training with AIMC hardware, but the methodology developed in this paper can be transferred to the study of other PNNs.

# B   Relation with the result in [21]

Similar to this paper, [21] also attempts to model the dynamics of analog training. They show that `Analog SGD` converges to a critical point of problem (1) inexactly with an asymptotic error, and `Tiki-Taka` converges to a critical point exactly. In this section, we compare our results with our results and theirs.

As discussed in Section 1, [21] studies the analog training on special linear response functions

$$q_+(w) = 1 - \frac{w}{\tau}, \quad q_-(w) = 1 + \frac{w}{\tau}. \tag{16}$$

It can be checked that the symmetric point is 0 while the dynamic range of it is $[-\tau, \tau]$. The symmetric and asymmetric components are defined by $F(W) = 1$ and $G(W) = \frac{W}{\tau}$, respectively. It indicates $F_{\max} = 1$. Furthermore, they assume the bounded weight saturation by assuming bounded weights, i.e., $\|W_k\|_\infty \leq W_{\max}, \forall k \in [K]$ with a constant $W_{\max} < \tau$. Under this assumption, the lower bounds of response functions are given by

$$q_{\max} = 1 + \frac{W_{\max}}{\tau}, \quad q_{\min} = 1 - \frac{W_{\max}}{\tau}, \tag{17}$$

$$\min\{M(W_k)\} = \min\{Q_+(W_k) \odot Q_-(W_k)\} = 1 - \left(\frac{\|W_k\|_\infty}{\tau}\right)^2 \tag{18}$$

$$M_{\min}^{\mathrm{ASGD}} = \min\{M(W_k)\} = 1 - \left(\frac{W_{\max}}{\tau}\right)^2. \tag{19}$$

**Challenge of analyzing the convergence of `Tiki-Taka` with generic response functions.** For linear response functions (16), the recursion of residual array $P_k$ has a special structure, where the

first and the biased term can be combined

$$P_{k+1} = P_k - \alpha \nabla f(\bar{W}_k; \xi_k) - \frac{\alpha}{\tau} |\nabla f(\bar{W}_k; \xi_k)| \odot P_k \tag{20}$$
$$= \left(1 - \frac{\alpha}{\tau} |\nabla f(\bar{W}_k; \xi_k)|\right) \odot P_k - \alpha \nabla f(\bar{W}_k; \xi_k)$$

which is a weighted average of $P_k$ and $\nabla f(\bar{W}_k; \xi_k)$. Consequently, $P_k$ can be interpreted as an approximation of the average gradient. From this perspective, the transfer operation can be interpreted as biased gradient descent. However, given a generic $G(\cdot)$, the combination is no longer viable, bringing difficulties to the analysis.

**Convergence of `Analog SGD`.** As we will show in Remark 1 at the end of Appendix H, inequality (8) can be improved when the saturation never happens

$$\frac{1}{K} \sum_{k=0}^{K-1} \mathbb{E}[\|\nabla f(W_k)\|^2] \tag{21}$$

$$\leq \frac{4F_{\max}^2}{M_{\min}^{\mathrm{RL}}} \sqrt{\frac{(f(W_0) - f^*)\sigma^2 L}{K}} + 2F_{\max}\sigma^2 \times \frac{1}{K} \sum_{k=0}^{K-1} \left\| \frac{G(W_k)}{\sqrt{F(W_k)}} \right\|_\infty^2 \bigg/ \min\{M(W_k)\}$$

$$\leq O\left( \sqrt{\frac{(f(W_0) - f^*)\sigma^2 L}{K}} \frac{1}{1 - W_{\max}^2/\tau^2} \right) + 2\sigma^2 \times \frac{1}{K} \sum_{k=0}^{K} \frac{\|W_k\|_\infty^2/\tau^2}{1 - \|W_k\|_\infty^2/\tau^2}$$

which is exactly the result in [21].

**Convergence of `Tiki-Taka`.** It is shown empirically that a non-zero $\gamma$ in (10) improves the training accuracy [22]. However, [21] only considers $\gamma = 0$ while this paper considers a non-zero $\gamma$.

With the linear response, if we also assume the bounded saturation of $P_k$ by letting $\|P_k\|_\infty \leq P_{\max}$, the minimal average response function is given by $M_{\min}^{\mathrm{RL}} = 1 - \left(\frac{P_{\max}}{\tau}\right)^2$. The upper bound in Corollary 1 becomes

$$\frac{1}{K} \sum_{k=0}^{K-1} \|\nabla f(\bar{W}_k)\|^2 \leq O\left( \frac{1}{1 - P_{\max}^2/\tau^2} \sqrt{\frac{(f(W_0) - f^*)\sigma^2 L}{K}} \right). \tag{22}$$

As a comparison, without a non-zero $\gamma$, [21] shows that convergence rate of `Tiki-Taka` is only

$$\frac{1}{K} \sum_{k=0}^{K-1} \|\nabla f(W_k)\|^2 \leq O\left( \frac{1}{1 - 33P_{\max}^2/\tau^2} \sqrt{\frac{(f(W_0) - f^*)\sigma^2 L}{K}} \right). \tag{23}$$

Even though it is not a completely fair comparison, since the two papers rely on different assumptions, it is still worth comparing their analyses. [21] assumes the noise should be non-zero, i.e. $[\mathbb{E}_\xi[|\nabla f(W; \xi)|]]_d \geq c_{\mathtt{noise}}\sigma, \forall d \in [D]$ holds for a non-zero constant $c_{\mathtt{noise}}$. Instead, this paper does not make this assumption but assumes that the objective is strongly convex. As mentioned in Section 4, the strong convexity is introduced only to ensure the existence of $P^*(W_k)$. Therefore, we believe it can be relaxed and that the convergence rate can remain unchanged, which is left for future work. Taking that into account, we believe the comparison can provide insight into how the non-zero $\gamma$ improves the convergence rate of `Tiki-Taka`.

**Why does non-zero $\gamma$ improve the convergence rate of `Tiki-Taka`?** As discussed in Section 4, $P_k$ is interpreted as a residual array that optimizes $f(W_k + \gamma P)$. In the ideal setting that $F(W) = 1$ and $G(W) = 0$, it can be shown that $P_k$ converges to $P^*(W_k)$ if $W_k$ is fixed and $P_k$ is kept updated, even though the $W_k \neq W^*$ (hence $\nabla f(W_k) \neq 0$).

Instead, without a non-zero $\gamma$, [21] interprets $P_k$ as an approximation of clear gradient by showing

$$\mathbb{E}_{\xi_k}[\|P_{k+1} - C\nabla f(W_k)\|^2] \tag{24}$$

$$\leq \left(1 - \frac{\beta}{C}\right) \|P_k - C\nabla f(W_k)\|^2 + O(\beta C')\|\nabla f(W_k)\|^2 + \text{remainder}$$

where $C, C'$ are constants depending on the resistive element and model dimension, and the "remainder" is the non-essential terms. Consider the case that $W_k$ is fixed and (10) is kept iterating, in which

case the increment on $P_k$ is constant since $\gamma = 0$. Telescoping (24), we find that the upper bound above only guarantees that

$$\limsup_{k \to \infty} \mathbb{E}[\|P_{k+1} - C\nabla f(W_k)\|^2] \leq O(CC'\|\nabla f(W_k)\|^2) \tag{25}$$

which means that $P_k$ tracks the gradient accurately only when $\nabla f(W_k)$ reaches zero asymptotically. The less accurate approximation results in a slower rate than the one reported in this paper.

## C  Dynamics of Non-ideal Analog Update

This section presents details on how to obtain the dynamics of the analog update (3) appearing in Section 2, along with its error analysis. The primary distinction between digital and analog training is the method of updating the weight. As discussed in Section 1, the weight update in AIMC hardware is implemented by `Analog Update`, which sends a series of pulses to the resistive elements.

**Pulse update.** Consider the response of one resistive element in one cycle, which involves only one pulse. Given the initial weight $w$, the updated weight increases or decreases by about $\Delta w_{\min}$ depending on the pulse polarity, where $\Delta w_{\min} > 0$ is the *response granularity* determined by elements. The granularity is further scaled by a factor, which varies by the update direction due to the *asymmetric update* property of resistive elements. The notations $q_+(\cdot)$ and $q_-(\cdot)$ are used to denote the *response functions* on positive or negative sides, respectively, to describe the dominating part of the factor. In practice, the analog noise also causes a deviation of the effective factor from the response functions, referred to as *cycle variation*. It is represented by the magnitude $\sigma_c$ times a random variable $\xi_c$ with expectation 0 and variance 1. Taking all of them into account, with $s \in \{+, -\}$ being the update direction, the updated weight after receiving one pulse is $\tilde{U}_q(w, s)$ where $\tilde{U}_q(\cdot, \cdot) : \mathbb{R} \times \{+, -\} \to \mathbb{R}$ is the element-dependent update that implements the resistive element, which can be expressed as

$$\tilde{U}_q(w, s) := w + \Delta w_{\min} \cdot (q_s(w) + \sigma_c \xi) \tag{26}$$
$$= \begin{cases} w + \Delta w_{\min} \cdot (q_+(w) + \sigma_c \xi_c), & s = +, \\ w - \Delta w_{\min} \cdot (q_-(w) + \sigma_c \xi_c), & s = -. \end{cases}$$

The typical signal and noise ratio $\sigma_c/q_s(w)$ is roughly 5%-100% [80, 49], varied by the type of resistive elements. Furthermore, the response functions also vary by elements due to the imperfection in fabrication, called *element variation* (also referred to as *device variation* in literature [16]).

Equation (26) is a resistive element level equation. Existing work exploring the candidates of resistive elements usually reports the response curves similar to Figure 1, [73, 52, 49]. Taking the difference between weights in two consecutive pulse cycles and adopting statistical approaches [80], all the element-dependent quantities, including $\Delta w_{\min}$, $q_+(\cdot)$, $q_-(\cdot)$ and $\sigma_c$, can be estimated from the response curves of the resistive elements.

**Analog update implemented by pulse updates.** Even though the update scheme has evolved over the years [16, 71], we discuss a simplified version, called `Analog Update`, to retain the essential properties. To update the weight $w$ by $\Delta w$, a series of pulses are sent, whose *bit length (BL)* is computed by $\mathrm{BL} := \left\lceil \frac{|\Delta w|}{\Delta w_{\min}} \right\rceil$. After received BL pulses, the updated weight $w'$ can be expressed as the function composition of (26) by BL times

$$w' = \underbrace{\tilde{U}_q \circ \tilde{U}_q \circ \cdots \circ \tilde{U}_q}_{\times \text{ BL}}(w, s) =: \tilde{U}_q^{\mathrm{BL}}(w, s). \tag{27}$$

Roughly speaking, given an ideal response $q_+(w) = q_-(w) = 1$ and $\sigma_c = 0$, BL pulses, with $\Delta w_{\min}$ increment for each individual pulse, incur the weight update $\Delta w$. Since the response granularity $\Delta w_{\min}$ is scaled by the response function $q_s(w)$, the expected increment is approximately scaled by $q_s(w)$ as well. Accordingly, we propose an approximate dynamics of `Analog Update` is given by $w' \approx U_q(w, \Delta w)$, where $U_q(w, \Delta w)$ is defined in (3). The following theorem provides an estimation of the approximation error. It has been shown empirically that the response granularity can be made sufficiently small for updating [81, 82], implying $\Delta w_{\min} \ll \Delta w$. Therefore, we establish the error estimate for the approximation under a small-response-granularity condition.

**Theorem 4** (Error from discrete pulse update). *Suppose the response granularity is sufficiently small such that $\Delta w_{\min} \leq o(\Delta w)$. With the update direction $s = \text{sign}(\Delta w)$, the error between the true update $\tilde{U}_q^{\text{BL}}(w, s)$ and the approximated $U_q(w, \Delta w)$ is bounded by*

$$\lim_{\Delta w \to 0} \frac{|\tilde{U}_q^{\text{BL}}(w, s) - U_q(w, \Delta w)|}{|\tilde{U}_q^{\text{BL}}(w, s) - w|} = 0. \tag{28}$$

In Theorem 4, $|\tilde{U}_q^{\text{BL}}(w, s) - U_q(w, \Delta w)|$ is the error between the true update and the proposed dynamics, while $|\tilde{U}_q^{\text{BL}}(w, s) - w|$ is the difference between original weight and the updated one. Theorem 4 shows that the proposed dynamics dominate the update, and the approximation error is negligible when $\Delta w$ is small, which holds as $\Delta w$ always includes a small learning rate in gradient-based training.

> **Takeaway.** Theorem 4 enables us to discuss the impact of response functions directly without dealing with element-specific details like response granularity $\Delta w_{\min}$ and cycle variation $\sigma_c$. Response functions are the bridge between the resistive element level equation (pulse update (26)) and the algorithm level equation (dynamics of `Analog Update` (3)).

*Proof of Theorem 4.* Recall the definition of the bit length is

$$\text{BL} := \left\lceil \frac{|\Delta w|}{\Delta w_{\min}} \right\rceil = \Theta\left( \frac{|\Delta w|}{\Delta w_{\min}} \right) \tag{29}$$

leading to

$$|\text{BL}\,\Delta w_{\min} - |\Delta w|| \leq \Delta w_{\min} \quad \text{or} \quad |s\,\text{BL}\,\Delta w_{\min} - \Delta w| \leq \Delta w_{\min}. \tag{30}$$

Notice that the update responding to each pulse is a $\Theta(\Delta w_{\min})$ term. Directly manipulating $U_p^{\text{BL}}(w, s)$ and expanding it in Taylor series to the first-order term yields

$$
\begin{aligned}
U_p^{\text{BL}}(w, s) &= w + s \cdot \Delta w_{\min} \sum_{t=0}^{\text{BL}-1} q_s(w + \Theta(t \Delta w_{\min})) + \Delta w_{\min} \sum_{t=0}^{\text{BL}-1} \sigma_c \xi_t \tag{31} \\
&= w + s \cdot \Delta w_{\min} \sum_{t=0}^{\text{BL}-1} q_s(w) + \sum_{t=0}^{\text{BL}-1} \Theta(t(\Delta w_{\min})^2) + \Delta w_{\min} \sum_{t=0}^{\text{BL}-1} \sigma_c \xi_t \\
&= w + s \cdot \Delta w_{\min} \cdot \text{BL} \cdot q_s(w) + \Theta(\text{BL}^2 (\Delta w_{\min})^2) + \Delta w_{\min} \cdot \sqrt{\text{BL}} \cdot \sigma_c \xi \\
&= w + \Delta w \cdot q_s(w) + (s\,\text{BL}\,\Delta w_{\min} - \Delta w) + \Theta((\Delta w)^2) + \sqrt{\Delta w_{\min}} \cdot \sqrt{\Delta w} \cdot \sigma_c \xi \\
&= U_q(w, \Delta w) + \Theta(\Delta w_{\min}) + \Theta((\Delta w)^2) + \Theta(\sqrt{\Delta w_{\min}} \cdot \sqrt{\Delta w} \cdot \sigma_c)
\end{aligned}
$$

where $\xi := \frac{1}{\sqrt{\text{BL}}} \sum_{t=0}^{\text{BL}-1} \xi_t$ is the accumulated noise with variance 1. The proof is completed. $\square$

## D  Comparison of Residual Learning v2 and Tiki-Taka v2

Introducing a digital buffer, the proposed `Residual Learning v2` has a similar form of `Tiki-Taka v2` [23]. However, there are slight differences. `Tiki-Taka v2` updates the digital buffer by

$$H_{k+\frac{1}{2}} = H_k + \beta(P_{k+1} + \varepsilon_k) \tag{32}$$

which do not include a decay coefficient in front of $H_k$. Furthermore, `Tiki-Taka v2` uses the gradient $\nabla f(W_k; \xi_k)$ that are solely computed on the main array $W_k$. Instead, `Residual Learning v2` computes gradient on a mixed weight $\bar{W}_k = W_k + \gamma P_k$. As suggested by the ablation simulations in 6, the training benefits from a non-zero $\gamma$.

# E Estimation of time consumption

`Residual Learning` introduces an extra resistive element array, which increases overhead. However, the extra overhead is affordable in practice. Compared to `Analog SGD`, the analog memory requirement doubles, but the latency remains almost unchanged since Residual Learning does not explicitly compute the mixed weights during the forward and backward passes. As [83] suggests, $W_k$ and $P_k$ can share the same analog-digital convertor (ADC), which implements the weight mixing without introducing extra latency. On the other hand, as suggested by [22], the forward, backward, and update steps for $W_k$ and $P_k$ are performed in parallel, thereby avoiding a significant increase in latency. Consequently, introducing an extra residual array does not incur substantial extra latency.

Following the evaluation in Table 1 in [24], we compared the latency of `Analog SGD` and `Residual Learning` in Table 3. We consider that each gradient update step requires 32 pulse cycles, each consuming 5 nanoseconds (ns). Following the estimation in [24], preprocessing the input vectors for each MVM operator takes 5.9ns. Compared with `Analog SGD`, `Residual Learning` adds an extra MVM step to read from $P_k$. The results suggest that the overhead is only about $2\times$ that of `Analog SGD`. As the update is typically not the bottleneck of the whole training process, the extra overhead is affordable.

|                      | `Analog SGD` | `Residual Learning` |
| -------------------- | ------------ | ------------------- |
| Forward/backward [84] | 40.0         | 40.0                |
| Update               | 165.9        | 371.8               |

Table 3: Comparison of time (nanosecond) consumption in each layer

# F Useful Lemmas and Proofs

## F.1 Lemma 1: Properties of weighted norm

**Lemma 1.** $\|W\|_S$ *has the following properties:* (a) $\|W\|_S = \|W \odot \sqrt{S}\|$; (b) $\|W\|_S \leq \|W\|\sqrt{\|S\|_\infty}$; (c) $\|W\|_S \geq \|W\|\sqrt{\min\{S\}}$.

*Proof of Lemma 1.* The lemma can be proven easily by definition. $\qquad\square$

## F.2 Lemma 2: Properties of weighted norm

A direct property from Definition 1 is that all $q_+(\cdot)$, $q_-(\cdot)$, and $F(\cdot)$ are bounded, as guaranteed by the following lemma.

**Lemma 2.** *The following statements are valid for all $W \in \mathcal{R}$.* (a) $F(\cdot)$ *is element-wise upper bounded by a constant $F_{\max} > 0$, i.e., $\|F(W)\|_\infty \leq F_{\max}$;* (b) $Q_+(\cdot)$ *and $\nabla Q_-(\cdot)$ are element-wise bounded by $L_Q$, i.e., $\|\nabla Q_+(W)\|_\infty \leq L_Q$, $\|\nabla Q_-(W)\|_\infty \leq L_Q$.*

## F.3 Lemma 3: Lipschitz continuity of analog update

**Lemma 3.** *The increment defined in* (5) *is Lipschitz continuous with respect to $\Delta W$ under any weighted norm $\|\cdot\|_S$, i.e., for any $W, \Delta W, \Delta W' \in \mathbb{R}^D$ and $S \in \mathbb{R}^D_+$, it holds*

$$\|\Delta W \odot F(W) - |\Delta W| \odot G(W) - (\Delta W' \odot F(W) - |\Delta W'| \odot G(W))\|_S \qquad (33)$$
$$\leq F_{\max}\|\Delta W - \Delta W'\|_S.$$

*Proof of Lemma 3.* We prove for the case where $D = 1$ and $S = 1$, and the general case can be proven similarly. Notice that the absolute value $|\cdot|$ and vector norm $\|\cdot\|$, scalar multiplication $\times$ and element-wise multiplication $\odot$, are equivalent at that situation. We adopt both notations just for readability.

$$\|\Delta W \odot F(W) - |\Delta W| \odot G(W) - (\Delta W' \odot F(W) - |\Delta W'| \odot G(W))\| \qquad (34)$$

$$= \|(\Delta W - \Delta W') \odot F(W) - (|\Delta W| - |\Delta W'|) \odot G(W)\|.$$

Since $\|\Delta W - \Delta W'\| \geq \||\Delta W| - |\Delta W'|\|$ and $|G(W)| \leq |F(W)|$, we have

$$|(\Delta W - \Delta W') \odot F(W) - (|\Delta W| - |\Delta W'|) \odot G(W)| \tag{35}$$
$$\leq |(\Delta W - \Delta W') \odot (F(W) - |G(W)|)|$$
$$\leq |\Delta W - \Delta W'| \, |F(W) - |G(W)||$$
$$\leq F_{\max}|\Delta W - \Delta W'|$$

which completes the proof. $\qquad\square$

### F.4  Lemma 4: Element-wise product error

**Lemma 4.** *Let $U, V, Q \in \mathbb{R}^D$ be vectors indexed by $[D]$. Then the following inequality holds*

$$\langle U, V \odot Q \rangle \geq C_+ \langle U, V \rangle - C_- \langle |U|, |V| \rangle \tag{36}$$

*where the constant $C_+$ and $C_-$ are defined by*

$$C_+ := \frac{1}{2}\left(\max\{Q\} + \min\{Q\}\right), \quad C_- := \frac{1}{2}\left(\max\{Q\} - \min\{Q\}\right). \tag{37}$$

*Proof of Lemma 4.* For any vectors $U, V, Q \in \mathbb{R}^D$, it is always valid that

$$\langle U, V \odot Q \rangle = \sum_{d \in [D]} [U]_d [V]_d [Q]_d \tag{38}$$

$$= \sum_{d \in [D], [U]_d [V]_d \geq 0} [U]_d [V]_d [Q]_d \; + \sum_{d \in [D], [U]_d [V]_d < 0} [U]_d [V]_d [Q]_d$$

$$\geq \min\{Q\} \times \left( \sum_{d \in [D], [U]_d [V]_d \geq 0} [U]_d [V]_d \right) + \max\{Q\} \times \left( \sum_{d \in [D], [U]_d [V]_d < 0} [U]_d [V]_d \right)$$

$$\stackrel{(a)}{=} C_+ \left( \sum_{d \in [D], [U]_d [V]_d \geq 0} [U]_d [V]_d \right) - C_- \left( \sum_{d \in [D], [U]_d [V]_d \geq 0} |[U]_d [V]_d| \right)$$

$$+ C_+ \left( \sum_{d \in [D], [U]_d [V]_d < 0} [U]_d [V]_d \right) - C_- \left( \sum_{d \in [D], [U]_d [V]_d < 0} |[U]_d [V]_d| \right)$$

$$= C_+ \sum_{d \in [D]} [U]_d [V]_d - C_- \sum_{d \in [D]} |[U]_d [V]_d|$$

$$= C_+ \langle U, V \rangle - C_- \langle |U|, |V| \rangle$$

where $(a)$ uses the following equality

$$\min\{Q\}[U]_d [V]_d = C_+ [U]_d [V]_d - C_- |[U]_d [V]_d|, \quad \text{if } [U]_d [V]_d \geq 0, \tag{39}$$
$$\max\{Q\}[U]_d [V]_d = C_+ [U]_d [V]_d - C_- |[U]_d [V]_d|, \quad \text{if } [U]_d [V]_d < 0. \tag{40}$$

This completes the proof. $\qquad\square$

## G  Proof of Theorem 1: Implicit Bias of Analog Training

In this section, we provide a full version of Theorem 1. Before that, we introduce the *asymmetry ratio* and its element-wise anti-derivative. Since $F(\cdot)$ and $G(\cdot)$ act element-wise, we define $R(\cdot) : \mathbb{R}^D \to \mathbb{R}^D$ and $R_c(\cdot) : \mathbb{R}^D \to \mathbb{R}^D$ element-wise by

$$R(W) := \frac{G(W)}{F(W)} \quad \text{and} \quad [R_c(W)]_d := \int_{[W^\circ]_d}^{[W]_d} [R(W')]_d \, \mathrm{d}[W']_d, \quad d \in [D], \tag{41}$$

where the division is taken element-wise. It holds that $\frac{\mathrm{d}}{\mathrm{d}[W]_d}[R_c(W)]_d = [R(W)]_d$ for each coordinate $d \in [D]$, and $\nabla \langle C, R_c(W) \rangle = C \odot R(W)$ for any vector $C \in \mathbb{R}^D$. Consequently, if $R(W)$ is strictly monotonic, $R_c(W) \geq 0$, and it reaches its minimum value at the symmetric point $W^\diamond$ where $R(W^\diamond) = 0$.

Define the *scaled effective update* of `Analog SGD` at $W$ as

$$T(W) := \mathbb{E}_\xi[\nabla f(W; \xi)] + \mathbb{E}_\xi[|\nabla f(W; \xi)|] \odot \frac{G(W)}{F(W)}. \tag{42}$$

A direct verification shows that if $T(W_k) = 0$, then `Analog SGD` (2) (equivalently, (5) with $\Delta W_k = -\alpha \nabla f(W_k; \xi_k)$) is stable at $W_k$ in conditional expectation, i.e., $\mathbb{E}_{\xi_k}[W_{k+1}] = W_k$.

Rather than characterizing the exact limit of `Analog SGD`, we establish a local statement showing the existence of a point that is simultaneously (i) nearly critical for $f_\Sigma$; and, (ii) nearly stationary for the mean `Analog SGD` dynamics.

**Theorem 5** (Implicit Penalty, full version of Theorem 1). *Suppose $W^*$ is the unique minimizer of problem* (1). *Define*

$$\tilde{W}^* := \left( \nabla^2 f(W^*) + \mathrm{Diag}(\Sigma)\, \nabla R(W^\diamond) \right)^{-1} \left( \nabla^2 f(W^*)\, W^* + \mathrm{Diag}(\Sigma)\, \nabla R(W^\diamond) W^\diamond \right) \tag{43}$$

*where* $\mathrm{Diag}(M) \in \mathbb{R}^{D \times D}$ *denotes the diagonal matrix whose diagonal entries are given by the vector* $M \in \mathbb{R}^D$. *Let* $\Sigma := \mathbb{E}_\xi[|\nabla f(W^*; \xi)|] \in \mathbb{R}^D$ *be the element-wise first moment of stochastic gradients at* $W^*$. `Analog SGD` *implicitly optimizes*

$$\min_{W \in \mathbb{R}^D}\ f_\Sigma(W) := f(W) + \langle \Sigma, R_c(W) \rangle \tag{44}$$

*in the sense that as* $\|W^\diamond - W^*\| \to 0$,

$$\frac{\|\nabla f_\Sigma(\tilde{W}^*)\|}{\min\{\|\nabla f_\Sigma(W^*)\|, \|\nabla f_\Sigma(W^\diamond)\|\}} \to 0 \quad \text{and} \quad \frac{\|T(\tilde{W}^*)\|}{\min\{\|T(W^*)\|, \|T(W^\diamond)\|\}} \to 0. \tag{45}$$

*Proof of Theorem 5.* By the definition of $\tilde{W}^*$, it holds that

$$\begin{aligned}
\|W^\diamond - \tilde{W}^*\| &= \|(\nabla^2 f(W^*) + \mathrm{Diag}(\Sigma)\, \nabla R(W^\diamond))^{-1} \nabla^2 f(W^*)\, (W^* - W^\diamond)\| \\
&= \Theta(\|W^\diamond - W^*\|) \tag{46} \\
\|W^* - \tilde{W}^*\| &= \|(\nabla^2 f(W^*) + \mathrm{Diag}(\Sigma)\, \nabla R(W^\diamond))^{-1} \mathrm{Diag}(\Sigma)\, \nabla R(W^\diamond)(W^* - W^\diamond)\| \\
&= \Theta(\|W^\diamond - W^*\|). \tag{47}
\end{aligned}$$

We separately show the two parts of Theorem 5. We first show that $\|T(\tilde{W}^*)\| \leq \mathcal{O}(\|W^\diamond - W^*\|^2)$, and $\|T(W^\diamond)\| = \Theta(\|W^\diamond - W^*\|)$ and $\|T(W^*)\| = \Theta(\|W^\diamond - W^*\|)$. It reaches the first limit in (45). Similarly, we show the second limit by showing that $\|\nabla f_\Sigma(\tilde{W}^*)\| \leq \mathcal{O}(\|W^\diamond - W^*\|^2)$, and $\|\nabla f_\Sigma(W^\diamond)\| = \Theta(\|W^\diamond - W^*\|)$ and $\|\nabla f_\Sigma(W^*)\| = \Theta(\|W^\diamond - W^*\|)$.

**(Step 1a) Proof of** $\|\nabla f_\Sigma(\tilde{W}^*)\| \leq \mathcal{O}(\|W^\diamond - W^*\|^2)$. The gradient of $f_\Sigma(W)$ is given by

$$\nabla f_\Sigma(W) = \nabla f(W) + \Sigma \odot R(W). \tag{48}$$

Leveraging the fact that $\nabla f(W^*) = 0$, $\frac{G(W^\diamond)}{F(W^\diamond)} = 0$, as well as Taylor expansion given by

$$\nabla f(\tilde{W}^*) = \nabla^2 f(W^*)(\tilde{W}^* - W^*) + \mathcal{O}((\tilde{W}^* - W^*)^2), \tag{49}$$

$$\frac{G(\tilde{W}^*)}{F(\tilde{W}^*)} = \nabla R(W^\diamond)(\tilde{W}^* - W^\diamond) + \mathcal{O}((\tilde{W}^* - W^\diamond)^2), \tag{50}$$

where $\mathcal{O}((\tilde{W}^* - W^*)^2)$ and $\mathcal{O}((\tilde{W}^* - W^\diamond)^2)$ are vectors with norms $\|\tilde{W}^* - W^*\|^2$ and $\|\tilde{W}^* - W^\diamond\|^2$, respectively. We bound the gradient of the penalized objective as follows

$$\|\nabla f_\Sigma(\tilde{W}^*)\| = \left\| \nabla f(\tilde{W}^*) + \Sigma \odot \frac{G(\tilde{W}^*)}{F(\tilde{W}^*)} \right\| \tag{51}$$

$$= \left\| \nabla^2 f(W^*)(\tilde{W}^* - W^*) + \mathcal{O}((\tilde{W}^* - W^*)^2) + \Sigma \odot (\nabla R(W^\diamond)(\tilde{W}^* - W^\diamond)) + \mathcal{O}((\tilde{W}^* - W^\diamond)^2) \right\|$$

$$= \|\mathcal{O}((\tilde{W}^* - W^*)^2) + \mathcal{O}((\tilde{W}^* - W^\diamond)^2)\| = \mathcal{O}(\|W^* - W^\diamond\|^2)$$

where the last inequality holds by (46) and (47).

**(Step 1b) Proof of** $\|T(W^\diamond)\| = \Theta(\|W^\diamond - W^*\|)$ **and** $\|T(W^*)\| = \Theta(\|W^\diamond - W^*\|)$**.**

$$\|\nabla f_\Sigma(W^\diamond)\| = \|\nabla f(W^\diamond) + \Sigma \odot R(W^\diamond)\| = \|\Sigma \odot R(W^*)\| = \Theta(\|W^* - W^\diamond\|), \tag{52}$$

$$\|\nabla f_\Sigma(W^*)\| = \|\nabla f(W^*) + \Sigma \odot R(W^*)\| = \|\nabla f(W^\diamond)\| = \Theta(\|W^* - W^\diamond\|). \tag{53}$$

**(Step 2a) Proof of** $\|T(\tilde{W}^*)\| \leq \mathcal{O}(\|W^\diamond - W^*\|^2)$**.** By the definition of $T(\tilde{W}^*)$, we have

$$\|T(\tilde{W}^*)\| = \left\| \mathbb{E}_\xi[\nabla f(\tilde{W}^*; \xi)] - \mathbb{E}_\xi[\nabla f(\tilde{W}^*; \xi)] \odot \frac{G(\tilde{W}^*)}{F(\tilde{W}^*)} \right\| \tag{54}$$

$$\leq \left\| \nabla f(\tilde{W}^*) - \mathbb{E}_\xi[\nabla f(\tilde{W}^*; \xi)] \odot \frac{G(\tilde{W}^*)}{F(\tilde{W}^*)} \right\|$$

$$\leq \left\| \nabla f(\tilde{W}^*) - \mathbb{E}_\xi[\nabla f(W^*; \xi)] \odot \frac{G(\tilde{W}^*)}{F(\tilde{W}^*)} \right\| + \left\| (\mathbb{E}_\xi[\nabla f(W^*; \xi)] - \mathbb{E}_\xi[\nabla f(\tilde{W}^*; \xi)]) \odot \frac{G(\tilde{W}^*)}{F(\tilde{W}^*)} \right\|$$

$$= \|\nabla f_\Sigma(\tilde{W}^*)\| + \left\| (\mathbb{E}_\xi[\nabla f(W^*; \xi)] - \mathbb{E}_\xi[\nabla f(\tilde{W}^*; \xi)]) \odot \frac{G(\tilde{W}^*)}{F(\tilde{W}^*)} \right\|$$

The first term in the right-hand side (RHS) of (54) is bounded by (51). By applying $||x| - |y|| \leq |x - y|$ for any $x, y \in \mathbb{R}$ at all components, the second term in the RHS of (54) is bounded by

$$\left\| (\mathbb{E}_\xi[\nabla f(W^*; \xi)] - \mathbb{E}_\xi[\nabla f(\tilde{W}^*; \xi)]) \odot \frac{G(\tilde{W}^*)}{F(\tilde{W}^*)} \right\| \tag{55}$$

$$\leq \left\| \mathbb{E}_\xi[\nabla f(W^*; \xi) - \nabla f(\tilde{W}^*; \xi)] \odot \frac{G(\tilde{W}^*)}{F(\tilde{W}^*)} - \frac{G(W^\diamond)}{F(W^\diamond)} \right\|$$

$$\leq \left\| \mathbb{E}_\xi[\nabla f(W^*; \xi) - \nabla f(\tilde{W}^*; \xi)] \right\| \left\| \frac{G(\tilde{W}^*)}{F(\tilde{W}^*)} - \frac{G(W^\diamond)}{F(W^\diamond)} \right\|$$

$$\leq \mathbb{E}_\xi \left[ \left\| \nabla f(W^*; \xi) - \nabla f(\tilde{W}^*; \xi) \right\| \right] \left\| \frac{G(\tilde{W}^*)}{F(\tilde{W}^*)} - \frac{G(W^\diamond)}{F(W^\diamond)} \right\|$$

$$\leq \mathcal{O}(\|\tilde{W}^* - W^*\| \|\tilde{W}^* - W^\diamond\|) = \mathcal{O}(\|W^* - W^\diamond\|^2).$$

Plugging back (51) and (55) into (54) shows $T(\tilde{W}^*) \leq \mathcal{O}((W^\diamond - W^*)^2)$.

**(Step 2b) Proof of** $\|T(W^\diamond)\| = \Theta(\|W^\diamond - W^*\|)$ **and** $\|T(W^*)\| = \Theta(\|W^\diamond - W^*\|)$**.**

$$\|T(W^\diamond)\| = \left\| \mathbb{E}_\xi[\nabla f(W^\diamond; \xi)] - \mathbb{E}_\xi[\nabla f(W^\diamond; \xi)] \odot \frac{G(W^\diamond)}{F(W^\diamond)} \right\| = \|\nabla f(W^\diamond)\| \tag{56}$$
$$= \Theta(\|W^* - W^\diamond\|),$$

$$\|T(W^*)\| = \left\| \mathbb{E}_\xi[\nabla f(W^*; \xi)] - \mathbb{E}_\xi[\nabla f(W^*; \xi)] \odot \frac{G(W^*)}{F(W^*)} \right\| = \|\Sigma \odot R(W^*)\| \tag{57}$$
$$= \Theta(\|W^* - W^\diamond\|).$$

Now we complete the proof. $\square$

**Special case with** $D = 1$**.** Before proceeding to the next section, we also present a special case where the response functions are linear, i.e., $Q_+(W) = 1 - \frac{W}{\tau}$, $Q_-(W) = 1 + \frac{W}{\tau}$. $F(W) = 1$ and $G(W) = \frac{W}{\tau}$ based on definition (4); and hence $R(W) = \frac{G(W)}{F(W)} = \frac{W}{\tau}$. Accordingly, the accumulated asymmetric function is given by

$$[R_c(W)]_d = \int_{\tau_i^{\min}}^{[W]_d} [R(W)]_d \, d[W]_d = \int_{\tau_i^{\min}}^{[W]_d} \frac{[W]_d}{\tau} \, d[W]_d \tag{58}$$

$$= \frac{1}{2\tau}([W]_d)^2 - \frac{1}{2\tau}(\tau_i^{\min})^2.$$

Therefore, the last term in the objective (7) becomes

$$\langle \Sigma, R_c(W) \rangle = \sum_{i=1}^{D}[\Sigma]_d[R_c(W)]_d = \sum_{i=1}^{D}[\Sigma]_d\left(\frac{1}{2\tau}([W]_d)^2 - \frac{1}{2\tau}(\tau_i^{\min})^2\right) \qquad (59)$$

$$= \frac{1}{2\tau}\|W\|_{\Sigma}^2 + \text{const.}$$

which is a weighted $\ell_2$ norm regularization term. Furthermore, if $W$ is a scalar, i.e., $D = 1$, (44) reduces to $\min_W \ f_\Sigma(W) := f(W) + \frac{\Sigma}{2\tau}\|W\|^2$, which is a $\ell_2$-regularized problem with an approximated solution

$$W^S := \frac{f''(W^*)\,W^* - R'(W^\diamond)\Sigma\,W^\diamond}{f''(W^*) - R'(W^\diamond)\Sigma}. \qquad (60)$$

# H Proof of Theorem 2: Convergence of Analog SGD

**Theorem 2** (Inexact convergence of `Analog SGD`). *Under Assumption 1–2, if the learning rate is set as $\alpha = O(1/\sqrt{K})$, it holds that*

$$E_K^{\textit{ASGD}} \leq O\left(\sqrt{\sigma^2/K} + \sigma^2 S_K^{\textit{ASGD}}\right) \qquad (8)$$

*where $S_K^{\textit{ASGD}}$ denotes the amplification factor given by $S_K^{\textit{ASGD}} := \frac{1}{K}\sum_{k=0}^{K-1}\left\|\frac{G(W_k)}{\sqrt{F(W_k)}}\right\|_\infty^2$.*

*Proof of Theorem 2.* The $L$-smooth assumption (Assumption 1) implies that

$$\mathbb{E}_{\xi_k}[f(W_{k+1})] \leq f(W_k) + \underbrace{\mathbb{E}_{\xi_k}[\langle \nabla f(W_k), W_{k+1} - W_k \rangle]}_{(a)} + \underbrace{\frac{L}{2}\mathbb{E}_{\xi_k}[\|W_{k+1} - W_k\|^2]}_{(b)}. \qquad (61)$$

Next, we will handle the second and the third terms in the RHS of (61) separately.

**Bound of the second term (a).** To bound term (a) in the RHS of (61), we leverage the assumption that noise has expectation 0 (Assumption 2)

$$\mathbb{E}_{\xi_k}[\langle \nabla f(W_k), W_{k+1} - W_k \rangle] \qquad (62)$$

$$= \alpha\mathbb{E}_{\xi_k}\left[\left\langle \nabla f(W_k) \odot \sqrt{F(W_k)}, \frac{W_{k+1} - W_k}{\alpha\sqrt{F(W_k)}} + (\nabla f(W_k;\xi_k) - \nabla f(W_k)) \odot \sqrt{F(W_k)}\right\rangle\right]$$

$$= -\frac{\alpha}{2}\|\nabla f(W_k) \odot \sqrt{F(W_k)}\|^2$$

$$-\frac{1}{2\alpha}\mathbb{E}_{\xi_k}\left[\left\|\frac{W_{k+1} - W_k}{\sqrt{F(W_k)}} + \alpha(\nabla f(W_k;\xi_k) - \nabla f(W_k)) \odot \sqrt{F(W_k)}\right\|^2\right]$$

$$+\frac{1}{2\alpha}\mathbb{E}_{\xi_k}\left[\left\|\frac{W_{k+1} - W_k}{\sqrt{F(W_k)}} + \alpha\nabla f(W_k;\xi_k) \odot \sqrt{F(W_k)}\right\|^2\right].$$

The second term of the RHS of (62) is bounded by

$$\frac{1}{2\alpha}\mathbb{E}_{\xi_k}\left[\left\|\frac{W_{k+1} - W_k}{\sqrt{F(W_k)}} + \alpha(\nabla f(W_k;\xi_k) - \nabla f(W_k)) \odot \sqrt{F(W_k)}\right\|^2\right] \qquad (63)$$

$$= \frac{1}{2\alpha}\mathbb{E}_{\xi_k}\left[\left\|\frac{W_{k+1} - W_k + \alpha(\nabla f(W_k;\xi_k) - \nabla f(W_k)) \odot F(W_k)}{\sqrt{F(W_k)}}\right\|^2\right]$$

$$\geq \frac{1}{2\alpha F_{\max}} \mathbb{E}_{\xi_k} \left[ \| W_{k+1} - W_k + \alpha (\nabla f(W_k; \xi_k) - \nabla f(W_k)) \odot F(W_k) \|^2 \right].$$

The third term in the RHS of (62) can be bounded by variance decomposition and bounded variance assumption (Assumption 2)

$$\frac{1}{2\alpha} \mathbb{E}_{\xi_k} \left[ \left\| \frac{W_{k+1} - W_k}{\sqrt{F(W_k)}} + \alpha \nabla f(W_k; \xi_k) \odot \sqrt{F(W_k)} \right\|^2 \right] \tag{64}$$

$$= \frac{\alpha}{2} \mathbb{E}_{\xi_k} \left[ \left\| |\nabla f(W_k; \xi_k)| \odot \frac{G(W_k)}{\sqrt{F(W_k)}} \right\|^2 \right]$$

$$\leq \frac{\alpha}{2} \left\| |\nabla f(W_k)| \odot \frac{G(W_k)}{\sqrt{F(W_k)}} \right\|^2 + \frac{\alpha \sigma^2}{2} \left\| \frac{G(W_k)}{\sqrt{F(W_k)}} \right\|_\infty^2.$$

Define the saturation vector $M(W_k) \in \mathbb{R}^D$ by

$$M(W_k) := F(W_k)^{\odot 2} - G(W_k)^{\odot 2} = (F(W_k) + G(W_k)) \odot (F(W_k) - G(W_k)) \tag{65}$$
$$= Q_+(W_k) \odot Q_-(W_k).$$

Note that the first term in the RHS of (62) and the second term in the RHS of (64) can be bounded by

$$-\frac{\alpha}{2} \| \nabla f(W_k) \odot \sqrt{F(W_k)} \|^2 + \frac{\alpha}{2} \left\| |\nabla f(W_k)| \odot \frac{G(W_k)}{\sqrt{F(W_k)}} \right\|^2 \tag{66}$$

$$= -\frac{\alpha}{2} \sum_{d \in [D]} \left( [\nabla f(W_k)]_d^2 \left( [F(W_k)]_d - \frac{[G(W_k)]_d^2}{[F(W_k)]_d} \right) \right)$$

$$= -\frac{\alpha}{2} \sum_{d \in [D]} \left( [\nabla f(W_k)]_d^2 \left( \frac{[F(W_k)]_d^2 - [G(W_k)]_d^2}{[F(W_k)]_d} \right) \right)$$

$$\leq -\frac{\alpha}{2F_{\max}} \sum_{d \in [D]} \left( [\nabla f(W_k)]_d^2 \left( [F(W_k)]_d^2 - [G(W_k)]_d^2 \right) \right)$$

$$= -\frac{\alpha}{2F_{\max}} \| \nabla f(W_k) \|_{M(W_k)}^2 \leq 0.$$

Plugging (63) to (66) into (62), we bound the term (a) by

$$\mathbb{E}_{\xi_k} [\langle \nabla f(W_k), W_{k+1} - W_k \rangle] \tag{67}$$

$$= -\frac{\alpha}{2F_{\max}} \| \nabla f(W_k) \|_{M(W_k)}^2 + \frac{\alpha \sigma^2}{2} \left\| \frac{G(W_k)}{\sqrt{F(W_k)}} \right\|_\infty^2$$

$$- \frac{1}{2\alpha F_{\max}} \mathbb{E}_{\xi_k} \left[ \| W_{k+1} - W_k + \alpha (\nabla f(W_k; \xi_k) - \nabla f(W_k)) \odot F(W_k) \|^2 \right].$$

**Bound of the third term (b).** The third term (b) in the RHS of (61) is bounded by

$$\frac{L}{2} \mathbb{E}_{\xi_k} [\| W_{k+1} - W_k \|^2] \tag{68}$$

$$\leq L \mathbb{E}_{\xi_k} [\| W_{k+1} - W_k + \alpha (\nabla f(W_k; \xi_k) - \nabla f(W_k)) \odot F(W_k) \|^2]$$

$$+ \alpha^2 L \mathbb{E}_{\xi_k} [\| (\nabla f(W_k; \xi_k) - \nabla f(W_k)) \odot F(W_k) \|^2]$$

$$\leq L \mathbb{E}_{\xi_k} [\| W_{k+1} - W_k + \alpha (\nabla f(W_k; \xi_k) - \nabla f(W_k)) \odot F(W_k) \|^2] + \alpha^2 L F_{\max}^2 \sigma^2$$

where the last inequality leverages the bounded variance of noise (Assumption 2) and the fact that $F(W_k)$ is bounded by $F_{\max}$ element-wise.

Substituting (67) and (68) back into (61), we have

$$\mathbb{E}_{\xi_k} [f(W_{k+1})] \tag{69}$$

$$\leq f(W_k) - \frac{\alpha}{2F_{\max}}\|\nabla f(W_k)\|_{M(W_k)}^2 + \alpha^2 LF_{\max}^2\sigma^2 + \frac{\alpha\sigma^2}{2}\left\|\frac{G(W_k)}{\sqrt{F(W_k)}}\right\|_\infty^2$$

$$- \frac{1}{F_{\max}}\left(\frac{1}{2\alpha} - LF_{\max}\right)\mathbb{E}_{\xi_k}[\|W_{k+1} - W_k + \alpha(\nabla f(W_k;\xi_k) - \nabla f(W_k))\odot F(W_k)\|^2].$$

The third term in the RHS of (69) can be bounded by

$$\mathbb{E}_{\xi_k}[\|W_{k+1} - W_k + \alpha(\nabla f(W_k;\xi_k) - \nabla f(W_k))\odot F(W_k)\|^2] \tag{70}$$

$$= \alpha^2\mathbb{E}_{\xi_k}[\|\nabla f(W_k)\odot F(W_k) + |\nabla f(W_k;\xi_k)|\odot G(W_k)\|^2]$$

$$\geq \frac{1}{2}\alpha^2\mathbb{E}_{\xi_k}[\|\nabla f(W_k)\odot F(W_k) + |\nabla f(W_k)|\odot G(W_k)\|^2]$$

$$- \alpha^2\mathbb{E}_{\xi_k}[\|(|\nabla f(W_k)| - |\nabla f(W_k;\xi_k)|)\odot G(W_k)\|^2]$$

$$\geq \frac{1}{2}\alpha^2\mathbb{E}_{\xi_k}[\|\nabla f(W_k)\odot F(W_k) + |\nabla f(W_k)|\odot G(W_k)\|^2]$$

$$- \alpha^2\mathbb{E}_{\xi_k}[\|(\nabla f(W_k) - \nabla f(W_k;\xi_k))\odot G(W_k)\|^2]$$

$$\geq \frac{1}{2}\alpha^2\mathbb{E}_{\xi_k}[\|\nabla f(W_k)\odot F(W_k) + |\nabla f(W_k)|\odot G(W_k)\|^2] - \alpha^2 F_{\max}\sigma^2\left\|\frac{G(W_k)}{\sqrt{F(W_k)}}\right\|_\infty^2$$

where the first inequality holds because $\|x\|^2 \geq \frac{1}{2}\|x - y\|^2 - \|y\|^2$ for any $x, y \in \mathbb{R}^D$, the second inequality comes from $||x| - |y|| \leq |x - y|$ for any $x, y \in \mathbb{R}$, and the last inequality holds because

$$\mathbb{E}_{\xi_k}[\|(\nabla f(W_k) - \nabla f(W_k;\xi_k))\odot G(W_k)\|^2] \tag{71}$$

$$= \mathbb{E}_{\xi_k}\left[\left\|(\nabla f(W_k) - \nabla f(W_k;\xi_k))\odot \frac{G(W_k)}{\sqrt{F(W_k)}}\odot\sqrt{F(W_k)}\right\|^2\right]$$

$$\leq F_{\max}\mathbb{E}_{\xi_k}\left[\left\|(\nabla f(W_k) - \nabla f(W_k;\xi_k))\odot \frac{G(W_k)}{\sqrt{F(W_k)}}\right\|^2\right]$$

$$\leq F_{\max}\mathbb{E}_{\xi_k}\left[\|\nabla f(W_k) - \nabla f(W_k;\xi_k)\|^2\right]\left\|\frac{G(W_k)}{\sqrt{F(W_k)}}\right\|_\infty^2$$

$$\leq F_{\max}\sigma^2\left\|\frac{G(W_k)}{\sqrt{F(W_k)}}\right\|_\infty^2.$$

The learning rate $\alpha \leq \frac{1}{4LF_{\max}}$ implies that $\frac{1}{2\alpha} - LF_{\max} \leq \frac{1}{4\alpha}$ in (69), which leads (61) to

$$\mathbb{E}_{\xi_k}[f(W_{k+1})] \leq f(W_k) - \frac{\alpha}{2F_{\max}}\|\nabla f(W_k)\|_{M(W_k)}^2 + \alpha^2 LF_{\max}^2\sigma^2 + \alpha\sigma^2\left\|\frac{G(W_k)}{\sqrt{F(W_k)}}\right\|_\infty^2 \tag{72}$$

$$- \frac{\alpha}{8F_{\max}}\|\nabla f(W_k)\odot F(W_k) + |\nabla f(W_k)|\odot G(W_k)\|^2.$$

Reorganizing (72), taking expectation over all $\xi_K, \xi_{K-1}, \cdots, \xi_0$, and averaging them for $k$ from 0 to $K - 1$ deduce that

$$E_K^{\texttt{ASGD}} = \frac{1}{K}\sum_{k=0}^{K}\mathbb{E}[\|\nabla f(W_k)\odot F(W_k) + |\nabla f(W_k)|\odot G(W_k)\|^2 + 4\|\nabla f(W_k)\|_{M(W_k)}^2] \tag{73}$$

$$\leq \frac{8F_{\max}(f(W_0) - \mathbb{E}[f(W_{k+1})])}{\alpha K} + 8\alpha LF_{\max}^3\sigma^2 + 8F_{\max}\sigma^2 \times \frac{1}{K}\sum_{k=0}^{K-1}\left\|\frac{G(W_k)}{\sqrt{F(W_k)}}\right\|_\infty^2$$

$$\leq \frac{8F_{\max}(f(W_0) - f^*)}{\alpha K} + 8\alpha LF_{\max}^3\sigma^2 + 8F_{\max}\sigma^2 \times \frac{1}{K}\sum_{k=0}^{K-1}\left\|\frac{G(W_k)}{\sqrt{F(W_k)}}\right\|_\infty^2$$

$$= 16 F_{\max}^2 \sqrt{\frac{(f(W_0) - f^*)\sigma^2 L}{K}} + 8 F_{\max}\sigma^2 S_K^{\text{ASGD}}$$

where the last equality chooses the learning rate as $\alpha = \frac{1}{F_{\max}}\sqrt{\frac{f(W_0) - f^*}{\sigma^2 L K}}$. The proof is completed. □

**Remark 1** (Tighter bound without saturation). *Assuming the saturation never happens during the training, i.e. $M(W_k) \geq M_{\min}^{RL} > 0$ for all $k \in [K]$, we get a tighter bound in* (72) *by leveraging $\|\nabla f(W_k)\|_{M(W_k)}^2 \geq \min\{M(W_k)\} \|\nabla f(W_k)\|^2 \geq M_{\min}^{RL} \|\nabla f(W_k)\|^2$*

$$\mathbb{E}_{\xi_k}[f(W_{k+1})] \leq f(W_k) - \frac{\alpha}{2F_{\max}}\|\nabla f(W_k)\|_{M(W_k)}^2 + \alpha^2 L F_{\max}^2 \sigma^2 + \alpha\sigma^2 \left\|\frac{G(W_k)}{\sqrt{F(W_k)}}\right\|_\infty^2 \quad (74)$$

*which leads to*

$$\frac{1}{K}\sum_{k=0}^{K}[\|\nabla f(W_k)\|^2] \quad (75)$$

$$= \frac{4F_{\max}^2}{M_{\min}^{RL}}\sqrt{\frac{(f(W_0) - f^*)\sigma^2 L}{K}} + 2F_{\max}\sigma^2 \times \frac{1}{K}\sum_{k=0}^{K-1}\left\|\frac{G(W_k)}{\sqrt{F(W_k)}}\right\|_\infty^2 \bigg/ \min\{M(W_k)\}.$$

*It exactly reduces to the result for the convergence of `Analog SGD` in [21] on special linear repsonse functions, as discussed in Appendix B.*

# I   Proof of Theorem 3: Convergence of Residual Learning

This section provides the convergence guarantee of the `Residual Learning` under the strongly convex assumption.

**Theorem 3** (Convergence of `Residual Learning`). *Under Assumptions 1–3, with the learning rate $\alpha = O\left(\sqrt{1/\sigma^2 K}\right)$, $\beta = O(\alpha\gamma^{3/2})$, it holds for `Residual Learning` that*

$$E_K^{RL} \leq O\left(\sqrt{\sigma^2/K} + \sigma^2 S_K^{RL}\right) \quad (13)$$

*where $S_K^{RL}$ denotes the amplification factor of $P_k$ given by $S_K^{RL} := \frac{1}{K}\sum_{k=0}^{K}\left\|\frac{G(P_k)}{\sqrt{F(P_k)}}\right\|_\infty^2$.*

## I.1   Main proof

*Proof of Theorem 3.* The proof relies on the following two lemmas, which provide the sufficient descent of $W_k$ and $\bar{W}_k$, respectively.

**Lemma 5** (Descent Lemma of $\bar{W}_k$). *Suppose Assumptions 1–2 hold. It holds for `Residual Learning` that*

$$\mathbb{E}_{\xi_k}[f(\bar{W}_{k+1})] \leq f(\bar{W}_k) - \frac{\alpha}{4F_{\max}}\|\nabla f(\bar{W}_k)\|_{M(P_k)}^2 + 2\alpha\sigma^2\left\|\frac{G(P_k)}{\sqrt{F(P_k)}}\right\|_\infty^2 + 2\alpha^2 L F_{\max}^2 \sigma^2$$

$$(76)$$

$$- \frac{\alpha\gamma}{8F_{\max}}\|\nabla f(\bar{W}_k) \odot F(P_k) + |\nabla f(\bar{W}_k)| \odot G(P_k)\|^2$$

$$+ \frac{F_{\max}}{\alpha}\mathbb{E}_{\xi_k}\left[\|W_{k+1} - W_k\|_{M(P_k)^\dagger}^2\right] + \mathbb{E}_{\xi_k}[\|W_{k+1} - W_k\|^2].$$

**Lemma 6** (Descent Lemma of $W_k$). *It holds for `Residual Learning` that*

$$\|W_{k+1} - W^*\|^2 \leq \|W_k - W^*\|^2 - \frac{\beta}{2\gamma F_{\max}}\|W_k - W^*\|_{M(W_k)}^2 \quad (77)$$

$$- \frac{\beta\gamma}{2F_{\max}} \|P^*(W_k) \odot F(W_k) - |P^*(W_k)| \odot G(W_k)\|^2$$

$$+ \frac{2\beta F_{\max}^3}{\gamma} \|P_{k+1} - P^*(W_k)\|_{M(W_k)^\dagger}^2 + 2\beta^2 \|P_{k+1} - P^*(W_k)\|^2.$$

The proof of Lemma 5 and 6 are deferred to Section I.2 and I.3, respectively.

For a sufficiently large $\gamma$, $P^*(W_k)$ is ensured to be located in the dynamic range of the analog array $P_k$. Therefore, we may assume both $q_+(P_k)$ and $q_-(P_k)$ are non-zero, equivalently, there exists a non-zero constant $M_{\min}^{\mathtt{RL}}$ such that $\min\{M(P_k)\} \geq M_{\min}^{\mathtt{RL}}$ for all $k$. Under this condition, we have the following inequalities

$$\frac{\alpha}{4F_{\max}} \|\nabla f(\bar{W}_k)\|_{M(P_k)}^2 \geq \frac{\alpha M_{\min}^{\mathtt{RL}}}{4F_{\max}} \|\nabla f(\bar{W}_k)\|^2, \tag{78}$$

$$\frac{F_{\max}}{\alpha\gamma} \|W_{k+1} - W_k\|_{M(P_k)^\dagger}^2 \leq \frac{F_{\max}}{\alpha\gamma M_{\min}^{\mathtt{RL}}} \|W_{k+1} - W_k\|^2. \tag{79}$$

Similarly, we bound the term $\|P_{k+1} - P^*(W_k)\|_{M(W_k)^\dagger}^2$ in (76) by

$$\frac{2\beta F_{\max}^3}{\gamma} \|P_{k+1} - P^*(W_k)\|_{M(W_k)^\dagger}^2 \leq \frac{2\beta F_{\max}^3}{\gamma \min\{M(W_k)\}} \|P_{k+1} - P^*(W_k)\|^2. \tag{80}$$

Notice it is only required to have $\min\{M(W_k)\} > 0$ for the inequality to hold.

By inequality (79), the last two terms in the RHS of (76) is bounded by

$$\frac{F_{\max}}{\alpha} \mathbb{E}_{\xi_k} \left[ \|W_{k+1} - W_k\|_{M(P_k)^\dagger}^2 \right] + \mathbb{E}_{\xi_k} [\|W_{k+1} - W_k\|^2] \tag{81}$$

$$= \frac{F_{\max}}{\alpha M_{\min}^{\mathtt{RL}}} \mathbb{E}_{\xi_k} \left[ \|W_{k+1} - W_k\|^2 \right] + \mathbb{E}_{\xi_k} [\|W_{k+1} - W_k\|^2]$$

$$\overset{(a)}{\leq} \frac{2F_{\max}}{\alpha M_{\min}^{\mathtt{RL}}} \mathbb{E}_{\xi_k} \left[ \|W_{k+1} - W_k\|^2 \right] = \frac{2\beta^2 F_{\max}}{\alpha M_{\min}^{\mathtt{RL}}} \|P_{k+1} \odot F(W_k) - |P_{k+1}| \odot G(W_k)\|^2$$

$$\leq \frac{4\beta^2 F_{\max}}{\alpha M_{\min}^{\mathtt{RL}}} \|P^*(W_k) \odot F(W_k) - |P^*(W_k)| \odot G(W_k)\|^2$$

$$+ \frac{4\beta^2 F_{\max}}{\alpha M_{\min}^{\mathtt{RL}}} \|P_{k+1} \odot F(W_k) - |P_{k+1}| \odot G(W_k) - (P^*(W_k) \odot F(W_k) - |P^*(W_k)| \odot G(W_k))\|^2$$

$$\overset{(b)}{\leq} \frac{4\beta^2 F_{\max}}{\alpha M_{\min}^{\mathtt{RL}}} \|P^*(W_k) \odot F(W_k) - |P^*(W_k)| \odot G(W_k)\|^2 + \frac{4\beta^2 F_{\max}}{\alpha M_{\min}^{\mathtt{RL}}} \|P_{k+1} - P^*(W_k)\|^2.$$

where $(a)$ holds if learning rate $\alpha$ is sufficiently small such that $\frac{F_{\max}}{\alpha\gamma M_{\min}^{\mathtt{RL}}} \geq 1$; $(b)$ comes from the Lipschitz continuity of the analog update (c.f. Lemma 3).

With all the inequalities and lemmas above, we are ready to prove the main conclusion in Theorem 3 now. Define a Lyapunov function by

$$\mathbb{V}_k := f(\bar{W}_k) - f^* + C \|W_k - W^*\|^2. \tag{82}$$

By Lemmas 5 and 6, we show that $\mathbb{V}_k$ has sufficient descent in expectation

$$\mathbb{E}_{\xi_k} [\mathbb{V}_{k+1}] \tag{83}$$

$$= \mathbb{E}_{\xi_k} \left[ f(\bar{W}_{k+1}) - f^* + C \|W_{k+1} - W^*\|^2 \right]$$

$$\leq f(\bar{W}_k) - f^* - \frac{\alpha}{4F_{\max}} \|\nabla f(\bar{W}_k)\|_{M(P_k)}^2 + 2\alpha\sigma^2 \left\| \frac{G(P_k)}{\sqrt{F(P_k)}} \right\|_\infty^2 + 2\alpha^2 L F_{\max}^2 \sigma^2$$

$$- \frac{\alpha}{8F_{\max}} \|\nabla f(\bar{W}_k) \odot F(P_k) + |\nabla f(\bar{W}_k)| \odot G(P_k)\|^2$$

$$+ \frac{4\beta^2 F_{\max}}{\alpha M_{\min}^{\mathtt{RL}}} \|P^*(W_k) \odot F(W_k) - |P^*(W_k)| \odot G(W_k)\|^2 + \frac{4\beta^2 F_{\max}}{\alpha M_{\min}^{\mathtt{RL}}} \mathbb{E}_{\xi_k} [\|P_{k+1} - P^*(W_k)\|^2]$$

$$+ C\left( \|W_k - W^*\|^2 - \frac{\beta}{2\gamma F_{\max}} \|W_k - W^*\|^2_{M(W_k)} + \frac{3\beta F_{\max}^3}{\gamma \min\{M(W_k)\}} \mathbb{E}_{\xi_k}[\|P_{k+1} - P^*(W_k)\|^2] \right.$$

$$\left. - \frac{\beta\gamma}{2F_{\max}} \|P^*(W_k) \odot F(W_k) - |P^*(W_k)| \odot G(W_k)\|^2 \right)$$

$$\leq \mathbb{V}_k - \frac{\alpha M_{\min}^{\mathtt{RL}}}{4F_{\max}} \|\nabla f(\bar{W}_k)\|^2 + 2\alpha\sigma^2 \left\| \frac{G(P_k)}{\sqrt{F(P_k)}} \right\|_\infty^2 + 2\alpha^2 L F_{\max}^2 \sigma^2$$

$$- \frac{\alpha}{8F_{\max}} \|\nabla f(\bar{W}_k) \odot F(P_k) + |\nabla f(\bar{W}_k)| \odot G(P_k)\|^2$$

$$- \left( \frac{\beta\gamma}{2F_{\max}} C - \frac{4\beta^2 F_{\max}}{\alpha M_{\min}^{\mathtt{RL}}} \right) \|P^*(W_k) \odot F(W_k) - |P^*(W_k)| \odot G(W_k)\|^2$$

$$+ \left( \frac{3\beta F_{\max}^3}{\gamma \min\{M(W_k)\}} C + \frac{4\beta^2 F_{\max}}{\alpha M_{\min}^{\mathtt{RL}}} \right) \mathbb{E}_{\xi_k}[\|P_{k+1} - P^*(W_k)\|^2] - \frac{\beta}{2\gamma F_{\max}} C \|W_k - W^*\|^2_{M(W_k)}.$$

Let $C = \frac{10\beta F_{\max}^2}{\alpha M_{\min}^{\mathtt{RL}} \gamma}$, which leads to the positive coefficient in front of $\|P_{k+1} - P^*(W_k)\|^2$, i.e.,

$$\mathbb{E}_{\xi_k}[\mathbb{V}_{k+1}] \tag{84}$$

$$\leq \mathbb{V}_k - \frac{\alpha M_{\min}^{\mathtt{RL}}}{4F_{\max}} \|\nabla f(\bar{W}_k)\|^2 + 2\alpha\sigma^2 \left\| \frac{G(P_k)}{\sqrt{F(P_k)}} \right\|_\infty^2 + 2\alpha^2 L F_{\max}^2 \sigma^2$$

$$- \frac{\alpha}{8F_{\max}} \|\nabla f(\bar{W}_k) \odot F(P_k) + |\nabla f(\bar{W}_k)| \odot G(P_k)\|^2$$

$$- \frac{\beta^2 F_{\max}}{\alpha M_{\min}^{\mathtt{RL}}} \|P^*(W_k) \odot F(W_k) - |P^*(W_k)| \odot G(W_k)\|^2$$

$$+ \left( \frac{30\beta^2 F_{\max}^5}{\alpha\gamma \min\{M(W_k)\} M_{\min}^{\mathtt{RL}}} + \frac{4\beta^2 F_{\max}}{\alpha M_{\min}^{\mathtt{RL}}} \right) \mathbb{E}_{\xi_k}[\|P_{k+1} - P^*(W_k)\|^2] - \frac{5\beta^2 F_{\max}}{\alpha M_{\min}^{\mathtt{RL}}} \|W_k - W^*\|^2_{M(W_k)}.$$

Notice that the $\|P_{k+1} - P^*(W_k)\|^2$ appears in the RHS above, we also need the following lemma to bound it in terms of $\|P_k - P^*(W_k)\|^2$.

**Lemma 7** (Descent Lemma of $P_k$). *Suppose Assumptions 1-2 and 3 hold. It holds for* `Tiki-Taka` *that*

$$\mathbb{E}_{\xi_k}[\|P_{k+1} - P^*(W_k)\|^2] \tag{85}$$

$$\leq \left( 1 - \frac{\alpha\gamma\mu L}{4(\mu + L)} \right) \|P_k - P^*(W_k)\|^2 + \frac{2\alpha(\mu + L)F_{\max}\sigma^2}{\gamma\mu L} \left\| \frac{G(P_k)}{\sqrt{F(P_k)}} \right\|_\infty^2 + \alpha^2 F_{\max}^2 \sigma^2.$$

The proof of Lemma 7 is deferred to Section I.4. By Lemma 7, we bound the $\|P_{k+1} - P^*(W_k)\|^2$ in terms of $\|P_k - P^*(W_k)\|^2$ as

$$\left( \frac{30\beta^2 F_{\max}^5}{\alpha\gamma \min\{M(W_k)\} M_{\min}^{\mathtt{RL}}} + \frac{4\beta^2 F_{\max}}{\alpha M_{\min}^{\mathtt{RL}}} \right) \mathbb{E}_{\xi_k}[\|P_{k+1} - P^*(W_k)\|^2] \tag{86}$$

$$\overset{(a)}{\leq} \frac{32\beta^2 F_{\max}^5}{\alpha\gamma \min\{M(W_k)\} M_{\min}^{\mathtt{RL}}} \mathbb{E}_{\xi_k}[\|P_{k+1} - P^*(W_k)\|^2]$$

$$\leq \frac{32\beta^2 F_{\max}^5}{\alpha\gamma \min\{M(W_k)\} M_{\min}^{\mathtt{RL}}} \left( 1 - \frac{\alpha}{4} \frac{\mu L}{\gamma(\mu + L)} \right) \|P_k - P^*(W_k)\|^2$$

$$+ \frac{32\beta^2 F_{\max}^5}{\alpha\gamma \min\{M(W_k)\} M_{\min}^{\mathtt{RL}}} \left( \frac{2\alpha(\mu + L)F_{\max}\sigma^2}{\gamma\mu L} \left\| \frac{G(P_k)}{\sqrt{F(P_k)}} \right\|_\infty^2 + \alpha^2 F_{\max}^2 \sigma^2 \right)$$

$$\leq \frac{32\beta^2 F_{\max}^5}{\alpha\gamma \min\{M(W_k)\} M_{\min}^{\mathtt{RL}}} \|P_k - P^*(W_k)\|^2 + O\left( \beta^2\sigma^2 \left\| \frac{G(P_k)}{\sqrt{F(P_k)}} \right\|_\infty^2 + \alpha\beta^2 F_{\max}^2 \sigma^2 \right)$$

$$\overset{(b)}{\leq} \frac{32\beta^2 F_{\max}^5}{\alpha\gamma\min\{M(W_k)\}M_{\min}^{\text{RL}}}\|P_k - P^*(W_k)\|^2 + \alpha\sigma^2\left\|\frac{G(P_k)}{\sqrt{F(P_k)}}\right\|_\infty^2 + \alpha^2 LF_{\max}^2\sigma^2$$

where $(a)$ assumes $\frac{4\beta^2 F_{\max}}{\alpha M_{\min}^{\text{RL}}} \leq \frac{2\beta^2 F_{\max}^5}{\alpha\gamma\min\{M(W_k)\}M_{\min}^{\text{RL}}}$ with lost of generality to keep the formulations simple since $\gamma\min\{M(W_k)\}$ is typically small; (b) holds given $\alpha$ and $\beta$ is sufficiently small. In addition, the strong convexity of the objective (c.f. Assumption 3) implies that

$$\frac{\alpha M_{\min}^{\text{RL}}}{8F_{\max}}\|\nabla f(\bar{W}_k)\|^2 \geq \frac{\alpha\mu^2 M_{\min}^{\text{RL}}}{8F_{\max}}\|\bar{W}_k - W^*\|^2 = \frac{\alpha\mu^2 M_{\min}^{\text{RL}}}{8F_{\max}}\|W_k + \gamma P_k - W^*\|^2 \tag{87}$$

$$= \frac{\alpha\mu^2\gamma^2 M_{\min}^{\text{RL}}}{8F_{\max}}\left\|P_k - \frac{W^* - W_k}{\gamma}\right\|^2 = \frac{\alpha\mu^2\gamma^2 M_{\min}^{\text{RL}}}{8F_{\max}}\|P_k - P^*(W_k)\|^2.$$

Substituting (86) and (87) back into (84) yields

$$\mathbb{E}_{\xi_k}[\mathbb{V}_{k+1}] \tag{88}$$

$$\leq \mathbb{V}_k - \frac{\alpha M_{\min}^{\text{RL}}}{8F_{\max}}\|\nabla f(\bar{W}_k)\|^2 + 3\alpha\sigma^2\left\|\frac{G(P_k)}{\sqrt{F(P_k)}}\right\|_\infty^2 + 3\alpha^2 LF_{\max}^2\sigma^2$$

$$- \frac{\alpha}{8F_{\max}}\|\nabla f(\bar{W}_k)\odot F(P_k) + |\nabla f(\bar{W}_k)|\odot G(P_k)\|^2$$

$$- \frac{\beta^2 F_{\max}}{\alpha M_{\min}^{\text{RL}}}\|P^*(W_k)\odot F(W_k) - |P^*(W_k)|\odot G(W_k)\|^2$$

$$- \left(\frac{\alpha\mu^2\gamma^2 M_{\min}^{\text{RL}}}{8F_{\max}} - \frac{32\beta^2 F_{\max}^5}{\alpha\gamma\min\{M(W_k)\}M_{\min}^{\text{RL}}}\right)\|P_k - P^*(W_k)\|^2 - \frac{5\beta^2 F_{\max}}{\alpha M_{\min}^{\text{RL}}}\|W_k - W^*\|_{M(W_k)}^2$$

$$= \mathbb{V}_k - \frac{\alpha M_{\min}^{\text{RL}}}{8F_{\max}}\|\nabla f(\bar{W}_k)\|^2 + 3\alpha\sigma^2\left\|\frac{G(P_k)}{\sqrt{F(P_k)}}\right\|_\infty^2 + 3\alpha^2 LF_{\max}^2\sigma^2$$

$$- \frac{\alpha}{8F_{\max}}\|\nabla f(\bar{W}_k)\odot F(P_k) + |\nabla f(\bar{W}_k)|\odot G(P_k)\|^2$$

$$- \frac{\alpha\mu^2\gamma^3\min\{M(W_k)\}}{512F_{\max}^5 M_{\min}^{\text{RL}}}\|P^*(W_k)\odot F(W_k) - |P^*(W_k)|\odot G(W_k)\|^2$$

$$- \frac{\alpha\mu^2\gamma^2 M_{\min}^{\text{RL}}}{16F_{\max}}\|P_k - P^*(W_k)\|^2 - \frac{5\alpha\mu^2\gamma^3}{512F_{\max}^5 M_{\min}^{\text{RL}}}\|W_k - W^*\|_{M(W_k)}^2$$

where the last step chooses the transfer learning rate by

$$\beta = \frac{\alpha\mu\gamma^{\frac{3}{2}}\sqrt{\min\{M(W_k)\}}M_{\min}^{\text{RL}}}{16\sqrt{2}F_{\max}^3}. \tag{89}$$

Rearranging inequality (83) above, we have

$$\frac{\alpha}{8F_{\max}}\|\nabla f(\bar{W}_k)\odot F(P_k) + |\nabla f(\bar{W}_k)|\odot G(P_k)\|^2 + \frac{\alpha}{8F_{\max}M_{\min}^{\text{RL}}}\|\nabla f(\bar{W}_k)\|^2 \tag{90}$$

$$+ \frac{\alpha\mu^2\gamma^3\min\{M(W_k)\}}{512F_{\max}^5 M_{\min}^{\text{RL}}}\|P^*(W_k)\odot F(W_k) - |P^*(W_k)|\odot G(W_k)\|^2$$

$$+ \frac{5\alpha\mu^2\gamma^3\min\{M(W_k)\}}{512F_{\max}^5 M_{\min}^{\text{RL}}}\|W_k - W^*\|_{M(W_k)}^2 + \frac{\alpha\mu^2\gamma^2 M_{\min}^{\text{RL}}}{16F_{\max}}\|P_k - P^*(W_k)\|^2$$

$$\leq \mathbb{V}_k - \mathbb{E}_{\xi_k}[\mathbb{V}_{k+1}] + 3\alpha^2 LF_{\max}^2\sigma^2 + 3\alpha\sigma^2\left\|\frac{G(P_k)}{\sqrt{F(P_k)}}\right\|_\infty^2.$$

Define the convergence metric $E_K^{\text{RL}}$ as

$$E_K^{\text{RL}} := \frac{1}{K}\sum_{k=0}^{K-1}\mathbb{E}\left[\|\nabla f(\bar{W}_k)\odot F(P_k) + |\nabla f(\bar{W}_k)|\odot G(P_k)\|^2 + \frac{1}{M_{\min}^{\text{RL}}}\|\nabla f(\bar{W}_k)\|^2\right. \tag{91}$$

$$+ \frac{\mu^2 \gamma^3 \min\{M(W_k)\}}{64 F_{\max}^4 M_{\min}^{\texttt{RL}}} \|P^*(W_k) \odot F(W_k) - |P^*(W_k)| \odot G(W_k)\|^2$$

$$+ \frac{5\mu^2 \gamma^3}{64 F_{\max}^4 M_{\min}^{\texttt{RL}}} \|W_k - W^*\|_{M(W_k)}^2 + \frac{\mu^2 \gamma^2 M_{\min}^{\texttt{RL}}}{2} \|P_k - P^*(W_k)\|^2 \bigg].$$

Taking expectation over all $\xi_K, \xi_{K-1}, \cdots, \xi_0$, averaging (90) over $k$ from 0 to $K - 1$, and choosing the parameter $\alpha$ as $\alpha = O\left(\frac{1}{F_{\max}}\sqrt{\frac{\mathbb{V}_0}{\sigma^2 LK}}\right)$ deduce that

$$E_K^{\texttt{RL}} \le 8F_{\max}\left(\frac{\mathbb{V}_0 - \mathbb{E}[\mathbb{V}_{k+1}]}{\alpha K} + 3\alpha L F_{\max}^2 \sigma^2\right) + 24 F_{\max}\sigma^2 \times \frac{1}{K}\sum_{k=0}^{K-1}\left\|\frac{G(P_k)}{\sqrt{F(P_k)}}\right\|_\infty^2 \quad (92)$$

$$\le 8F_{\max}\left(\frac{\mathbb{V}_0}{\alpha K} + 3\alpha L F_{\max}^2 \sigma^2\right) + 24 F_{\max}\sigma^2 \times \frac{1}{K}\sum_{k=0}^{K-1}\left\|\frac{G(P_k)}{\sqrt{F(P_k)}}\right\|_\infty^2$$

$$= O\left(F_{\max}^2\sqrt{\frac{\mathbb{V}_0 \sigma^2 L}{K}}\right) + 24 F_{\max}\sigma^2 S_K^{\texttt{RL}}.$$

The strong convexity of the objective (Assumption 3) implies that

$$\mathbb{V}_0 = f(\bar{W}_0) - f^* + C\|W_0 - W^*\|^2 \le \left(1 + \frac{2C}{\mu}\right)(f(W_0) - f^*). \quad (93)$$

Plugging it back to the above inequality, we have

$$E_K^{\texttt{RL}} = O\left(F_{\max}^2\sqrt{\frac{(f(W_0) - f^*)\sigma^2 L}{K}}\right) + 24 F_{\max}\sigma^2 S_K^{\texttt{RL}}. \quad (94)$$

The proof is completed. □

## I.2 Proof of Lemma 5: Descent of sequence $\bar{W}_k$

**Lemma 5** (Descent Lemma of $\bar{W}_k$). *Suppose Assumptions 1–2 hold. It holds for* `Residual Learning` *that*

$$\mathbb{E}_{\xi_k}[f(\bar{W}_{k+1})] \le f(\bar{W}_k) - \frac{\alpha}{4F_{\max}}\|\nabla f(\bar{W}_k)\|_{M(P_k)}^2 + 2\alpha\sigma^2\left\|\frac{G(P_k)}{\sqrt{F(P_k)}}\right\|_\infty^2 + 2\alpha^2 L F_{\max}^2 \sigma^2$$

$$(76)$$

$$- \frac{\alpha\gamma}{8F_{\max}}\|\nabla f(\bar{W}_k) \odot F(P_k) + |\nabla f(\bar{W}_k)| \odot G(P_k)\|^2$$

$$+ \frac{F_{\max}}{\alpha}\mathbb{E}_{\xi_k}\left[\|W_{k+1} - W_k\|_{M(P_k)^\dagger}^2\right] + \mathbb{E}_{\xi_k}[\|W_{k+1} - W_k\|^2].$$

*Proof of Lemma 5.* The $L$-smooth assumption (Assumption 1) implies that

$$\mathbb{E}_{\xi_k}[f(\bar{W}_{k+1})] \le f(\bar{W}_k) + \mathbb{E}_{\xi_k}[\langle \nabla f(\bar{W}_k), \bar{W}_{k+1} - \bar{W}_k\rangle] + \frac{L}{2}\mathbb{E}_{\xi_k}[\|\bar{W}_{k+1} - \bar{W}_k\|^2] \quad (95)$$

$$= f(\bar{W}_k) + \gamma\underbrace{\mathbb{E}_{\xi_k}[\langle \nabla f(\bar{W}_k), P_{k+1} - P_k\rangle]}_{(a)} + \underbrace{\mathbb{E}_{\xi_k}[\langle \nabla f(\bar{W}_k), W_{k+1} - W_k\rangle]}_{(b)} + \frac{L}{2}\underbrace{\mathbb{E}_{\xi_k}[\|\bar{W}_{k+1} - \bar{W}_k\|^2]}_{(c)}.$$

Next, we will handle the each term in the RHS of (95) separately.

**Bound of the second term (a).** To bound term (a) in the RHS of (95), we leverage the assumption that noise has expectation 0 (Assumption 2)

$$\mathbb{E}_{\xi_k}[\langle \nabla f(\bar{W}_k), P_{k+1} - P_k\rangle] \quad (96)$$

$$= \alpha\mathbb{E}_{\xi_k}\left[\left\langle \nabla f(\bar{W}_k) \odot \sqrt{F(P_k)}, \frac{P_{k+1} - P_k}{\alpha\sqrt{F(P_k)}} + (\nabla f(\bar{W}_k; \xi_k) - \nabla f(\bar{W}_k)) \odot \sqrt{F(P_k)}\right\rangle\right]$$

$$= -\frac{\alpha}{2}\|\nabla f(\bar{W}_k) \odot \sqrt{F(P_k)}\|^2$$

$$- \frac{1}{2\alpha}\mathbb{E}_{\xi_k}\left[\left\|\frac{P_{k+1} - P_k}{\sqrt{F(P_k)}} + \alpha(\nabla f(\bar{W}_k;\xi_k) - \nabla f(\bar{W}_k)) \odot \sqrt{F(P_k)}\right\|^2\right]$$

$$+ \frac{1}{2\alpha}\mathbb{E}_{\xi_k}\left[\left\|\frac{P_{k+1} - P_k}{\sqrt{F(P_k)}} + \alpha\nabla f(\bar{W}_k;\xi_k) \odot \sqrt{F(P_k)}\right\|^2\right].$$

The second term in the RHS of (96) can be bounded by

$$\frac{1}{2\alpha}\mathbb{E}_{\xi_k}\left[\left\|\frac{P_{k+1} - P_k}{\sqrt{F(P_k)}} + \alpha(\nabla f(\bar{W}_k;\xi_k) - \nabla f(\bar{W}_k)) \odot \sqrt{F(P_k)}\right\|^2\right] \tag{97}$$

$$= \frac{1}{2\alpha}\mathbb{E}_{\xi_k}\left[\left\|\frac{P_{k+1} - P_k + \alpha(\nabla f(\bar{W}_k;\xi_k) - \nabla f(\bar{W}_k)) \odot F(P_k)}{\sqrt{F(P_k)}}\right\|^2\right]$$

$$\geq \frac{1}{2\alpha F_{\max}}\mathbb{E}_{\xi_k}\left[\|P_{k+1} - P_k + \alpha(\nabla f(\bar{W}_k;\xi_k) - \nabla f(\bar{W}_k)) \odot F(P_k)\|^2\right].$$

The third term in the RHS of (96) can be bounded by variance decomposition and bounded variance assumption (Assumption 2)

$$\frac{1}{2\alpha}\mathbb{E}_{\xi_k}\left[\left\|\frac{P_{k+1} - P_k}{\sqrt{F(P_k)}} + \alpha\nabla f(\bar{W}_k;\xi_k) \odot \sqrt{F(P_k)}\right\|^2\right] \tag{98}$$

$$\leq \frac{\alpha}{2}\mathbb{E}_{\xi_k}\left[\left\||\nabla f(\bar{W}_k;\xi_k)| \odot \frac{G(P_k)}{\sqrt{F(P_k)}}\right\|^2\right]$$

$$\leq \frac{\alpha}{2}\left\||\nabla f(\bar{W}_k)| \odot \frac{G(P_k)}{\sqrt{F(P_k)}}\right\|^2 + \frac{\alpha\sigma^2}{2}\left\|\frac{G(P_k)}{\sqrt{F(P_k)}}\right\|_\infty^2.$$

Notice that the first term in the RHS of (96) and the second term in the RHS of (98) can be bounded together

$$-\frac{\alpha}{2}\|\nabla f(\bar{W}_k) \odot \sqrt{F(P_k)}\|^2 + \frac{\alpha}{2}\left\||\nabla f(\bar{W}_k)| \odot \frac{G(P_k)}{\sqrt{F(P_k)}}\right\|^2 \tag{99}$$

$$= -\frac{\alpha}{2}\sum_{d\in[D]}\left([\nabla f(\bar{W}_k)]_d^2\left([F(P_k)]_d - \frac{[G(P_k)]_d^2}{[F(P_k)]_d}\right)\right)$$

$$= -\frac{\alpha}{2}\sum_{d\in[D]}\left([\nabla f(\bar{W}_k)]_d^2\left(\frac{[F(P_k)]_d^2 - [G(P_k)]_d^2}{[F(P_k)]_d}\right)\right)$$

$$\leq -\frac{\alpha}{2F_{\max}}\sum_{d\in[D]}\left([\nabla f(\bar{W}_k)]_d^2\left([F(P_k)]_d^2 - [G(P_k)]_d^2\right)\right)$$

$$= -\frac{\alpha}{2F_{\max}}\|\nabla f(\bar{W}_k)\|_{M(P_k)}^2 \leq 0.$$

Plugging (97) to (99) into (96), we bound the term (a) by

$$\mathbb{E}_{\xi_k}[\langle\nabla f(\bar{W}_k), P_{k+1} - P_k\rangle] \tag{100}$$

$$\leq -\frac{\alpha}{2F_{\max}}\|\nabla f(\bar{W}_k)\|_{M(P_k)}^2 + \frac{\alpha\sigma^2}{2}\left\|\frac{G(P_k)}{\sqrt{F(P_k)}}\right\|_\infty^2$$

$$- \frac{1}{2\alpha F_{\max}}\mathbb{E}_{\xi_k}\left[\|P_{k+1} - P_k + \alpha(\nabla f(\bar{W}_k;\xi_k) - \nabla f(\bar{W}_k)) \odot F(P_k)\|^2\right].$$

**Bound of the third term (b).** By Young's inequality, we have

$$\mathbb{E}_{\xi_k}[\langle \nabla f(\bar{W}_k), W_{k+1} - W_k \rangle] \leq \frac{\alpha}{4F_{\max}} \|\nabla f(\bar{W}_k)\|_{M(P_k)}^2 + \frac{F_{\max}}{\alpha} \mathbb{E}_{\xi_k}[\|W_{k+1} - W_k\|_{M(P_k)^\dagger}^2]. \tag{101}$$

**Bound of the third term (c).** Repeatedly applying inequality $\|U + V\|^2 \leq 2\|U\|^2 + 2\|V\|^2$ for any $U, V \in \mathbb{R}^D$, we have

$$\frac{L}{2} \mathbb{E}_{\xi_k}[\|\bar{W}_{k+1} - \bar{W}_k\|^2] \tag{102}$$

$$\leq L\mathbb{E}_{\xi_k}[\|W_{k+1} - W_k\|^2] + L\mathbb{E}_{\xi_k}[\|P_{k+1} - P_k\|^2]$$

$$\leq L\mathbb{E}_{\xi_k}[\|W_{k+1} - W_k\|^2] + 2L\mathbb{E}_{\xi_k}\left[\left\|P_{k+1} - P_k + \alpha(\nabla f(\bar{W}_k; \xi_k) - \nabla f(\bar{W}_k)) \odot F(P_k)\right\|^2\right]$$

$$+ 2\alpha^2 L\mathbb{E}_{\xi_k}\left[\left\|(\nabla f(\bar{W}_k; \xi_k) - \nabla f(\bar{W}_k)) \odot F(P_k)\right\|^2\right]$$

$$\leq \mathbb{E}_{\xi_k}[\|W_{k+1} - W_k\|^2] + 2L\mathbb{E}_{\xi_k}\left[\left\|P_{k+1} - P_k + \alpha(\nabla f(\bar{W}_k; \xi_k) - \nabla f(\bar{W}_k)) \odot F(P_k)\right\|^2\right]$$

$$+ 2\alpha^2 LF_{\max}^2 \sigma^2$$

where the last inequality comes from the bounded variance assumption (Assumption 2)

$$2\alpha^2 L\mathbb{E}_{\xi_k}\left[\left\|(\nabla f(\bar{W}_k; \xi_k) - \nabla f(\bar{W}_k)) \odot F(P_k)\right\|^2\right] \tag{103}$$

$$\leq 2\alpha^2 LF_{\max}^2 \mathbb{E}_{\xi_k}\left[\left\|\nabla f(\bar{W}_k; \xi_k) - \nabla f(\bar{W}_k)\right\|^2\right]$$

$$\leq 2\alpha^2 LF_{\max}^2 \sigma^2.$$

**Combination of the upper bound** $(a)$**,** $(b)$**, and** $(c)$**.** Plugging (100), (101), (102) into (95), we derive

$$\mathbb{E}_{\xi_k}[f(\bar{W}_{k+1})] \leq f(\bar{W}_k) - \frac{\alpha}{4F_{\max}} \|\nabla f(\bar{W}_k)\|_{M(P_k)}^2 + \frac{\alpha\sigma^2}{2} \left\|\frac{G(P_k)}{\sqrt{F(P_k)}}\right\|_\infty^2 \tag{104}$$

$$- \left(\frac{1}{2\alpha F_{\max}} - 2L\right) \mathbb{E}_{\xi_k}\left[\left\|P_{k+1} - P_k + \alpha(\nabla f(\bar{W}_k; \xi_k) - \nabla f(\bar{W}_k)) \odot F(P_k)\right\|^2\right]$$

$$+ \frac{F_{\max}}{\alpha} \mathbb{E}_{\xi_k}\left[\|W_{k+1} - W_k\|_{M(P_k)^\dagger}^2\right] + \mathbb{E}_{\xi_k}[\|W_{k+1} - W_k\|^2] + 2\alpha^2 LF_{\max}^2 \sigma^2.$$

We bound the fourth term in the RHS of (104) using the similar technique as in (70)

$$\mathbb{E}_{\xi_k}\left[\left\|P_{k+1} - P_k + \alpha(\nabla f(\bar{W}_k; \xi_k) - \nabla f(\bar{W}_k)) \odot F(P_k)\right\|^2\right] \tag{105}$$

$$\geq \frac{\alpha^2}{2} \|\nabla f(\bar{W}_k) \odot F(P_k) + |\nabla f(\bar{W}_k)| \odot G(P_k)\|^2 - \alpha^2 F_{\max}\sigma^2 \left\|\frac{G(P_k)}{\sqrt{F(P_k)}}\right\|_\infty^2.$$

Inequality (105) as well as the learning rate rule $\alpha \leq \frac{1}{4LF_{\max}}$ leads to the conclusion

$$\mathbb{E}_{\xi_k}[f(\bar{W}_{k+1})] \leq f(\bar{W}_k) - \frac{\alpha}{4F_{\max}} \|\nabla f(\bar{W}_k)\|_{M(P_k)}^2 + 2\alpha\sigma^2 \left\|\frac{G(P_k)}{\sqrt{F(P_k)}}\right\|_\infty^2 + 2\alpha^2 LF_{\max}^2 \sigma^2 \tag{106}$$

$$- \frac{\alpha\gamma}{8F_{\max}} \|\nabla f(\bar{W}_k) \odot F(P_k) + |\nabla f(\bar{W}_k)| \odot G(P_k)\|^2$$

$$+ \frac{F_{\max}}{\alpha} \mathbb{E}_{\xi_k}\left[\|W_{k+1} - W_k\|_{M(P_k)^\dagger}^2\right] + \mathbb{E}_{\xi_k}[\|W_{k+1} - W_k\|^2].$$

The proof is completed. $\qquad \square$

### I.3 Proof of Lemma 6: Descent of sequence $W_k$

**Lemma 6** (Descent Lemma of $W_k$). *It holds for $\mathtt{Residual\ Learning}$ that*

$$\|W_{k+1} - W^*\|^2 \leq \|W_k - W^*\|^2 - \frac{\beta}{2\gamma F_{\max}} \|W_k - W^*\|^2_{M(W_k)} \tag{77}$$

$$- \frac{\beta\gamma}{2F_{\max}} \|P^*(W_k) \odot F(W_k) - |P^*(W_k)| \odot G(W_k)\|^2$$

$$+ \frac{2\beta F_{\max}^3}{\gamma} \|P_{k+1} - P^*(W_k)\|^2_{M(W_k)^\dagger} + 2\beta^2 \|P_{k+1} - P^*(W_k)\|^2.$$

*Proof of Lemma 6.* The proof begins from manipulating the norm $\|W_{k+1} - W^*\|^2$

$$\|W_{k+1} - W^*\|^2 = \|W_k - W^*\|^2 + 2\langle W_k - W^*, W_{k+1} - W_k\rangle + \|W_{k+1} - W_k\|^2. \tag{107}$$

Revisit that we interpret $P_k$ as the residual of $W_k$, namely $P^*(W) := \frac{W^* - W}{\gamma}$. Therefore, we bound the second term in the RHS of (107) by

$$2\langle W_k - W^*, W_{k+1} - W_k\rangle \tag{108}$$
$$= 2\langle W_k - W^*, \beta P_{k+1} \odot F(W_k) - \beta|P_{k+1}| \odot G(W_k)\rangle$$
$$= 2\beta\langle W_k - W^*, P^*(W_k) \odot F(W_k) - |P^*(W_k)| \odot G(W_k)\rangle$$
$$+ 2\beta\langle W_k - W^*, P_{k+1} \odot F(W_k) - |P_{k+1}| \odot G(W_k) - (P^*(W_k) \odot F(W_k) - |P^*(W_k)| \odot G(W_k))\rangle.$$

The first term in the RHS of (108) is bounded by

$$2\beta\langle W_k - W^*, P^*(W_k) \odot F(W_k) - |P^*(W_k)| \odot G(W_k)\rangle \tag{109}$$

$$= 2\beta\left\langle (W_k - W^*) \odot \sqrt{F(W_k)}, \frac{P^*(W_k) \odot F(W_k) - |P^*(W_k)| \odot G(W_k)}{\sqrt{F(W_k)}}\right\rangle$$

$$= -\frac{2\beta}{\gamma}\left\langle (W_k - W^*) \odot \sqrt{F(W_k)}, (W_k - W^*) \odot \sqrt{F(W_k)}\right\rangle$$

$$+ \frac{2\beta}{\gamma}\left\langle (W_k - W^*) \odot \sqrt{F(W_k)}, |W_k - W^*| \odot \frac{G(W_k)}{\sqrt{F(W_k)}}\right\rangle$$

$$\overset{(a)}{=} -\frac{\beta}{\gamma}\|(W_k - W^*) \odot \sqrt{F(W_k)}\|^2 + \frac{\beta}{\gamma}\left\||W_k - W^*| \odot \frac{G(W_k)}{\sqrt{F(W_k)}}\right\|^2$$

$$- \frac{\beta}{\gamma}\left\|(W_k - W^*) \odot \sqrt{F(W_k)} + |W_k - W^*| \odot \frac{G(W_k)}{\sqrt{F(W_k)}}\right\|^2$$

$$\overset{(b)}{\leq} -\frac{\beta}{\gamma F_{\max}}\|W_k - W^*\|^2_{M(W_k)} - \frac{\beta}{\gamma}\left\|(W_k - W^*) \odot \sqrt{F(W_k)} + |W_k - W^*| \odot \frac{G(W_k)}{\sqrt{F(W_k)}}\right\|^2$$

$$\overset{(c)}{\leq} -\frac{\beta}{\gamma F_{\max}}\|W_k - W^*\|^2_{M(W_k)} - \frac{\beta\gamma}{F_{\max}}\|P^*(W_k) \odot F(W_k) - |P^*(W_k)| \odot G(W_k)\|^2$$

where $(a)$ leverages the equality $2\langle U, V\rangle = \|U\|^2 - \|V\|^2 - \|U - V\|^2$ for any $U, V \in \mathbb{R}^D$, $(b)$ is achieved by similar technique (66), and $(c)$ comes from

$$-\frac{\beta}{\gamma}\left\|(W_k - W^*) \odot \sqrt{F(W_k)} + |W_k - W^*| \odot \frac{G(W_k)}{\sqrt{F(W_k)}}\right\|^2 \tag{110}$$

$$= -\beta\gamma\left\|\frac{1}{\sqrt{F(W_k)}} \odot \left(\frac{W_k - W^*}{\gamma} \odot F(W_k) + \left|\frac{W_k - W^*}{\gamma}\right| \odot G(W_k)\right)\right\|^2$$

$$\leq -\frac{\beta\gamma}{F_{\max}}\|P^*(W_k) \odot F(W_k) - |P^*(W_k)| \odot G(W_k)\|^2.$$

The second term in the RHS of (108) is bounded by the Lipschitz continuity of analog update (c.f. Lemma 3)

$$
\frac{2\beta}{\gamma} \langle W_k - W^*, P_{k+1} \odot F(W_k) - |P_{k+1}| \odot G(W_k) - (P^*(W_k) \odot F(W_k) - |P^*(W_k)| \odot G(W_k)) \rangle
$$

$$
\leq \frac{\beta}{2\gamma F_{\max}} \|W_k - W^*\|^2_{M(W_k)} + \frac{2\beta F_{\max}}{\gamma} \tag{111}
$$

$$
\times \|P_{k+1} \odot F(W_k) - |P_{k+1}| \odot G(W_k) - (P^*(W_k) \odot F(W_k) - |P^*(W_k)| \odot G(W_k))\|^2_{M(W_k)^\dagger}
$$

$$
\leq \frac{\beta}{2\gamma F_{\max}} \|W_k - W^*\|^2_{M(W_k)} + \frac{2\beta F_{\max}^3}{\gamma} \|P_{k+1} - P^*(W_k)\|^2_{M(W_k)^\dagger}.
$$

Substituting (109) and (111) into (108), we bound the second term in the RHS of (107) by

$$
2\langle W_k - W^*, W_{k+1} - W_k \rangle \tag{112}
$$

$$
\leq -\frac{\beta}{\gamma F_{\max}} \|W_k - W^*\|^2_{M(W_k)} - \frac{\beta\gamma}{F_{\max}} \|P^*(W_k) \odot F(W_k) - |P^*(W_k)| \odot G(W_k)\|^2
$$

$$
+ \frac{2\beta F_{\max}^3}{\gamma} \|P_{k+1} - P^*(W_k)\|^2_{M(W_k)^\dagger}.
$$

The third term in the RHS of (107) is bounded by the Lipschitz continuity of analog update (c.f. Lemma 3)

$$
\|W_{k+1} - W_k\|^2 = \beta^2 \|P_{k+1} \odot F(W_k) - |P_{k+1}| \odot G(W_k)\|^2 \tag{113}
$$

$$
\leq 2\beta^2 \|P^*(W_k) \odot F(W_k) - |P^*(W_k)| \odot G(W_k)\|^2
$$

$$
+ 2\beta^2 \|P_{k+1} \odot F(W_k) - |P_{k+1}| \odot G(W_k) - (P^*(W_k) \odot F(W_k) - |P^*(W_k)| \odot G(W_k))\|^2
$$

$$
\leq 2\beta^2 \|P^*(W_k) \odot F(W_k) - |P^*(W_k)| \odot G(W_k)\|^2 + 2\beta^2 \|P_{k+1} - P^*(W_k)\|^2.
$$

Plugging (112) and (113) into (107) yields

$$
\|W_{k+1} - W^*\|^2 \leq \|W_k - W^*\|^2 - \frac{\beta}{2\gamma F_{\max}} \|W_k - W^*\|^2_{M(W_k)} \tag{114}
$$

$$
- \left( \frac{\beta\gamma}{F_{\max}} - 2\beta^2 \right) \|P^*(W_k) \odot F(W_k) - |P^*(W_k)| \odot G(W_k)\|^2
$$

$$
+ \frac{2\beta F_{\max}^3}{\gamma} \|P_{k+1} - P^*(W_k)\|^2_{M(W_k)^\dagger} + 2\beta^2 \|P_{k+1} - P^*(W_k)\|^2.
$$

Notice the learning rate $\beta$ is chosen as $\beta \leq \frac{\gamma}{2F_{\max}}$, we have

$$
\|W_{k+1} - W^*\|^2 \leq \|W_k - W^*\|^2 - \frac{\beta}{2\gamma F_{\max}} \|W_k - W^*\|^2_{M(W_k)} \tag{115}
$$

$$
- \frac{\beta\gamma}{2F_{\max}} \|P^*(W_k) \odot F(W_k) - |P^*(W_k)| \odot G(W_k)\|^2
$$

$$
+ \frac{2\beta F_{\max}^3}{\gamma} \|P_{k+1} - P^*(W_k)\|^2_{M(W_k)^\dagger} + 2\beta^2 \|P_{k+1} - P^*(W_k)\|^2
$$

which completes the proof. □

### I.4 Proof of Lemma 7: Descent of sequence $P_k$

**Lemma 7** (Descent Lemma of $P_k$). *Suppose Assumptions 1-2 and 3 hold. It holds for `Tiki-Taka` that*

$$
\mathbb{E}_{\xi_k}[\|P_{k+1} - P^*(W_k)\|^2] \tag{85}
$$

$$
\leq \left( 1 - \frac{\alpha\gamma\mu L}{4(\mu + L)} \right) \|P_k - P^*(W_k)\|^2 + \frac{2\alpha(\mu + L)F_{\max}\sigma^2}{\gamma\mu L} \left\| \frac{G(P_k)}{\sqrt{F(P_k)}} \right\|^2_\infty + \alpha^2 F_{\max}^2 \sigma^2.
$$

*Proof of Lemma 7.* The proof begins from manipulating the norm $\|P_{k+1} - P^*(W_k)\|^2$

$$\|P_{k+1} - P^*(W_k)\|^2 = \|P_k - P^*(W_k)\|^2 + 2\langle P_k - P^*(W_k), P_{k+1} - P_k\rangle + \|P_{k+1} - P_k\|^2. \tag{116}$$

To bound the second term, we need the following equality.

$$
\begin{aligned}
&2\mathbb{E}_{\xi_k}[\langle P_k - P^*(W_k), P_{k+1} - P_k\rangle] \tag{117}\\
&= -2\alpha\mathbb{E}_{\xi_k}[\langle P_k - P^*(W_k), \nabla f(\bar{W}_k;\xi_k) \odot F(P_k) - |\nabla f(\bar{W}_k;\xi_k)| \odot G(P_k)\rangle]\\
&= -2\alpha\mathbb{E}_{\xi_k}[\langle P_k - P^*(W_k), \nabla f(\bar{W}_k;\xi_k) \odot F(P_k)\rangle]\\
&\quad + 2\alpha\mathbb{E}_{\xi_k}[\langle P_k - P^*(W_k), |\nabla f(\bar{W}_k;\xi_k)| \odot G(P_k)\rangle]\\
&= -2\alpha\langle P_k - P^*(W_k), \nabla f(\bar{W}_k) \odot F(P_k)\rangle + 2\alpha\langle P_k - P^*(W_k), |\nabla f(\bar{W}_k)| \odot G(P_k)\rangle\\
&\quad + 2\alpha\mathbb{E}_{\xi_k}[\langle P_k - P^*(W_k), (|\nabla f(\bar{W}_k)| - |\nabla f(\bar{W}_k;\xi_k)|) \odot G(P_k)\rangle]\\
&= -2\alpha\underbrace{\langle P_k - P^*(W_k), \nabla f(\bar{W}_k) \odot F(P_k) - |\nabla f(\bar{W}_k)| \odot G(P_k)\rangle}_{(T1)}\\
&\quad + 2\alpha\underbrace{\mathbb{E}_{\xi_k}[\langle P_k - P^*(W_k), (|\nabla f(\bar{W}_k)| - |\nabla f(\bar{W}_k;\xi_k)|) \odot G(P_k)\rangle]}_{(T2)}
\end{aligned}
$$

**Upper bound of the first term** $(T1)$. With Lemma 4, the second term in the RHS of (116) can be bounded by

$$
\begin{aligned}
&-2\alpha\langle P_k - P^*(W_k), \nabla f(\bar{W}_k) \odot F(P_k) - |\nabla f(\bar{W}_k)| \odot G(P_k)\rangle \tag{118}\\
&= -2\alpha\langle P_k - P^*(W_k), \nabla f(\bar{W}_k) \odot q_s(P_k)\rangle\\
&\leq -2\alpha C_{k,+}\langle P_k - P^*(W_k), \nabla f(\bar{W}_k)\rangle + 2\alpha C_{k,-}\langle |P_k - P^*(W_k)|, |\nabla f(\bar{W}_k)|\rangle
\end{aligned}
$$

where $C_{k,+}$ and $C_{k,-}$ are defined by

$$C_{k,+} := \frac{1}{2}\left(\max_{d\in[D]}\{q_s([P_k]_d)\} + \min_{d\in[D]}\{q_s([P_k]_d)\}\right), \tag{119}$$

$$C_{k,-} := \frac{1}{2}\left(\max_{d\in[D]}\{q_s([P_k]_d)\} - \min_{d\in[D]}\{q_s([P_k]_d)\}\right). \tag{120}$$

In the inequality above, the first term can be bounded by the strong convexity of $f$. Let $\varphi(P) := f(W + \gamma P)$ which is $\gamma^2 L$-smooth and $\gamma^2 \mu$-strongly convex. It can be verified that $\varphi(P)$ has gradient $\nabla\varphi(P_k) = \nabla_{P_k} f(W_k + \gamma P_k) = \gamma\nabla f(\bar{W}_k)$ and optimal point $P^*(W)$. Leveraging Theorem 2.1.9 in [85], we have

$$\langle\nabla f(\bar{W}_k), P_k - P^*(W_k)\rangle = \frac{1}{\gamma}\langle\nabla\varphi(P_k), P_k - P^*(W_k)\rangle \tag{121}$$

$$\geq \frac{1}{\gamma}\left(\frac{\gamma^2\mu\cdot\gamma^2 L}{\gamma^2\mu + \gamma^2 L}\|P_k - P^*(W_k)\|^2 + \frac{1}{\gamma^2\mu + \gamma^2 L}\|\nabla\varphi(P_k)\|^2\right)$$

$$= \frac{\gamma\mu L}{\mu + L}\|P_k - P^*(W_k)\|^2 + \frac{1}{\gamma(\mu + L)}\|\nabla f(\bar{W}_k)\|^2.$$

The second term in the RHS of (118) can be bounded by Young's inequality $2\langle x,y\rangle \leq u\|x\|^2 + \frac{1}{u}\|y\|^2$ with any $u > 0$ and $x, y \in \mathbb{R}^D$

$$2\alpha C_{k,-}\langle |P_k - P^*(W_k)|, |\nabla f(\bar{W}_k)|\rangle \tag{122}$$

$$\leq \frac{\alpha C_{k,-}^2\gamma(\mu + L)}{C_{k,+}}\|P_k - P^*(W_k)\|^2 + \frac{\alpha C_{k,+}}{\gamma(\mu + L)}\|\nabla f(\bar{W}_k)\|^2$$

where $u$ is chosen to align the coefficient in front of $\|\nabla f(\bar{W}_k)\|^2$. Therefore, $(T1)$ in (118) becomes

$$-2\alpha\langle P_k - P^*(W_k), \nabla f(\bar{W}_k) \odot F(P_k) - |\nabla f(\bar{W}_k)| \odot G(P_k)\rangle \tag{123}$$

$$\leq -\left(\frac{2\alpha\gamma\mu LC_{k,+}}{\mu+L} - \frac{\alpha C_{k,-}^2\gamma(\mu+L)}{C_{k,+}}\right)\|P_k - P^*(W_k)\|^2 - \frac{\alpha C_{k,+}}{\gamma(\mu+L)}\|\nabla f(\bar{W}_k)\|^2.$$

**Upper bound of the second term** $(T2)$. Leveraging the Young's inequality $2\langle x,y\rangle \leq u\|x\|^2 + \frac{1}{u}\|y\|^2$ with any $u > 0$ and $x, y \in \mathbb{R}^D$, we have

$$2\alpha\mathbb{E}_{\xi_k}[\langle P_k - P^*(W_k), (|\nabla f(\bar{W}_k)| - |\nabla f(\bar{W}_k;\xi_k)|) \odot G(P_k)\rangle] \tag{124}$$

$$= 2\alpha\mathbb{E}_{\xi_k}\left[\left\langle (P_k - P^*(W_k)) \odot \sqrt{F(P_k)}, (|\nabla f(\bar{W}_k)| - |\nabla f(\bar{W}_k;\xi_k)|) \odot \frac{G(P_k)}{\sqrt{F(P_k)}}\right\rangle\right]$$

$$\overset{(a)}{\leq} \frac{\alpha\gamma\mu LC_{k,+}}{(\mu+L)F_{\max}}\|(P_k - P^*(W_k)) \odot \sqrt{F(P_k)}\|^2$$

$$+ \frac{\alpha(\mu+L)F_{\max}}{\gamma\mu LC_{k,+}}\mathbb{E}_{\xi_k}\left[\left\|(|\nabla f(\bar{W}_k)| - |\nabla f(\bar{W}_k;\xi_k)|) \odot \frac{G(P_k)}{\sqrt{F(P_k)}}\right\|^2\right]$$

$$\overset{(b)}{\leq} \frac{\alpha\gamma\mu LC_{k,+}}{(\mu+L)F_{\max}}\|(P_k - P^*(W_k)) \odot \sqrt{F(P_k)}\|^2$$

$$+ \frac{\alpha(\mu+L)F_{\max}}{\gamma\mu LC_{k,+}}\mathbb{E}_{\xi_k}\left[\left\|(|\nabla f(\bar{W}_k) - \nabla f(\bar{W}_k;\xi_k)|) \odot \frac{G(P_k)}{\sqrt{F(P_k)}}\right\|^2\right]$$

$$\overset{(c)}{=} \frac{\alpha\gamma\mu LC_{k,+}}{(\mu+L)F_{\max}}\|(P_k - P^*(W_k)) \odot \sqrt{F(P_k)}\|^2 + \frac{\alpha(\mu+L)F_{\max}\sigma^2}{\gamma\mu LC_{k,+}}\left\|\frac{G(P_k)}{\sqrt{F(P_k)}}\right\|_\infty^2$$

$$\overset{(d)}{\leq} \frac{\alpha\gamma\mu LC_{k,+}}{\mu+L}\|P_k - P^*(W_k)\|^2 + \frac{\alpha(\mu+L)F_{\max}\sigma^2}{\gamma\mu LC_{k,+}}\left\|\frac{G(P_k)}{\sqrt{F(P_k)}}\right\|_\infty^2$$

where $(a)$ choose $u > 0$ to align the coefficient in front of $\|P_k - P^*(W_k)\|^2$ in the RHS of (123), $(b)$ applies $||x| - |y|| \leq |x - y|$ for any $x, y \in \mathbb{R}$, $(c)$ uses the bounded variance assumption (c.f. Assumption 2), and $(d)$ leverages the fact that $F(P_k)$ is bounded by $F_{\max}$ element-wise.

Combining the upper bound of $(T1)$ and $(T2)$, we bound (117) by

$$2\mathbb{E}_{\xi_k}[\langle P_k - P^*(W_k), P_{k+1} - P_k\rangle] \tag{125}$$

$$\leq -\left(\frac{\alpha\gamma\mu LC_{k,+}}{\mu+L} - \frac{\alpha C_{k,-}^2\gamma(\mu+L)}{C_{k,+}}\right)\|P_k - P^*(W_k)\|^2$$

$$- \frac{\alpha C_{k,+}}{\gamma(\mu+L)}\|\nabla f(\bar{W}_k)\|^2 + \frac{\alpha(\mu+L)F_{\max}\sigma^2}{\gamma\mu LC_{k,+}}\left\|\frac{G(P_k)}{\sqrt{F(P_k)}}\right\|_\infty^2$$

$$\leq -\frac{\alpha\gamma\mu LC_{k,+}}{2(\mu+L)}\|P_k - P^*(W_k)\|^2 - \frac{\alpha C_{k,+}}{\gamma(\mu+L)}\|\nabla f(\bar{W}_k)\|^2 + \frac{\alpha(\mu+L)F_{\max}\sigma^2}{\gamma\mu LC_{k,+}}\left\|\frac{G(P_k)}{\sqrt{F(P_k)}}\right\|_\infty^2$$

where the last inequality holds when $\gamma$ is sufficiently large, $P_k$ as well as $C_{k,-}$ are sufficiently closed to 0, and the following inequality holds

$$(\mu+L)\frac{C_{k,-}^2}{C_{k,+}^2} \leq \frac{\mu L}{2(\mu+L)}. \tag{126}$$

Furthermore, the last term in the RHS of (116) can be bounded by the Lipschitz continuity of analog update (c.f. Lemma 3) and the bounded variance assumption (c.f. Assumption 2)

$$\mathbb{E}_{\xi_k}[\|P_{k+1} - P_k\|^2] = \mathbb{E}_{\xi_k}[\|\alpha\nabla f(\bar{W}_k;\xi_k) \odot F(P_k) - \alpha|\nabla f(\bar{W}_k;\xi_k)| \odot G(P_k)\|^2] \tag{127}$$

$$\leq \alpha^2 F_{\max}^2 \mathbb{E}_{\xi_k}[\|\nabla f(\bar{W}_k;\xi_k)\|^2]$$

$$= \alpha^2 F_{\max}^2\|\nabla f(\bar{W}_k)\|^2 + \alpha^2 F_{\max}^2\sigma^2$$

$$\leq \frac{\alpha C_{k,+}}{\gamma(\mu + L)} \|\nabla f(\bar{W}_k)\|^2 + \alpha^2 F_{\max}^2 \sigma^2$$

where the last inequality holds if $\alpha$ is sufficiently small.

Plugging inequality (125) and (127) above into (116) yields

$$\mathbb{E}_{\xi_k}[\|P_{k+1} - P^*(W_k)\|^2] \tag{128}$$

$$\leq \left(1 - \frac{\alpha\gamma\mu L C_{k,+}}{2(\mu + L)}\right) \|P_k - P^*(W_k)\|^2 + \frac{\alpha(\mu + L)F_{\max}^2 \sigma^2}{\gamma\mu L C_{k,+}} \left\|\frac{G(P_k)}{\sqrt{F(P_k)}}\right\|_\infty^2 + \alpha^2 F_{\max}^2 \sigma^2.$$

By definition of $C_{k,+}$, when the saturation degree of $P_k$ is properly limited, we have $C_{k,+} \geq \frac{1}{2}$. Therefore, we have

$$\mathbb{E}_{\xi_k}[\|P_{k+1} - P^*(W_k)\|^2] \tag{129}$$

$$\leq \left(1 - \frac{\alpha\gamma\mu L}{4(\mu + L)}\right) \|P_k - P^*(W_k)\|^2 + \frac{2\alpha(\mu + L)F_{\max}^2 \sigma^2}{\gamma\mu L} \left\|\frac{G(P_k)}{\sqrt{F(P_k)}}\right\|_\infty^2 + \alpha^2 F_{\max}^2 \sigma^2$$

which completes the proof. $\qquad\square$

## I.5  Proof of Corollary 1: Exact convergence of Residual Learning

**Corollary 1** (Exact convergence of `Residual Learning`). *Under Assumption 4 and the conditions in Theorem 3, if $\gamma \geq \Omega(q_{\min}^{-2/5})$, it holds that $E_K^{\mathtt{RL}} \leq O\left(\sqrt{\sigma^2 L / K}\right)$.*

*Proof of Corollary 1.* From Theorem 3, we have

$$\|\nabla f(\bar{W}_k)\|^2 \leq O(E_K^{\mathtt{RL}}) \leq O\left(F_{\max}^2 \sqrt{\frac{(f(W_0) - f^*)\sigma^2 L}{K}}\right) + 24 F_{\max}\sigma^2 S_K^{\mathtt{RL}}. \tag{130}$$

Under the zero-shift assumption (Assumption 4) and the Lipschitz continuity of the response functions, it holds directly that

$$\left\|\frac{G(P_k)}{\sqrt{F(P_k)}}\right\|_\infty^2 \leq \left\|\frac{G(P_k)}{\sqrt{F(P_k)}}\right\|^2 = \left\|\frac{G(P_k)}{\sqrt{F(P_k)}} - \frac{G(0)}{\sqrt{F(0)}}\right\|^2 \leq L_S^2 \|P_k\|^2 \tag{131}$$

where $L_S \geq 0$ is a constant. Using $\|U + V\|^2 \leq 2\|U\|^2 + 2\|V\|^2$ for any $U, V \in \mathbb{R}^D$, we have

$$\|P_k\|^2 \leq 2\|P_k - P^*(W_k)\|^2 + 2\|P^*(W_k)\|^2 = 2\|P_k - P^*(W_k)\|^2 + \frac{2}{\gamma^2}\|W_k - W^*\|^2 \tag{132}$$

where the last inequality comes from the definition of $P^*(W_k)$, as well as the definition of $P^*(W)$. Recall that convergence metric $E_K^{\mathtt{RL}}$ defined in (91) is in the order of

$$E_K^{\mathtt{RL}} \geq \Omega\left(\gamma^3 \|W_k - W^*\|_{M(W_k)}^2 + \gamma^2 \|P_k - P^*(W_k)\|^2\right) \tag{133}$$

$$\geq \Omega\left(\min\{M(W_k)\}\gamma^3 \|W_k - W^*\|^2 + \gamma^2 \|P_k - P^*(W_k)\|^2\right)$$

$$\geq \Omega\left(\frac{1}{\gamma^2}\|W_k - W^*\|^2 + \gamma^2 \|P_k - P^*(W_k)\|^2\right).$$

Therefore, we have

$$S_K^{\mathtt{RL}} = \frac{1}{K}\sum_{k=0}^{K}\left\|\frac{G(P_k)}{\sqrt{F(P_k)}}\right\|_\infty^2 \leq \frac{1}{K}\sum_{k=0}^{K}\left(2\|P_k - P^*(W_k)\|^2 + \frac{2}{\gamma^2}\|W_k - W^*\|^2\right) \leq O(E_K^{\mathtt{RL}}) \tag{134}$$

where the last inequality holds if $\gamma$ is sufficiently large. Considering that, $E_K^{\mathtt{RL}} - S_K^{\mathtt{RL}} \geq \Omega(E_K^{\mathtt{RL}}) \geq 0$ and the conclusion is reached directly from Theorem 3. $\qquad\square$

# J Proof of Theorem 6: Convergence of Analog GD

In Section 3.2, we showed that `Analog SGD` converges to a critical point inexactly with asymptotic error proportional to the noise variance $\sigma^2$. Intuitively, without the effect of noise, `Analog GD` converges to the critical point. Define the convergence metric by

$$E_K^{\texttt{AGD}} := \frac{1}{K} \sum_{k=0}^{K-1} \left( \left\| \nabla f(W_k) \odot F(W_k) - |\nabla f(W_k)| \odot G(W_k) \right\|^2 + \|\nabla f(W_k)\|_{M(W_k)}^2 \right). \quad (135)$$

The convergence is guaranteed by the following theorem.

**Theorem 6** (Convergence of `Analog GD`). *Under Assumption 1–2, it holds that*

$$E_K^{AGD} \leq \frac{8L(f(W_0) - f^*)F_{\max}^2}{K}. \quad (136)$$

*Further, if $M_{\min}^{ASGD} := \min_{k \in [K]} \min\{Q_+(W_k) \odot Q_-(W_k)\} > 0$, it holds that*

$$\frac{1}{K} \sum_{k=0}^{K-1} \|\nabla f(W_k)\|^2 \leq \frac{2L(f(W_0) - f^*)F_{\max}^2}{K M_{\min}^{ASGD}}. \quad (137)$$

*Proof of Theorem 6.* The $L$-smooth assumption (Assumption 1) implies that

$$f(W_{k+1}) \leq f(W_k) + \langle \nabla f(W_k), W_{k+1} - W_k \rangle + \frac{L}{2}\|W_{k+1} - W_k\|^2 \quad (138)$$

$$= f(W_k) - \frac{\alpha}{2}\|\nabla f(W_k) \odot \sqrt{F(W_k)}\|^2 - \frac{1}{F_{\max}}\left(\frac{1}{2\alpha} - \frac{LF_{\max}}{2}\right)\|W_{k+1} - W_k\|^2$$

$$+ \frac{1}{2\alpha}\left\|\frac{W_{k+1} - W_k}{\sqrt{F(W_k)}} + \alpha\nabla f(W_k) \odot \sqrt{F(W_k)}\right\|^2$$

where the second inequality comes from

$$\langle \nabla f(W_k), W_{k+1} - W_k \rangle = \alpha \left\langle \nabla f(W_k) \odot \sqrt{F(W_k)}, \frac{W_{k+1} - W_k}{\alpha\sqrt{F(W_k)}} \right\rangle \quad (139)$$

$$= -\frac{\alpha}{2}\|\nabla f(W_k) \odot \sqrt{F(W_k)}\|^2 - \frac{1}{2\alpha}\left\|\frac{W_{k+1} - W_k}{\sqrt{F(W_k)}}\right\|^2$$

$$+ \frac{1}{2\alpha}\left\|\frac{W_{k+1} - W_k}{\sqrt{F(W_k)}} + \alpha\nabla f(W_k) \odot \sqrt{F(W_k)}\right\|^2$$

as well as the inequality

$$\left\|\frac{W_{k+1} - W_k}{\sqrt{F(W_k)}}\right\|^2 \geq \frac{1}{F_{\max}}\|W_{k+1} - W_k\|^2. \quad (140)$$

The third term in the RHS of (138) can be bounded by

$$\frac{1}{2\alpha}\left\|\frac{W_{k+1} - W_k}{\sqrt{F(W_k)}} + \alpha\nabla f(W_k) \odot \sqrt{F(W_k)}\right\|^2 = \frac{\alpha}{2}\left\||\nabla f(W_k)| \odot \frac{G(W_k)}{\sqrt{F(W_k)}}\right\|^2. \quad (141)$$

Define the saturation vector $M(W_k) \in \mathbb{R}^D$ by

$$M(W_k) := F(W_k)^{\odot 2} - G(W_k)^{\odot 2} = (F(W_k) + G(W_k)) \odot (F(W_k) - G(W_k)) \quad (142)$$
$$= q_+(W_k) \odot q_-(W_k).$$

Notice the following inequality is valid

$$-\frac{\alpha}{2}\|\nabla f(W_k) \odot \sqrt{F(W_k)}\|^2 + \frac{\alpha}{2}\left\||\nabla f(W_k)| \odot \frac{G(W_k)}{\sqrt{F(W_k)}}\right\|^2 \quad (143)$$

$$= -\frac{\alpha}{2} \sum_{d \in [D]} \left( [\nabla f(W_k)]_d^2 \left( [F(W_k)]_d - \frac{[G(W_k)]_d^2}{[F(W_k)]_d} \right) \right)$$

$$= -\frac{\alpha}{2} \sum_{d \in [D]} \left( [\nabla f(W_k)]_d^2 \left( \frac{[F(W_k)]_d^2 - [G(W_k)]_d^2}{[F(W_k)]_d} \right) \right)$$

$$\leq -\frac{\alpha}{2 F_{\max}} \sum_{d \in [D]} \left( [\nabla f(W_k)]_d^2 \left( [F(W_k)]_d^2 - [G(W_k)]_d^2 \right) \right)$$

$$= -\frac{\alpha}{2 F_{\max}} \|\nabla f(W_k)\|_{S_k}^2 \leq 0.$$

Substituting (141) and (143) back into (138) yields

$$\frac{1}{F_{\max}} \left( \frac{1}{2\alpha} - \frac{L F_{\max}}{2} \right) \|W_{k+1} - W_k\|^2 \leq f(W_k) - f(W_{k+1}). \tag{144}$$

Noticing that $\|W_{k+1} - W_k\|^2 = \alpha^2 \|\nabla f(W_k) \odot F(W_k) - |\nabla f(W_k)| \odot G(W_k)\|^2$ and averaging for $k$ from 0 to $K-1$, we have

$$E_K^{\mathtt{AGD}} = \frac{1}{K} \sum_{k=0}^{K-1} \left( \|\nabla f(W_k) \odot F(W_k) - |\nabla f(W_k)| \odot G(W_k)\|^2 + \|\nabla f(W_k)\|_{M(W_k)}^2 \right) \tag{145}$$

$$\leq \frac{2(f(W_0) - f(W_{K+1})) F_{\max}}{\alpha(1 - \alpha L F_{\max})K} \leq \frac{8L(f(W_0) - f^*) F_{\max}^2}{K}$$

where the last inequality choose $\alpha = \frac{1}{2L F_{\max}}$.

Further, given the response functions are bounded, (138)–(143) implies that there is a lower bound $M_{\min}^{\mathtt{AGD}}$ such that

$$\frac{\alpha M_{\min}^{\mathtt{AGD}}}{2} \|\nabla f(W_k)\|^2 \leq \frac{\alpha}{2} \|\nabla f(W_k)\|_{M(W_k)}^2 \leq f(W_k) - f(W_{k+1}). \tag{146}$$

Averaging (146) for $k$ from 0 to $K$ deduce that

$$\frac{1}{K} \sum_{k=0}^{K-1} \|\nabla f(W_k)\|^2 \leq \frac{2(f(W_0) - f(W_{K+1})) F_{\max}}{\alpha K M_{\min}^{\mathtt{AGD}}} \leq \frac{2L(f(W_0) - f^*) F_{\max}^2}{K M_{\min}^{\mathtt{AGD}}} \tag{147}$$

where the second inequality holds because the learning rate is selected as $\alpha = \frac{1}{L F_{\max}}$. □

## K   Simulation Details and Additional Results

This section provides details about the experiments in Section 6. All simulation is performed under the PYTORCH framework https://github.com/pytorch/pytorch. The analog training algorithms, including `Analog SGD` and `Tiki-Taka`, are provided by the open-source simulation toolkit AIHWKIT [44], which has MIT license; see github.com/IBM/aihwkit.

**Optimizer.** The baseline `Digital SGD` optimizer is implemented by `FloatingPointRPUConfig` in AIHWKIT, which is equivalent to the SGD implemented in PYTORCH. The `Analog SGD` is implemented by selecting `SingleRPUConfig` as configuration, and `Tiki-Taka` optimizers are implemented by `UnitCellRPUConfig` with `TransferCompound` devices in AIHWKIT.

As suggested by [22], in the implementation of `Residual Learning`, only a few columns of $P_k$ are transferred per time to $W_k$ in the recursion (11) to balance the communication and computation. In our simulations, we transfer 1 column every time.

**RPU Configuration.** AIHWKIT offers fine-grained simulations of the hardware imperfections, such as the IO noise, analog-digital conversion, and so on. They are specified by the resistive processing unit (RPU) configurations. Without other specifications, we use the configuration list in Table 4. The experimental setup uses a specific I/O configuration, as detailed in the relevant table. The system's input and output signal bounds are explicitly defined. Regarding signal quality, the setup employs no

input noise but introduces additive Gaussian noise to the output signal, the statistical properties of which are precisely specified. Finally, the resolution of the digital conversion process is determined by distinct bit values for both the input (DAC) and the output (ADC).

In addition, noise, bound, and update management techniques are used [71]. A learnable scaling factor is applied after each analog layer and updated using SGD. For each gradient update step, if more than BL $= 32$ pulses are desired, only BL pulses are fired.

Table 4: Hardware imperfection setting

| configuration | value |
|---|---|
| input bound | 1.0 |
| input noise | None |
| input resolution (DAC) | 7 bits |
| output bound | 12.0 |
| output noise | additive Gaussian noise $\mathcal{N}(0, 0.06^2)$ |
| output resolution (ADC) | 9 bits |
| Update granularity $\Delta w_{\min}$ | $1 \times 10^{-3}$ |
| Bit length BL | 32 |

**Simulation hardware.** We conduct our experiments on one NVIDIA RTX 3090 GPU, which has 24GB of memory and a maximum power of 350W. The simulations take from 30 minutes to 5 hours, depending on model sizes and datasets.

**Statistical Significance.** The simulation data reported in all tables is repeated three times. The randomness originates from the data shuffling, random initialization, and random noise in the analog hardware. The mean and standard deviation are calculated using *statistics* library.

### K.1 Power and Exponential Response Functions

We consider two types of response functions in our simulations: power and exponential response functions with dynamic ranges $[-\tau, \tau]$, The *power response* is a power function, given by

$$q_+(w) = \left(1 - \frac{w}{\tau}\right)^{\gamma_{\text{res}}}, \qquad q_-(w) = \left(1 + \frac{w}{\tau}\right)^{\gamma_{\text{res}}} \tag{148}$$

which can be changed by adjusting the dynamic radius $\tau$ and shape parameter $\gamma_{\text{res}}$. We also consider the *exponential response*, whose response is an exponential function, defined by

$$q_+(w) = \frac{\exp\left(\gamma_{\text{res}}(1 - w/\tau)\right) - 1}{\exp\left(\gamma_{\text{res}}\right) - 1}, \qquad q_-(w) = \frac{\exp\left(\gamma_{\text{res}}(1 + w/\tau)\right) - 1}{\exp\left(\gamma_{\text{res}}\right) - 1}. \tag{149}$$

It could be checked that the boundary of their dynamic ranges are $\tau^{\max} = \tau$ and $\tau^{\min} = -\tau$, while the symmetric point is 0, as required by Corollary 1. Figure 6 illustrates how the response functions change with different $\gamma_{\text{res}}$.

### K.2 Least squares problem

In Figure 2 (see Section 1.1), we consider the least squares problem on a synthetic dataset and a ground truth $W^* \in \mathbb{R}^D$. The problem can be formulated by

$$\min_{W \in \mathbb{R}^D} f(W) := \frac{1}{2}\|AW - b\|^2 = \frac{1}{2}\|A(W - W^*)\|^2. \tag{150}$$

The elements of $W^*$ are sampled from a Gaussian distribution with mean 0 and variance $\sigma_{W^*}^2$. Consider a matrix $A \in \mathbb{R}^{D_{\text{out}} \times D}$ of size $D = 50$ and $D_{\text{out}} = 100$ whose elements are sampled from a Gaussian distribution with variance $\sigma_A^2$. The label $b \in \mathbb{R}^{D_{\text{out}}}$ is generated by $b = AW^*$ where $W^*$ are sampled from a standard Gaussian distribution with $\sigma_{W^*}^2$. The response granularity $\Delta w_{\min}$=1e-4 while $\tau = 3.5$. The maximum bit length is 8. The variance are set as $\sigma_A^2 = 1.00^2$, $\sigma_{W^*}^2 = 0.5^2$.

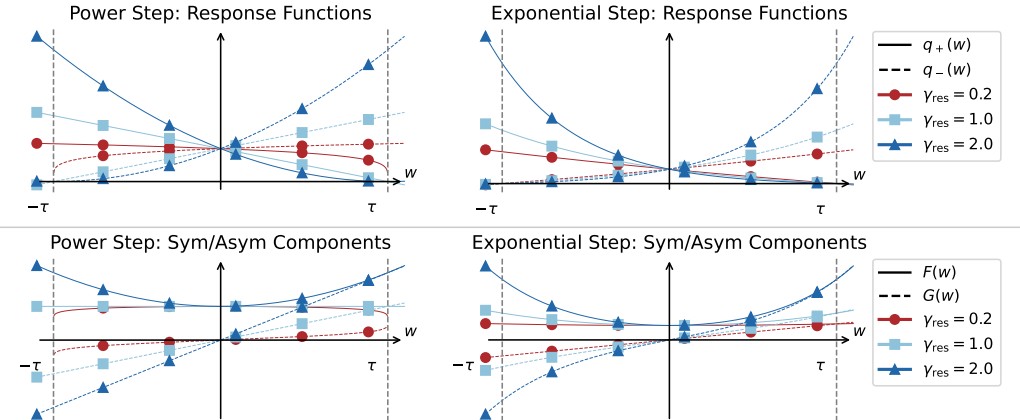

Figure 6: Examples of response functions. The dependence of the response function on the weight $w$ can grow at various rates, including but not limited to power (Left) or exponential rate (Right). $\tau$ is the radius of the dynamic range, and $\gamma_{\mathtt{res}}$ is a parameter that needs to be determined by physical measurements.

### K.3   Classification problem

We conduct training simulations of image classification tasks on a series of real datasets.

**3-FC @ MNIST.** Following the setting in [16], we train a model with 3 fully connected layers. The hidden sizes are 256 and 128. The activation functions are Sigmoid. The learning rates are $\alpha = 0.1$ for `Digital SGD`, $\alpha = 0.05, \beta = 0.01$ for `Analog SGD` and `Tiki-Taka`. The batch size is 10 for all algorithms. In Figure 4, the power response functions with $\gamma_{\mathrm{res}} = 0.5$ are used, and various $\tau$ are used as indicated in the legend.

**CNN @ MNIST.** We train a convolutional neural network, which contains 2 convolutional layers, 2 max-pooling layers, and 2 fully connected layers. The activation functions are Tanh. The first two convolutional layers use 5×5 kernels with 16 and 32 kernels, respectively. Each convolutional layer is followed by a subsampling layer implemented by the max pooling function over non-overlapping pooling windows of size $2 \times 2$. The output of the second pooling layer, consisting of 512 neuron activations, feeds into a fully connected layer consisting of 128 tanh neurons, which is then connected to a 10-way softmax output layer. The learning rates are set as $\alpha = 0.1$ for `Digital SGD`, $\alpha = 0.05, \beta = 0.01$ for `Analog SGD` are `Residual Learning/Tiki-Taka`. The batch size is 8 for all algorithms. In Figure 4, the power response functions with $\gamma_{\mathrm{res}} = 0.5$ are used, and various $\tau$ are used as indicated in the legend.

**ResNet/MobileNet @ CIFAR10/CIFAR100.** We train different models from the ResNet family, including ResNet18, 34, and 50. The base model is pre-trained on the ImageNet dataset. The last fully connected layer is replaced by an analog layer. The learning rates are set as $\alpha = 0.075$ for `Digital SGD`, $\alpha = 0.075, \beta = 0.01$ for `Analog SGD`, `Residual Learning/Tiki-Taka`, `Tiki-Taka v2`, and `Residual Learning v2`. `Tiki-Taka` adopts $\gamma = 0.4$ unless stated otherwise. The batch size is 128 for all algorithms.

### K.4   Additional performance on real datasets

We train different models from the MobileNet family, including MobileNet2, MobileNetV3L, MobileNetV3S. The base model is pre-trained on ImageNet dataset. The last fully connected layer is replaced by an analog layer. The learning rates are set as $\alpha = 0.075$ for `Digital SGD`, $\alpha = 0.075, \beta = 0.01$ for `Analog SGD` or `Tiki-Taka`. `Tiki-Taka` adopts $\gamma = 0.4$ unless stated otherwise. The batch size is 128 for all algorithms. Power response function with $\gamma_{\mathtt{res}} = 4.0$ and $\tau = 0.05$ is used in the simulations.

**ResNet @ CIFAR10/CIFAR100.** We fine-tune three models from the ResNet family with different scales on CIFAR10/CIFAR100 datasets. The power response functions with $\gamma_{\mathrm{res}} = 3.0$ and $\tau = 0.1$,

and the exponential response functions with $\gamma_{\text{res}} = 4.0$ and $\tau = 0.1$ are used, whose results are shown in Table 1 and 5, respectively. The results show that the `Tiki-Taka` outperforms `Analog SGD` by about $1.0\%$ in most of the cases in ResNet34/50, and the gap even reaches about $10.0\%$ for ResNet18 training on the CIFAR100 dataset.

| | CIFAR10 | | | | |
|---|---|---|---|---|---|
| | DSGD | ASGD | TT/RL | TTv2 | RLv2 |
| ResNet18 | 95.43$\pm$0.13 | 84.47$\pm$3.40 | 94.81$\pm$0.09 | 95.31$\pm$0.05 | 95.12$\pm$0.14 |
| ResNet34 | 96.48$\pm$0.02 | 95.43$\pm$0.12 | 96.29$\pm$0.12 | 96.60$\pm$0.05 | 96.42$\pm$0.13 |
| ResNet50 | 96.57$\pm$0.10 | 94.36$\pm$1.16 | 96.34$\pm$0.04 | 96.63$\pm$0.09 | 96.56$\pm$0.08 |
| | CIFAR100 | | | | |
| | DSGD | ASGD | TT/RL | TTv2 | RLv2 |
| ResNet18 | 81.12$\pm$0.25 | 68.98$\pm$1.01 | 76.17$\pm$0.23 | 78.56$\pm$0.29 | 79.83$\pm$0.13 |
| ResNet34 | 83.86$\pm$0.12 | 78.98$\pm$0.55 | 80.58$\pm$0.11 | 81.81$\pm$0.15 | 82.85$\pm$0.19 |
| ResNet50 | 83.98$\pm$0.11 | 79.88$\pm$1.26 | 80.80$\pm$0.22 | 82.82$\pm$0.33 | 83.90$\pm$0.20 |

Table 5: Fine-tuning ResNet models with the *exponential response* on CIFAR10/100 datasets. Test accuracy is reported. `DSGD`, `ASGD`, and `TT` represent `Digital SGD`, `Analog SGD`, `Tiki-Taka`, respectively.

**MobileNet @ CIFAR10/CIFAR100.** We fine-tune three MobileNet models with different scales on CIFAR10/CIFAR100 datasets. The response function is set as the power response with the parameter $\gamma_{\text{res}} = 4.0$ and $\tau = 0.05$, whose results are shown in Table 6. In the simulations, the accuracy of `Analog SGD` drops significantly by about $10\%$ in most cases, while `Tiki-Taka` remains comparable to the `Digital SGD` with only a slight drop.

### K.5 Ablation study on cycle variation

To verify the conclusion of Theorem 4 that the error introduced by cycle variation is a higher-order term, we conduct a numerical simulation training on an image classification task on the MNIST dataset using Fully-connected network (FCN) or convolution neural network (CNN) network. In the pulse update (26), the parameter $\sigma_c$ is varied from 10% to 120%, where the noise signal is already larger than the response function signal itself. The results are shown in Table 7. The results show that the test accuracy of both `Analog SGD` and `Tiki-Taka` is not significantly affected by the cycle variation, which complies with the theoretical analysis.

### K.6 Ablation study on various response functions

We also train a FCN model on the MNIST dataset under various response functions. As shown in the figure, larger $\gamma_{\text{res}}$ leads to a steeper response function. The results are shown in Table 8. The

| | CIFAR10 | | | | |
|---|---|---|---|---|---|
| | DSGD | ASGD | TT/RL | TTv2 | RLv2 |
| MobileNetV2 | 95.28$\pm$0.20 | 94.34$\pm$0.27 | 95.05$\pm$0.11 | 95.20$\pm$0.14 | 95.26$\pm$0.03 |
| MobileNetV3S | 94.45$\pm$0.10 | 80.66$\pm$6.18 | 93.65$\pm$0.24 | 93.54$\pm$0.06 | 93.79$\pm$0.00 |
| MobileNetV3L | 95.95$\pm$0.08 | 80.79$\pm$2.97 | 95.39$\pm$0.27 | 95.27$\pm$0.09 | 95.33$\pm$0.08 |
| | CIFAR100 | | | | |
| | DSGD | ASGD | TT/RL | TTv2 | RLv2 |
| MobileNetV2 | 80.60$\pm$0.18 | 63.41$\pm$1.20 | 73.33$\pm$0.94 | 78.41$\pm$0.15 | 79.60$\pm$0.10 |
| MobileNetV3S | 78.94$\pm$0.05 | 51.79$\pm$1.05 | 71.14$\pm$0.93 | 74.51$\pm$0.37 | 75.39$\pm$0.00 |
| MobileNetV3L | 82.16$\pm$0.26 | 66.80$\pm$1.40 | 78.81$\pm$0.52 | 79.56$\pm$0.10 | 80.18$\pm$0.07 |

Table 6: Fine-tuning MobileNet models with *power response* on CIFAR10/100 datasets. Test accuracy is reported. `DSGD`, `ASGD`, and `TT` represent `Digital SGD`, `Analog SGD`, `Tiki-Taka`, respectively.

| | FCN | | | CNN | | |
|---|---|---|---|---|---|---|
| | DSGD | ASGD | TT | DSGD | ASGD | TT |
| $\sigma_c = 10\%$ | | $97.22{\pm}0.21$ | $97.66{\pm}0.04$ | | $92.68{\pm}0.45$ | $98.74{\pm}0.07$ |
| $\sigma_c = 30\%$ | | $96.97{\pm}0.12$ | $97.07{\pm}0.12$ | | $93.36{\pm}0.55$ | $98.89{\pm}0.05$ |
| $\sigma_c = 60\%$ | $98.17{\pm}0.05$ | $96.33{\pm}0.21$ | $97.70{\pm}0.09$ | $99.09{\pm}0.04$ | $93.07{\pm}0.53$ | $98.68{\pm}0.09$ |
| $\sigma_c = 90\%$ | | $95.99{\pm}0.15$ | $97.44{\pm}0.15$ | | $91.87{\pm}0.48$ | $98.92{\pm}0.02$ |
| $\sigma_c = 120\%$ | | $96.19{\pm}0.20$ | $96.97{\pm}0.20$ | | $91.57{\pm}0.58$ | $98.85{\pm}0.04$ |

Table 7: Test accuracy comparison under different cycle variation levels $\sigma_c$ on MNIST dataset. DSGD, ASGD, and TT represent `Digital SGD`, `Analog SGD`, `Tiki-Taka`, respectively

accuracy $< 15.00$ in the table implies that `Analog SGD` fails completely at all trials, which is close to random guess. The results show that `Analog SGD` works well only when the asymmetric is mild, i.e. $\gamma_{\text{res}}$ is small and $\tau$ is large, while `Tiki-Taka` outperforms `Analog SGD` and achieves comparable accuracy with `Digital SGD`.

| | | DSGD | Power response | | Exponential response | |
|---|---|---|---|---|---|---|
| | | | ASGD | TT/RL | ASGD | TT/RL |
| $\gamma_{\text{res}} = 0.5$ | $\tau = 0.6$ | | $96.01{\pm}0.26$ | $96.92{\pm}0.19$ | $<15.00$ | $97.27{\pm}0.07$ |
| | $\tau = 0.7$ | | $97.40{\pm}0.15$ | $97.05{\pm}0.05$ | $<15.00$ | $97.39{\pm}0.15$ |
| | $\tau = 0.8$ | | $97.38{\pm}0.10$ | $96.82{\pm}0.17$ | $94.00{\pm}0.63$ | $97.16{\pm}0.16$ |
| $\gamma_{\text{res}} = 1.0$ | $\tau = 0.6$ | $98.17{\pm}0.05$ | $<15.00$ | $97.39{\pm}0.05$ | $<15.00$ | $97.46{\pm}0.08$ |
| | $\tau = 0.7$ | | $<15.00$ | $97.33{\pm}0.05$ | $<15.00$ | $97.49{\pm}0.04$ |
| | $\tau = 0.8$ | | $<15.00$ | $97.34{\pm}0.09$ | $<15.00$ | $97.25{\pm}0.16$ |
| $\gamma_{\text{res}} = 2.0$ | $\tau = 0.6$ | | $<15.00$ | $96.93{\pm}0.15$ | $<15.00$ | $97.19{\pm}0.16$ |
| | $\tau = 0.7$ | | $<15.00$ | $97.27{\pm}0.02$ | $<15.00$ | $97.72{\pm}0.07$ |
| | $\tau = 0.8$ | | $<15.00$ | $97.18{\pm}0.04$ | $<15.00$ | $97.06{\pm}0.10$ |

Table 8: Test accuracy comparison under different response function parameters $\tau$ and $\gamma_{\text{res}}$ for FCN training on MNIST dataset with power or exponential response functions. DSGD, ASGD, and TT represent `Digital SGD`, `Analog SGD`, `Tiki-Taka`, respectively.

## L  Broader Impact

This paper focuses on developing a theoretical analysis for gradient-based training algorithms on a class of generic AIMC hardware, which can be leveraged to boost both energy and computational efficiency of training. While such efficiency gains could, in principle, enable broader and potentially unintended uses of machine learning models, we do not identify any specific societal risks that need to be highlighted in this context.

