# OpenReview forum: "Analog In-memory Training on General Non-ideal Resistive Elements: The Impact of Response Functions"
_NeurIPS.cc/2025/Conference — NeurIPS 2025 oral_

### Official Review · Reviewer_Uooc · 2025-07-01

**Clarity:** 3
**Significance:** 3
**Originality:** 4
**Rating:** 5
**Confidence:** 2

**Summary:**

The aim of this paper is to address nonlinearities involved in gradient-based training for Analog In-Memory Computing (AIMC) systems by proposing new methods for robust training. The authors have developed rigorous proofs to analyze the properties of the proposed residual learning methods and have demonstrated their working using simulation studies.

**Questions:**

While the proposed methods demonstrate performance improvements over the vanilla analog SGD, no discussion has been provided regarding its computational overhead. Is there any significant increase of computational complexity? How does it scale with the model size?

**Ethical Concerns:**

["NO or VERY MINOR ethics concerns only"]

**Final Justification:**

I have read the rebuttal and other reviewers' further comments. I maintain my support of this paper.

**Quality:**

3

**Strengths And Weaknesses:**

*** Strengths:

This paper primarily addresses two important types of nonliearities occurring in AIMC training: asymmetric weight update response, and other nonlinearities such as I/O reading errors. For the first problem, which is a key issue, the authors develop theoretical work to show that asymmetry influences training negatively by adding a penalty on the objective function, which provides a motivation for the development of the proposed methods.

The key contribution of this work is its  residual learning algorithm, which has a provable convergence to the critical point. This residual learning algorithm is further extended to deal with I/O reading errors.
The work provides an in-depth understanding of negative impacts of several nonliearities of the AIMC weight update process, and develops theoretically rigorous methods for reducing these negative impacts.

**** Weaknesses:

The experimental aspects of the work can be improved to make the proposed work more convincing.
Specifically, the application of the proposed residual learning algorithm and its variants have been only demonstrated on fairly small datasets for which, some modest performance gains have been observed compared to other recent methods.

More details of physical modeling of AMIC systems and weight updating process can be provided. For example, it is not clear what kind of I/O reading errors have been considered in the experimental studies.

---

> ### Author Rebuttal · Authors · 2025-07-31
>
> Thanks for acknowledging the theoretical contribution of our work and the support. We address other comments as follows.
>
> > W1: The experimental aspects of the work can be improved to make the proposed work more convincing. Specifically, the application of the proposed residual learning algorithm and its variants have been only demonstrated on fairly small datasets for which, some modest performance gains have been observed compared to other recent methods.
>
> Thanks for your valuable suggestion! We incorporate more simulations to make the paper more convincing, including training language models from scratch and their finetuning. Compared with the existing simulations, the additional ones involve deeper models and have more weights. The simulation results, presented in the response to **Q2 of reviewer GpyD**, show that the proposed approaches generalize well on more challenging tasks.
>
> ---
> > W2: More details of physical modeling of AMIC systems and weight updating process can be provided. For example, it is not clear what kind of I/O reading errors have been considered in the experimental studies.
>
> Thanks for your valuable suggestion. In all simulations, we adopt the following I/O settings. We will attach the details in the revision.
>
> Table T4.1: I/O setting in the paper
> | configuration | value |
> |---|---|
> | input bound | 1.0 |
> | input noise | None |
> | input resolution (DAC) | 7 bits |
> | output bound | 12.0 |
> | output noise | additive Gaussian noise $\mathcal{N}(0, 0.06^2)$ |
> | output resolution (ADC) | 9 bits |
>
>
> ---
> > Q1: While the proposed methods demonstrate performance improvements over the vanilla analog SGD, no discussion has been provided regarding its computational overhead. Is there any significant increase of computational complexity? How does it scale with the model size?
>
> Thanks for your construction question! We compare the computational overhead in the response to **Q4 in reviewer GpyD**. Although an extra residual array $P_k$ is introduced, the latency does not significantly increase. Furthermore, as existing work presents, the resistive crossbar array implements the MVM operator with $O(1)$ time complexity as the matrix size increases, ensuring its strong scalability [R1.5].
>
> ---
>
> We hope our response resolves your concerns. We are keen to engage in further discussion to ensure the clarity and rigor of our work. Thanks!

---

### Official Review · Reviewer_TGon · 2025-07-02

**Clarity:** 4
**Significance:** 4
**Originality:** 3
**Rating:** 5
**Confidence:** 3

**Summary:**

The authors provide a detailed analysis of the shortcomings of Analog SGD for training on AIMC hardware, and offer a learning algorithm that can converge to an exact critical point. They provide very detailed theoretical proofs, and very good motivation for the problem.

**Questions:**

I wonder if it would be possible to incorporate more simulations with more practical scenarios and other realistic imperfections in the rebuttal period. If the authors were able to achieve more significant and convincing improvements, as the theory seems to entail, it would make the paper extremely strong. As of now , although very good, it feels "top-heavy" although the challenges of simulating realistic hardware imperfections and noise are well-noted.

**Ethical Concerns:**

["NO or VERY MINOR ethics concerns only"]

**Final Justification:**

The authors have added simulations which was in my opinion the major missing component from this work.

**Limitations:**

Yes

**Quality:**

4

**Strengths And Weaknesses:**

This is an extremely strong paper theoretically and is very well written, well presented and well-motivated. In light of the foundational nature of the problems they highlight (for e.g. analog SGD, and for AIMC training in general) I would have expected more significant improvements on the simulations. As the authors note, this is perhaps due to the artificial nature of simulating hardware imperfections, and realistic noise (a famously hard problem!)

---

> ### Author Rebuttal · Authors · 2025-07-31
>
> Thanks for acknowledging the theoretical contribution of our work and the support. We address some other comments as follows.
>
> > Q1: I wonder if it would be possible to incorporate more simulations with more practical scenarios and other realistic imperfections in the rebuttal period.
>
> Thanks for your constructive suggestion! Following your suggestion, we incorporated more simulations to reflect more practical setting challenges in analog training:
>
> **1) More challenging machine learning tasks.** We conducted a series of additional simulations on various tasks, including training language models from scratch and their finetuning. Compared with the existing simulation results, the additional ones involve deeper models and more parameters in the analog domain. The simulation results, presented in the response to **Q2 in reviewer GpyD**, show that the proposed approaches generalize well on more challenging tasks.
>
> **2) Validation on real resistive element configuration.** To validate the approaches in real resistive elements, we performed simulations on various resistive elements, including ReRAM, EcRAM, and EcRAM-MO. The training on them is more challenging since they suffer from the asymmetric update issue, and they have limited update granularity and dynamic ranges. The simulation results, presented in the response **Q1 in reviewer Kwur**, show that it is promising to apply the proposed methods on real resistive elements.
>
> We hope our response resolves your concerns. We are keen to engage in further discussion to ensure the clarity and rigor of our work. Thanks!

---

### Official Review · Reviewer_Kwur · 2025-07-03

**Clarity:** 4
**Significance:** 3
**Originality:** 4
**Rating:** 5
**Confidence:** 3

**Summary:**

This paper addresses the question of how to overcome the fact that parameter updates in analog in-memory computing have non-linear responses to update pulses. They find that asymmetric response functions negatively impact analog SGD and propose residual learning as a solution. They evaluate their approach with numerical experiments and generally reach a good match to digital SGD.

The main contributions are:
- To propose an approximate discrete-time dynamic to the analog pulse update and to study the impact of response functions based on it
- They show that means that analog SGD implicitly optimizes for a penalized objective. This means that it cannot converge to the optimum of the non-penalized loss
- To address this they propose a "Residual Learning" algorithm, which they claim provably converges to a critical point under assumptions of strong convexity and expand this to include additional concerns that could occur in analog in memory computing hardware.

**Questions:**

- How would you validate your approach on either a hardware model or actual hardware?
- Can you illustrate the difference in the algorithms in a case where the assumptions of the theoretical results are met?

**Ethical Concerns:**

["NO or VERY MINOR ethics concerns only"]

**Final Justification:**

This paper would be a good fit for neurips. I recommend its acceptance in particular in light of the updates to the results.

**Limitations:**

Sufficiently discussed.

**Quality:**

4

**Strengths And Weaknesses:**

The paper addresses an important problem: How to deal with the fact that analog in memory computing can only approximately implement SGD updates.

Strength:
- The identified problem in analog learning is crucial to overcome for AIMC to succeed.
- Thorough theoretical analysis of Analog SGD.
- Well motivated and executed analyses of problem and solution.
- "Residual Learning" and its extension appears to be a novel solution.
- Numerical evaluation on multiple datasets and models (MNIST, CIFAR10, CIFAR100) and (FC/Conv, Resnet), among others demonstrates the ability of the method to achieve Digital SGD level of performance.

Weaknesses:
- No evaluation on any actual analog hardware
- Limited evaluation on more larger machine learning datasets like Imagenet
- No evaluation on a specific simulated analog hardware
- The paper would benefit from an illustration of the proposed alternative algorithm on toy datasets that meet the assumptions of the theoretical results.

---

> ### Author Rebuttal · Authors · 2025-07-31
>
> Thanks for acknowledging that **the identified problem in this paper is crucial to overcome for successful training on AIMC hardware**. We address other comments as follows.
>
> ---
> > W1 & W3 & Q1. No evaluation on any actual analog hardware. No evaluation on a specific simulated analog hardware. How would you validate your approach on either a hardware model or actual hardware?
>
> Thanks for your constructive suggestion! As mentioned in the response to **Q5 of Reviewer GpyD**, our work builds upon a wide range of empirical study [45-59] and all our simulations are done on AIHWKIT simulator [44]. Since the AIHWKIT already captures most of the known hardware imperfections, the conclusion from our paper should be able to generalize to actual hardware.
>
> Although we do not perform it in this work, we will discuss how to validate the proposed approach on actual hardware. We will follow the methods presented by [R2.1], which proposes a device-algorithm co-optimization framework that validates the Tiki-Taka [22-23] on HfO$_2$-based ReRAM with 14nm CMOS technology. Similar to them, we will first need to measure the response curve of a given resistive element (e.g., ReRAM) and simulate its behaviour in the analog hardware simulator. We then validate that the proposed approach can achieve comparable accuracy with digital training benchmarks on different machine learning tasks. Based on the simulation results, we can optimize the hardware to reproduce the simulation results on actual hardware.
>
> As the first step, we validate the proposed approaches on two real resistive element configurations. The results presented in Table T2.1 suggest that the proposed algorithm works well on various real resistive elements. The rest of the hardware validation involves chip design, fabrication, and extensive testing. We will do so in future work.
>
> Table T2.1: Test accuracy of training FCN models on MNIST dataset
> | Digital SGD | ReRAM [R2.1] | EcRAM [52] | EcRAM-MO [R2.2] |
> |---|---|---|---|
> | 98.17 | 96.10 | 97.55 | 96.76 |
>
>
> ---
> > W2. Limited evaluation on more larger machine learning datasets like Imagenet.
>
> Thanks for your valuable suggestion! We include additional simulations in the revision to make our paper more convincing. In the response to **Q2 of the reviewer GpyD**, we present additional simulation results in the language model training during the limited rebuttal period, including training from scratch and finetuning tasks. Since the simulations on ImageNet training take a longer time, we will complete them after the rebuttal and add them in the revision.
>
> ---
> > W4 & Q2. The paper would benefit from an illustration of the proposed alternative algorithm on toy datasets that meet the assumptions of the theoretical results.
>
>
> Thanks for your constructive suggestions! We will add a new subsection to discuss and illustrate the algorithms at the end of Section 4. We consider a toy regression problem $\min_{W\in{R}^2}||W-W^*||^2$, whose objective is smooth and strongly convex.
>
> We choose a symmetric point $W^\diamond\ne W^*$. We add a Gaussian noise with mean 0 and variance $\sigma^2\ne 0$ to the gradients. We plot the convergence trajectories of Analog SGD (equation (2)) and Residual Learning (equation (10)-(11)).
>
> We show that the weights generated by Analog SGD are attracted toward the symmetric point $W^\diamond$, as suggested by Theorem 1. Instead, Residual Learning converges to the optimal point and the auxiliary array $P_k$ is a good approximation of the residual $\frac{W^*-W_k}{\gamma}$, as suggested by the theory. We will discuss this example in the revision to provide more insight.
>
> ---
> We hope our response resolves the concerns. Thanks!
>
> ---
> [R2.1] Gong, et al. Deep learning acceleration in 14nm CMOS compatible ReRAM array: device, material and algorithm co-optimization. IEDM 2022.
>
> [R2.2] Kim, et al. Metal-oxide based, CMOS-compatible ECRAM for deep learning accelerator. IEDM 2019.

---

> > ### Comment · Reviewer_Kwur · 2025-08-05
> >
> > Thank you for addressing my questions. I will maintain my score.

---

> > > ### Author Response · Authors · 2025-08-07
> > >
> > > Thank you very much for your valuable feedback, acknowledging our response, and engaging in the discussion.
> > >
> > > Sincerely, authors.

---

### Official Review · Reviewer_GpyD · 2025-07-13

**Clarity:** 3
**Significance:** 4
**Originality:** 2
**Rating:** 5
**Confidence:** 3

**Summary:**

This paper investigates the theoretical foundations of gradient-based training on analog in-memory computing (AIMC) hardware, specifically focusing on how non-ideal response functions of resistive elements affect training dynamics. The authors develop a mathematical framework to analyze Analog SGD and propose a "Residual Learning" algorithm to address the convergence issues caused by asymmetric response functions. The key contributions include a discrete-time model for analog weight updates, theoretical proof that Analog SGD implicitly optimizes a penalized objective leading to inexact convergence, a Residual Learning algorithm that provably converges to critical points through bilevel optimization, and extensions to handle additional hardware imperfections like limited response granularity and noisy I/O.

**Questions:**

Please see the weaknesses above. Additionally, I am curious about a few questions:

1. Can the authors validate if the idealized response function model can capture most hardware complexities encountered in practice?

2. The analysis focuses primarily on resistive crossbar arrays. I am curious if it can generalize to other emerging analog computing paradigms.

**Ethical Concerns:**

["NO or VERY MINOR ethics concerns only"]

**Final Justification:**

The authors resolved most of my concerns, and hence, I increase my rating to 5.

**Limitations:**

yes

**Quality:**

3

**Strengths And Weaknesses:**

Strengths:

1. The paper provides a strong theoretical foundation with rigorous mathematical analysis of a practically important problem. The bilevel optimization formulation for Residual Learning is theoretically elegant and well-motivated, supported by comprehensive proofs with detailed assumptions and convergence guarantees.

2. The proposed response function model captures actual hardware behavior observed in various resistive memory technologies including PCM, ReRAM, and ECRAM.

3. The identification of implicit penalty effects in Analog SGD provides novel and valuable insights, particularly the connection between hardware asymmetries and optimization bias.

4. The authors demonstrate methodological rigor through careful treatment of different hardware imperfections, appropriate use of assumptions with clear justifications, and good experimental validation of theoretical predictions using standard datasets.

Weaknesses:

1. The experimental scope is somewhat limited, as simulations are conducted using AIHWKIT rather than actual hardware, which constrains real-world validation.

2. The dataset scale remains relatively modest with MNIST and CIFAR experiments, raising questions about scalability to larger, modern datasets (however, most papers on analog training adopt these small-scale datasets as benchmarks)

3. The strong convexity assumption for main convergence results is quite restrictive for neural networks, while the bounded saturation assumption may not hold as hardware degrades over time. The idealized response function model may not capture all hardware complexities encountered in practice.

4. The Residual Learning approach introduces computational overhead by requiring maintenance of an additional array and computing gradients on mixed weights.

---

> ### Author Rebuttal · Authors · 2025-07-31
>
> Thanks for acknowledging that **this paper provides a strong theoretical foundation with rigorous mathematical analysis**, and the support. We address your comments as follows.
>
>
> ---
> > Q1: The experimental scope is somewhat limited, as simulations are conducted using AIHWKIT rather than actual hardware, which constrains real-world validation.
>
>
> Thanks for your valuable suggestion! Analog in-memory computing is an emerging technology that attracts researchers from material science, device design, chip design, computer systems, and AI algorithm design. Our focus was mainly on the algorithm design for analog in-memory computing, where evaluation using off-the-shelf hardware simulators is very common. Since the reliable devices for analog in-memory training are not commercially available, we plan to leave the hardware evaluation for future work. Nonetheless, we still perform additional simulations to validate the approaches in real resistive element configurations, including ReRAM, EcRAM, and EcRAM-MO. The simulation results, presented in the response to **Q1 of reviewer Kwur**, demonstrate the potential in real-world applications.
>
> ---
> > Q2. The dataset scale remains relatively modest with MNIST and CIFAR experiments, raising questions about scalability to larger, modern datasets (however, most papers on analog training adopt these small-scale datasets as benchmarks)
>
> Thanks for your valuable suggestion. We evaluate the scalability by investigating their performance on more training tasks, where the models are deeper or have more weights. The preliminary results below indicate that analog training is promising for scaling to larger and more modern datasets and tasks.
>
> **Task 1: Training language models from scratch.**
> We train a Transformer model from scratch on the *Coriolanus* dataset, a play by Shakespeare. The model consists of 6 layers with ~10 million parameters. The weights for all query, key, value, and output projections in attention layers, as well as linear layers in feed-forward networks, are updated in the analog domain. The other parameters, such as token/position embeddings, scale/bias in layer normalization layers, are put and updated in the digital domain. For a fair comparison, we do not use an adaptive learning rate for either digital or analog training. In Table T1.1, we reach the same conclusion that Residual Learning outperforms Analog SGD significantly and is comparable with Digital SGD.
>
> Table T1.1: Loss (lower is better)
> | Digital SGD | Analog SGD | Residual Learning v1 | Residual Learning v2 |
> |---|---|---|---|
> | 2.46 | 3.10 | 2.67 | 2.61 |
>
> **Task 2: Fine-tuning language models.**
> We finetune a pre-training tiny MobilieBERT model with 17M parameters on the MRPC dataset via LoRA training. A total of 462 LoRA layers and 0.99M trainable parameters are introduced. We implemented different portions of the LoRA layers on analog arrays, while the others were left on digital hardware and trained using AdamW. The algorithms reduce to digital Adam if the percentage is 0%, while they reduce to a pure analog algorithm (Analog SGD or Residual Learning) when the percentage is 100%.
>
> Table T1.2: Accuracy (higher is better)
> | Analog percentage | Analog SGD | Residual Learning |
> |---|---|---|
> | 0% (Digital training) | 87.21 | |
> | 20% | 87.00 | **88.72** |
> | 40% | 86.02 | **87.00** |
> | 60% | 84.06 | **85.06** |
> | 80% | 78.01 | **80.01** |
> | 100% | 76.63 | **77.69** |
>
> Table T1.3: F1 score (higher is better)
> | Analog percent | Analog SGD | Residual Learning |
> |---|---|---|
> | 0% (Digital training) | 91.03 | |
> | 20% | 90.94 | **91.01** |
> | 40% | 90.35 | **90.87** |
> | 60% | 88.61 | **89.51** |
> | 80% | 81.61 | **84.35** |
> | 100% | 77.67 | **80.95** |
>
> An interesting observation is that Residual Learning achieves slightly higher accuracy than its digital counterpart when the analog percentage is 20%. We speculate that it happens since the analog imperfection implicitly applies a regularization on the training, leading to a better generalization performance. It is worth investigating in future work.
>
> ---
> > Q3. The strong convexity assumption for the main convergence results is quite restrictive for neural networks.
>
> Thank you for pointing it out. Following nonconvex bilevel optimization work [R2.1-R2.2], the strong convexity assumed in the current paper can be relaxed to the Polyak-Lojasiewicz (PL) condition with similar conclusions, which accommodates nonconvex landscapes such as those in overparameterized neural networks [R2.3]. We will discuss it in the revision.
>
> ---
> > Q4. The bounded saturation assumption may not hold as hardware degrades over time.
>
> Thank you for pointing it out. We did not consider hardware degradation over time since typical training processes for moderate-scale models or fine-tuning large models only take hours to days, during which hardware degradation is not a dominant imperfection. It is worth considering in large-scale model training tasks if the training takes a longer time, like weeks to months. We plan to relax this assumption in our future work.
>
> ---
> > Q5. Can the authors validate if the idealized response function model can capture most hardware complexities encountered in practice?
>
> Thanks for your valuable suggestion. As we discuss in Appendix A, a series of empirical works measured the response curves on various resistive elements [45-59], which demonstrate that the non-linear and asymmetric response is one of the main challenges of analog training. On top of that, AIHWKIT [44] provides a fine-grained simulator of the analog hardware, which includes most of the known hardware imperfections like the IO noise, limited granularity, device variation, and quantization error in analog-digital conversion. **Building upon their work, the validation in this paper can capture most hardware complexities.**
>
> To further validate the proposed idealized response function model, we compare two settings:
> - **S1)** training models following the proposed dynamics (5);
> - **S2)** training models following the standard settings in the AIHWKIT. See Appendix K and *Table 4.1 in the response to the Reviewer Uooc* for the concrete configurations.
>
> Table T1.4 presents the simulation results. The results show that the accuracy of S1 and S2 is similar for both Analog SGD and Residual Learning. It demonstrates that the proposed response function model captures the training complexities in practice well. In response to your feedback, we will discuss it in the revision.
>
> Table T1.4: Comparison of S1 and S2 in the CIFAR10 training task
> |  | ASGD-S1 | ASGD-S2 | RL-S1 | RL-S2 |
> |---|---|---|---|---|
> | Resnet18 | 87.21 | 84.87 | 95.04 | 94.81 |
> | Resnet34 | 95.56 | 95.43 | 96.31 | 96.29 |
> | Resnet50 | 95.50 | 94.36 | 96.50 | 96.34 |
>
> ---
> > Q6. The Residual Learning approach introduces computational overhead by requiring maintenance of an additional array and computing gradients on mixed weights.
>
> While Residual Learning introduces extra computational overhead, it is affordable in practice. Compared to Analog SGD, the analog memory requirement doubles, but the latency remains almost unchanged since Residual Learning does not explicitly compute the mixed weights during the forward and backward passes. As [R1.4] suggests, $W_k$ and $P_k$ can share a same analog-digital convertor (ADC), which implements the weight mixing without introducing extra latency. On the other hand, as suggested by [22] in our paper, the forward, backward, and update on $W_k$ and $P_k$ are performed in parallel, which avoids significant latency increase. Consequently, *introducing an extra residual array does not incur substantial extra latency*. Following the evaluation in Table 1 in [24], we compared the latency of Analog SGD and Residual Learning in Table T1.5. The results suggest that the extra overhead is mild. We will discuss it in the revision.
>
> Table T1.5: Comparison of time (nano-second) consumption in each layer
> |  | Analog SGD | Residual Learning |
> |---|---|---|
> | Forward/backward [R1.5] | 40.0 | 40.0 |
> | update                  | 30.9 | 50.8 |
>
> ---
> We hope our response resolves your concerns. Thanks!
>
> ---
>
>
> [R1.1] Shen, et al. On penalty-based bilevel gradient descent method. ICML 2023.
>
> [R1.2] Kwon, et al. On penalty methods for nonconvex bilevel optimization and first-order stochastic approximation. ICLR 2024.
>
> [R1.3] Liu, et al. Loss landscapes and optimization in over-parameterized non-linear systems and neural networks. 2022.
>
> [R1.4] Song, et al. Programming memristor arrays with arbitrarily high precision for analog computing. Science 2024.
>
> [R1.5] Jain, et al. A heterogeneous and programmable compute-in-memory accelerator architecture for analog-ai using dense 2-D mesh. VLSI 2022.

---

> > ### Comment · Reviewer_GpyD · 2025-08-05
> > **Reviewer response**
> >
> > Thanks for the authors' response, and most of my concerns are resolved. It would be great if the authors can update the new results in the revised version. I would increase my score.

---

> > > ### Author Response · Authors · 2025-08-05
> > >
> > > Thank you very much for acknowledging our response, engaging in the discussion, and updating your score. We will incorporate the new results in the revisions.
> > >
> > > Sincerely, authors.

---

### Decision · Program_Chairs · 2025-09-17

**Decision:**

Accept (oral)

**Comment:**

The paper considers training on analog in-memory hardware, providing a theoretical framework for gradient-based training on such hardware in the presence of non-idealities in response functions. This framework is used to show that these non-idealities negatively affect SGD, and as a mitigation, the paper proposes a residual learning algorithm that is shown to converge provably and through simulations.

Reviewers are unanimous in their positive opinion of the paper, with all four giving it a score of 5. AC concurs and enthusiastically recommends acceptance.